# QUALITY OF REJECTIONS MATTERS: PREFERENCE DATA CONSTRUCTION FOR FAITHFUL SUMMARIZATION

## ABSTRACT

Factual reliability is a central challenge in abstractive summarization, as LLMs continue to generate hallucinations. A widely adopted solution is preference optimization, training models to prefer faithful over unfaithful summaries, but prior work emphasized the quantity of rejections over their alignment quality. In this paper, we show that effective alignment in summarization arises when rejected summaries achieve high alignment potential, characterized by a small preference margin that keeps rejections non-trivial and a large factuality margin that enforces clear factual contrast. Through both theoretical analysis and controlled prompting, we show that three factors, hallucination level, summary length, and prompt complexity, critically shape alignment potential. Building on these insights, we propose SPICE, a simple prompting strategy that produces rejected summaries with strong alignment value. Across diverse model scales and alignment algorithms, SPICE consistently achieves superior factuality without sacrificing coherence, relevance, or abstractiveness, outperforming existing rejection strategies.

## 1 INTRODUCTION

Recent advances in large language models (LLMs) have substantially advanced text summarization, producing outputs that are more coherent, fluent, and stylistically natural than ever before (Pu et al., 2023; Song et al., 2025). Nevertheless, a persistent and fundamental challenge is *hallucination*, where generated summaries contain information inconsistent with the source text (Maynez et al., 2020; Song et al., 2024; Lee et al., 2024). Even high-performing proprietary LLM continue to exhibit non-trivial rates of hallucination at the sentence level (Song et al., 2024), underscoring that fluency does not guarantee factuality. This phenomenon reflects an alignment gap, a divergence between model outputs and source fidelity that undermines summary reliability.

A common remedy is preference optimization, where models learn to favor *chosen* (factually consistent) summaries over *rejected* (inconsistent) ones (Stiennon et al., 2020; Ouyang et al., 2022). However, collecting human feedback at scale is prohibitively costly, since annotators must carefully verify consistency against long source documents (Mishra et al., 2024; Song et al., 2025). To bypass this bottleneck, recent studies instead generate preference pairs automatically. *Model-based* methods (Choi et al., 2024; Song et al., 2025) sample multiple summaries from LLMs, score them with automated evaluators such as G-Eval (Liu et al., 2023) and FineSuRE (Song et al., 2024), and construct preference pairs by selecting high-scoring candidates as chosen via Best-of-$N$ and low-scoring ones as rejected via Worst-of-$N$. In contrast, *Reference-based* methods (Cao & Wang, 2021; Mishra et al., 2024) construct rejected summaries by synthetically editing a faithful, chosen reference summary by injecting or altering factual information using language models.

As rejected summaries are comparatively easier to generate, prior work has placed greater emphasis on generating rejections, often favoring quantity over quality under the assumption that more negatives yield stronger training signals (Cao & Wang, 2021; Mishra et al., 2024; Xie et al., 2025). Yet this focus on quantity leaves a crucial question underexplored: "whether these rejections truly provide the right signal?" That is, rejections should be plentiful and exhibit sufficient alignment potential, which we define as the joint condition where the difference between the factuality margin and the preference margin is large—the rejection remains difficult for the model to distinguish from the chosen summary before training (small preference margin) while still presenting a clear factual contrast with the chosen summary (large factuality margin) (refer to Section 3.1 for details).

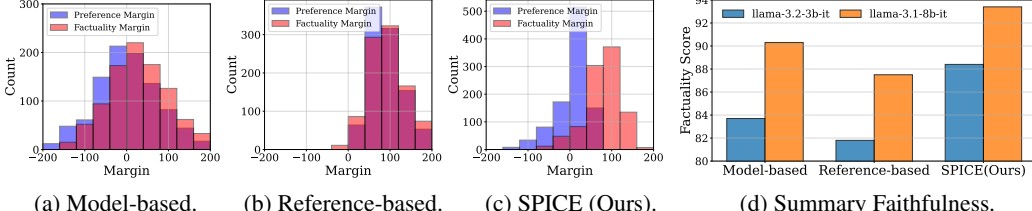

(a) Model-based.  (b) Reference-based.  (c) SPICE (Ours).  (d) Summary Faithfulness.

Figure 1: Comparison of (c) our proposed SPICE with (a) a model-based method, SummLlama (Song et al., 2025), and (b) a reference-based method, SYNFAC-EDIT (Mishra et al., 2024), w.r.t preference and factuality margins between generated rejected summaries and their corresponding chosen summaries. (d) denotes the average factuality score of summaries generated by two Llama-series models trained using DPO (Rafailov et al., 2023a) on preference pairs from three methods.

However, our analysis in Figure 1 reveals that neither approach consistently produces rejections that satisfy both conditions. Instead, the model-based method tends to produce rejections that closely resemble the chosen summary, resulting in only subtle hallucinations and thus small preference and factuality margins. By contrast, the reference-based method seeks strong contrast with the chosen summary, but in doing so often introduces obvious factual errors, producing large preference and factuality margins that make the rejections trivial to separate. These results show that existing methods fail to create *informative rejections*, which are summaries that mislead the model into treating them as plausible while still being factually distinct enough to reveal source misrepresentations.

In this paper, we theoretically and empirically establish what constitutes effective rejected summaries for faithfulness alignment through the notion of alignment potential, and in doing so identify key factors. First, the *hallucination level* in the rejected summary determines its factual distance from the chosen one, directly influencing the factuality margin. Second, the *summary length* and *prompt complexity* significantly impact the preference margin, as longer or more complex generations tend to be easier for the model to distinguish. Crucially, by controlling the target factor while isolating the others, we demonstrate that targeted margin control not only shapes alignment potential but also translates directly into substantial gains in summary faithfulness after alignment tuning, offering a principled recipe for constructing high-quality preference data.

Building on this foundation, we introduce SPICE (Simple Prompt for Informative and Contrastive Examples), a simple yet effective prompting strategy that generates rejected summaries with high hallucination, matched length, and minimal prompt instructions, thereby maximizing "alignment potential"—achieving a small preference margin and a large factuality margin in Figure 1(c), such that the gap between the two margins is maximized. This enlarged margin gap strengthens the alignment potential by ensuring that the rejected example is not only clearly incorrect (high factuality margin) but also non-trivially distinguishable from the chosen example (small preference margin), leading to more faithful text summarization.[v7Lg, jYGQ] Extensive experiments comparing SPICE with diverse rejection strategies, including rule-based methods as well as recent LLM-based approaches such as SummLlama (Song et al., 2025) and SYNFAC-EDIT (Mishra et al., 2024), demonstrate that SPICE substantially improves summary faithfulness across model sizes and alignment methods, including DPO (Rafailov et al., 2023a), SimPO (Meng et al., 2024), and KTO (Ethayarajh et al., 2024). Moreover, SPICE achieves these faithfulness gains without incurring an alignment tax: coherence, relevance, and abstractiveness remain comparable or even slightly improved after alignment tuning.

Our main contributions are summarized as follows:

- We present the first work to identify which rejected summaries are most effective for improving faithfulness in summarization alignment, grounded in a theoretical and empirical analysis that links preference and factuality margins to faithfulness gains.

- We introduce SPICE, a simple yet effective prompting strategy to generate high-quality rejected summaries that maximize alignment potential, *i.e., the gap between factuality and preference margin.*[v7Lg, jYGQ]

- We conduct extensive experiments comparing rule-based, model-based, and reference-based rejection strategies, analyzing their impact on faithfulness improvement.

- We release a high-quality dataset for faithfulness alignment to support the development of more effective summarization models.

## 2  RELATED WORK

**Text Summarization.** Text summarization is a long-standing task, but recent advances amplify concerns the factual reliability of generated summaries (Chen et al., 2022; Ji et al., 2023). There has been substantial effort to improve the faithfulness of summaries generated by language models. Earlier methods (Chen et al., 2022; Song et al., 2023) aimed to close the gap between generated and reference summaries by optimizing for ROUGE (Lin, 2004) and BERTScore (Zhang et al., 2019). A complementary line of work improves factuality by using unfaithful summaries as negative samples in contrastive learning (Liu & Liu, 2021; Cao & Wang, 2021; Xu et al., 2022), as in CLIFF (Cao & Wang, 2021), which leverages techniques like entity swapping to boost consistency.

With the advent of LLMs, recent efforts have increasingly focused on enhancing summary faithfulness via instruction tuning (Zhang et al., 2023; Zhou et al., 2023) and preference optimization (Stiennon et al., 2020; Lee et al., 2023). In this context, two approaches, including model- and reference-based methods, aim at constructing large chosen–rejected summary pairs for factuality alignment (Choi et al., 2024; Mishra et al., 2024; Song et al., 2025). While they improve faithfulness, which types of rejected summaries best support alignment remains unclear and underexplored.

**Preference Optimization.** Preference optimization serves as an effective approach to align language models with human intention (Jiang et al., 2024; Ryu et al., 2024). Two primary methods dominate this field: RM-based approaches such as PPO (Schulman et al., 2017), which guides model outputs using a reward model (RM) to reinforce desirable responses, and RM-free methods such as DPO (Rafailov et al., 2023a), SimPO (Meng et al., 2024), and KTO (Ethayarajh et al., 2024), which directly learns from preference comparisons without an explicit reward model. In the summarization task, Stiennon et al. (2020) collected pairwise human preferences on Reddit posts for PPO training, while SYNFAC-EDIT (Mishra et al., 2024) and SummLlama (Song et al., 2025) constructed synthetic chosen–rejected pairs and used them for DPO training to enhance faithful summary.

**Preference Data Quality.** High-quality preference pairs are critical for alignment, as it sharpens a model's ability to separate preferred from rejected responses (Stiennon et al., 2020; Gao et al., 2024; Duan et al., 2024). To reduce noise in preference data, recent work has explored filtering strategies (Khaki et al., 2024; Pace et al., 2024; Liu et al., 2024a), with growing attention to improving rejected responses (Liu et al., 2024c; Qiao et al., 2024; Xie et al., 2025). The quality is typically assessed by the external reward margin (score gap from external evaluators) (Khaki et al., 2024; Song et al., 2025), the internal policy margin (log-probability gap from the model itself) (Pace et al., 2024; Muldrew et al., 2024), or both (Liu et al., 2024a; Huang et al., 2025). While the most effective combination of these margins remains uncertain (Zhou et al., 2024; Cui et al., 2025), we systematically study how varying them influences faithfulness alignment in summarization.

**Hard Negative Mining.** Hard negative mining emphasizes that negatives should be both difficult and informative, an intuition that is conceptually related to our notion of alignment potential (Liu & Liu, 2021; Gao et al., 2021; Tang et al., 2022; Sun et al., 2023). However, in faithful summarization, what actually constitutes a hard negative for preference optimization has remained unclear, and prior work has largely focused on the quantity of rejected summaries (Mishra et al., 2024; Song et al., 2025). In contrast, our work is the first to clarify this, providing both theoretical and empirical evidence that effective rejected summaries are characterized by the factuality and preference margins.[v7Lg]

## 3  FORMULATION AND THEORETICAL ANALYSIS

### 3.1  TWO METRICS OF ALIGNMENT POTENTIAL FOR FAITHFUL SUMMARIZATION

The notion of alignment potential is a measure of how much training value a preference pair provides (Huang et al., 2025; Deng et al., 2025). While it was originally introduced in general preference optimization, we extend it to faithful summarization. In our context, an ideal summary model is expected to produce outputs where its implicit preference margin exactly matches the true factuality score difference between the chosen and rejected summaries. Real models, however, often diverge from this ideal, failing to separate faithful from unfaithful summaries. Thus, we capture this discrepancy by defining the *preference* and the *factuality margins*, which together quantify alignment potential.

Let $x$ be the text document to be summarized, and let $y_w$ and $y_l$ be its chosen and rejected summaries. The two margins are defined as follows.

**Preference Margin.** Let $\pi_\theta$ be the summarization model prior to faithfulness alignment training, *i.e.*, the policy model for preference optimization. Then, the preference margin $\Delta_{\text{pref}}$ between chosen and rejected summaries is computed by:

$$\Delta_{\text{pref}} = r_\theta(y_w; x) - r_\theta(y_l; x), \tag{1}$$

$$\text{where } r_\theta(y; x) = \sum_{t=1}^{|y|} \log \pi_\theta(y_t | x, y_{<t}), \tag{2}$$

and $y_t$ is the $t$-th token in the summary $y$.[1] While we do not take the absolute for analysis, a small margin reflects closeness to zero, not a negative.

**Factuality Margin.** Suppose that $f(\cdot)$ is an optimal external model that provides accurate assessments of the faithfulness of a summary $y$ grounded on a source document $x$. Then, the factuality margin $\Delta_{\text{fact}}$ between chosen and rejected summaries is computed by:

$$\Delta_{\text{fact}} = f(y_w; x) - f(y_l; x). \tag{3}$$

This margin can be computed by either prompting an external model to assign scores to each response, as in G-Eval (Liu et al., 2023) or FineSuRE (Song et al., 2024), and then taking their difference, or by directly replacing the policy model with an external model and calculating the same implicit margin using its probabilities (Shi et al., 2024; Huang et al., 2025).

Then, the discrepancy between the margin of the policy model and that of the external evaluator is translated into the *alignment potential*, highlighting how much factual distinction the policy model still fails to capture and thus how much room remains for alignment as:

$$|\Delta_{\text{fact}} - \Delta_{\text{pref}}| = \Big| \underbrace{\big(f(y_w; x) - f(y_l; x)\big)}_{\text{factuality margin}} - \underbrace{\big(r_\theta(y_w; x) - r_\theta(y_l; x)\big)}_{\text{preference margin}} \Big|. \tag{4}$$

Consequently, our objective is to design a method for generating rejected summaries $y_l$ such that, given that the chosen summary $y_w$ is fixed, the margin discrepancy $|\Delta_{\text{fact}} - \Delta_{\text{pref}}|$ is maximized. Unlike prior work (Mishra et al., 2024; Song et al., 2025), our approach is the first to explicitly target the quality of rejections by maximizing their alignment potential in faithful summarization.

## 3.2 THEORETICAL ANALYSIS

We perform a theoretical analysis to show the effectiveness of an adversarial sampling approach, which preferentially samples rejected summaries with larger factuality-preference margin discrepancies, thus providing stronger alignment signals.

**Setting.** Consistent with other theoretical work (Rosset et al., 2024; Shi et al., 2025; Huang et al., 2025), we analyze the convergence behavior under the contextual bandit setup when training with DPO using stochastic gradient descent (SGD), comparing how quickly our alignment objective can be achieved when sampling rejected summaries adversarially vs. uniformly. Hence, to quantify this, we define the pairwise squared margin error (SME), and set the objective as minimizing the mean squared margin error (MSME) with fixed $y_w$:

$$\text{MSME}(\theta \mid y_w) = \frac{1}{|\mathcal{Y}|} \sum_{y_l \in \mathcal{Y}} \text{SME}_\theta(x, y_w, y_l), \tag{5}$$

$$\text{where } \text{SME}_\theta(x, y_w, y_l) = \Big( \big(f(y_w; x) - f(y_l; x)\big) - \big(r_\theta(y_w; x) - r_\theta(y_l; x)\big) \Big)^2. \tag{6}$$

**Uniform vs. Adversarial Sampling.** We compare two approaches for sampling losers (rejected summaries) given a fixed winner $y_w$: *(i) uniform sampling*, where $y_l$ is drawn uniformly from $\mathcal{Y}$, and *(ii) adversarial sampling*, where $y_l$ is drawn with probability proportional to its margin discrepancy $|\Delta_{\text{fact}} - \Delta_{\text{pref}}|$. Our analysis shows that adversarial sampling converges faster by prioritizing losers with large factuality margin, yielding strong corrective signals for reducing MSME.

---

[1]Consistent with the DPO objective (Rafailov et al., 2023a), we compute sequence-level log-probability differences without length normalization. However, whereas DPO normalizes by a reference policy, we assume a uniform reference and omit this term since our goal is data construction rather than training (see Appendix A.1).

**Analysis.** Assume the distribution of factuality-preference margin discrepancies is approximately normal. Let $\Delta\text{MSME}_{\text{uni}}$, $\Delta\text{MSME}_{\text{hard-adv}}$ be the expected reduction in MSME per SGD step under uniform and hard adversarial sampling, respectively, where hard adversarial sampling deterministically selects the sample that maximizes the margin discrepancy. By computing the expectation of error reduction under both schemes, we can show (refer to the detailed proof in Appendix B):

$$\mathbb{E}_{\text{hard-adv}}[\Delta\text{MSME}] \geq \frac{4}{3}\,\mathbb{E}_{\text{uni}}[\Delta\text{MSME}]\,. \quad . \tag{7}$$

Therefore, hard adversarial sampling converges at least $33.3\%$ faster than uniform sampling, since it prioritizes losers with the larger margin discrepancies, which provide the strongest corrective signals for reducing MSME. $\square$

This result highlights that, when constructing preference pairs, favoring rejected summaries with high alignment potential in Eq. (4) yields high-quality preference pairs, as they effectively reduce the gap between the policy model's preference margin and the ideal factuality margin.

# 4 CONTROLLING MARGINS VIA LLM PROMPTING

Our theoretical analysis establishes that the effectiveness of preference optimization critically depends on the preference and factuality margins between chosen and rejected summaries. To operationalize this insight, we next investigate how these margins can be systematically controlled during data construction using LLMs. In particular, we identify three prompt-based factors that directly influence the resulting margins: (i) hallucination level, (ii) summary length, and (iii) prompt complexity.

## 4.1 BASE DATASET WITH CHOSEN SUMMARY

**Dataset.** To study how prompting can control factuality and preference margins in text summarization, we first construct a dataset of source documents paired with high-quality chosen summaries. Building on recent work highlighting the importance of source diversity for effective alignment (Song et al., 2025; Yu et al., 2025), we adopt the FeedSum corpus (Song et al., 2025), which comprises 13,672 documents spanning seven domains with varying lengths and both dialogue and non-dialogue formats. Each document is paired with summaries generated by 13 summarizers, including 3 non-LLMs, 7 open-source LLMs, and 3 proprietary LLMs.

In addition, FEEDSUM provides faithfulness scores for each summary, enabling fine-grained analysis of factual consistency. The dataset also includes a held-out test set of $1,400$ documents, which we later use is as test data to evaluate alignment performance. The training portion serves as the basis for constructing chosen–rejected summary pairs required for our margin-control experiments.

**Chosen Summary Selection.** For further analysis, we construct a set of chosen summaries used to synthesize the corresponding rejected summaries. High-quality chosen summaries are essential, as they provide clear reference points for analyzing the rejected summaries w.r.t preference and factuality margins in Eqs. (1) and (3). To do this, for each document in FEEDSUM, we select the highest-scoring summary with a faithfulness score above 0.8 (out of 1.0) as the chosen one. If no such summary is available, the document is discarded. It leads to 7,491 documents, each paired with a single high-quality reference summary used as the chosen one. Detailed statistics and additional post-processing steps are provided in Appendix C.

## 4.2 THREE FACTORS FOR EFFECTIVE MARGIN CONTROL

We design prompting strategies to modulate each factor independently, while carefully controlling the others to isolate its effect (see the isolation details in Appendix G.1). Given the full document and its chosen reference summary, each prompt generates a rejected summary with controlled variation in the target aspect. Unless noted otherwise, we use Llama3.1-70B-Instruct as the backbone, varying only the prompts, and report results with GPT-4o-mini and Gemma3-12B in Appendix G.7.

### 4.2.1 FACTOR 1: HALLUCINATION LEVEL

The hallucination level correlates with the factuality margin in Eq. (3), as a higher margin reflects greater factual discrepancy. To control this, we vary hallucination in rejected summary generation by explicitly specifying it in the prompt (*e.g.*, About 10% of the summary should be edited with hallucinated content). We consider four settings: None (0%, paraphrasing only), Mild (10%), Severe

(over 90%), and Default (no explicit control), as shown in Tables 31–33. We generate rejected summaries under each prompt setting and evaluate them with GPT-4o-mini using the G-EVAL prompt (Liu et al., 2023) to obtain faithfulness scores (see the prompt in Table 39).

Table 1 presents the distribution of rejected summaries across different faithfulness score intervals, along with their factuality margins, illustrating how hallucination control influences the factual margin of rejected summaries. The results demonstrate that hallucination level effectively controls the factuality margin. Each level yields a clearly distinct distribution, which directly translates into measurable differences in factuality margins. This suggests that controlling the level of hallucination is the key in constructing effective chosen–rejected pairs with higher alignment potential.

Table 1: Faithfulness score distribution across hallucination levels along with factuality margin.

| Score | None | Mild | Severe | Default |
|---|---|---|---|---|
| 7-10 | **96.8%** | 31.4% | 0.0% | 1.4% |
| 4-6 | 2.3% | **51.5%** | 1.1% | 7.3% |
| 1-3 | 0.9% | 17.1% | **98.9%** | **91.3%** |
| $\Delta_{\text{fact}}$ | 0.88 | 4.06 | 7.81 | 6.89 |

Furthermore, the Default setting performs comparably to Severe, suggesting that large factuality margins can emerge without explicit hallucination control, reducing the need for complex prompts.

### 4.2.2 FACTOR 2: SUMMARY LENGTH

Token length is a potential factor influencing the preference margin in Eq. (1), as longer (or shorter) summaries amplify log-probability differences with the chosen summary. To verify the impact of length, we control the token length of rejected summaries by explicitly instructing the model to generate a Shorter, Similar, or Longer version relative to the given chosen summary. As in Tables 34–36, we adapt the Default prompt used in the hallucination control experiment, by adding a single length instruction, keeping the overall prompt simple.

Figure 2 compares the token length gap[2] and preference margin of rejected summaries across three length settings: Shorter, Similar, and Longer. Controlling the length of rejected summaries proves to be an effective way to modulate the preference margin. Fewer tokens in the rejected summaries than in the chosen ones shift the preference margin to the left, while more tokens shift it to the right. This stems from the preference margin being computed over token-level log-likelihoods, as in Eq. (1); thus, longer summaries accumulate greater differences, amplifying

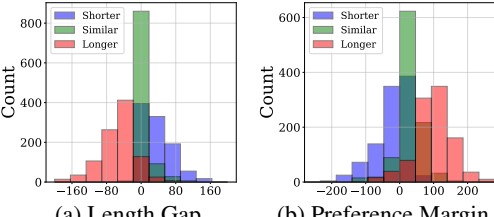

(a) Length Gap.      (b) Preference Margin.

Figure 2: Comparison of token length gaps and preference margins over rejected summaries across three summary length settings.

the model's preference, while shorter ones reduce it. This suggests that matching summary lengths reduces superficial cues and forces models to focus on factual differences.

### 4.2.3 FACTOR 3: PROMPT COMPLEXITY

The final factor is the complexity of the prompt (*i.e.*, number of instructions) for rejected summary generation. While it does not directly impact hallucination level or summary length, higher complexity often produces rejected summaries that differ from the chosen ones in superficial or exaggerated ways, increasing the preference margin, a pattern similarly observed in a recent study (Kim et al., 2025). This encourages models to rely on trivial cues rather than factual differences.

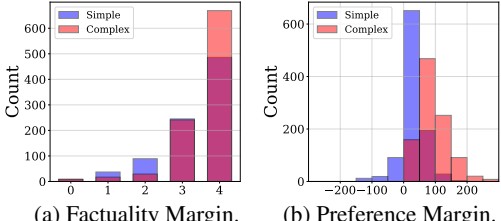

(a) Factuality Margin.      (b) Preference Margin.

Figure 3: Factuality and preference margins over rejected summaries using Simple and Complex prompts with varying instruction complexity.

To examine how prompt complexity affects alignment potential, we compare two prompts for generating high-level hallucinations in Section 4.2.1, both including length control to avoid unintended preference margin from length gaps. However, the prompts differ in instruction complexity: Simple, a length-controlled variant (in Table 35) of the Default prompt, and Complex, a length-controlled variant (in Table 37) of the Severe prompt.

---

[2]The length gap is defined as len($Y_c$) - len($Y_r$), where len($\cdot$) denotes the number of tokens in a given summary.

Figure 3 shows the factuality and preference margin across two prompts with varying prompt complexity. As both prompts are designed to induce high-level hallucinations, they generally produce rejected summaries with large factuality margins. Yet, the preference margin reveals a stark contrast. The `Complex` prompt yields many rejected summaries with preference margins far to the right of zero, indicating they are trivially dispreferred, whereas the `Simple` prompt produces margins closer to zero on average. That is, the complex prompt inflates hallucination levels by injecting superficial differences, making the rejected ones easier for the model to distinguish. Therefore, simpler prompts is key to generate more effective contrastive examples with higher alignment potential.

We further provide an in-depth analysis to understand how the instruction complexity and hallucination types of the `Complex` prompt affect the quality of rejected summaries in Appendix G.10.[Y2QS]

### 4.3 Connection to Enhancing Summary Faithfulness

While our analysis shows how factors affect the two margins and guide prompt design, it remains unclear if they enhance faithfulness in alignment tuning. We therefore test whether their improved alignment potential yields measurable gains. Table 2 reports summary faithfulness, measured by FineSurE, G-Eval, and AlignScore, after DPO training using Llama3.2-3B-Inst on rejected summaries generated under eight prompt designs. Rows 3–9 correspond to the three investigated factors.

The results show that alignment potential increases with larger factuality margins and smaller preference margins, which in turn lead directly to improved summary faithfulness. First, the `Severe` hallucination setting yields higher faithfulness scores than `Mild` and `None`, as its rejections exhibit larger factuality margins. Second, the `Simple(=Similar)` prompt design, which combines matched-length and concise instructions, consistently improves faithfulness by reducing preference margins compared to `Longer`, `Shorter`, and `Complex`. In particular, we observe that this trend remains consistent across different LLMs used to generate rejected summaries, including GPT-4o-mini and Gemma3-12B-Instruct, in addition to Llama3.1-70B-Instruct, as detailed in Appendix G.7.

Table 2: Comparison of rejected summaries w.r.t summary faithfulness.

| Factor | FineSurE | G-Eval | AlignScore |
|--------|----------|--------|------------|
| Default | 0.857 | 8.743 | 0.823 |
| Severe | 0.856 | 8.688 | 0.812 |
| Mild | 0.828 | 8.265 | 0.786 |
| None | 0.795 | 7.537 | 0.748 |
| Longer | 0.854 | 8.645 | 0.819 |
| Shorter | 0.859 | 8.712 | 0.821 |
| Complex | 0.850 | 8.682 | 0.810 |
| Simple | 0.885 | 9.037 | 0.854 |

## 5 Proposed Prompt: SPICE

We introduce SPICE, a prompting strategy grounded in three insights for generating more effective rejected summaries that improve faithfulness alignment, as detailed:

**Insight 1.** Rejected summaries should *contain a high level of hallucination*, ensuring a large factuality margin and making the factual contrast with the chosen summaries more effective.

**Insight 2.** Rejected summaries should *be similar in length* to the chosen summaries to keep the preference margin small, ensuring the model's preference is based on factuality difference, not length.

**Insight 3.** Prompts should *be kept sufficiently simple* to avoid generating summaries that are trivially dispreferred, thereby maintaining a small preference margin and informative difference.

Table 3: Final prompt used by SPICE to generate rejected summaries for alignment training.

```
You are given a document and a reference summary.
Your task is to generate a factually inconsistent summary based on the provided
document and reference summary.
Ensure that the generated summary has the same length as the reference summary.

Document: {Document}
Reference Summary: {Chosen Summary}

Your answer MUST be in JSON format.
The dictionary key should be "hallucinated_summary" as a string.
```

By these principles, our prompt in Table 3 is simple, consisting of the base prompt with only a single additional instruction for length control (highlighted in purple). We omit explicit hallucination

control to keep the prompt concise, as the LLM naturally can generate high-level hallucinations without additional guidance, as demonstrated in Section 4.2.1. This prompt is exactly the same as the `Simple` prompt in Section 4.2.3, and as shown in Figure 3 (see the blue distribution), it yields a large factuality margin while keeping the preference margin small. Therefore, SPICE uses this final prompt to generate rejected summaries for the 7,491 training examples secured in Section 4.1.

# 6 EVALUATION

To verify the effectiveness of SPICE, we compare the summarization quality after alignment tuning using rejected summaries generated by our method versus those produced by existing baselines.

**Baseline Method.** We compare SPICE with two reference-based and one model-based methods. (i) CLIFF (Cao & Wang, 2021) injects factual errors using two variants: Entity, which replaces named entities with same-type alternatives using a rule-based approach, and Mask, which uses a masked language model to generate plausible but incorrect content. (ii) SYNFAC-EDIT (Mishra et al., 2024) uses an LLM to generate factually inconsistent summaries. Originally designed for the clinical domain, we adapt its prompt for general summarization while preserving the core structure (Table 44). (iii) SummLlama (Song et al., 2025) selects rejected summaries from 13 summarizers based on faithfulness scores computed by FineSurE (Song et al., 2024). Details is in Appendix D.

We further include two baselines: a zero-shot model (Zero-Shot) without alignment tuning; and the model trained with supervised fine-tuning (SFT) using only the chosen summaries. For all reference-based methods, including ours, rejected summaries are generated using Llama3.1-70B-Inst for consistency. Results with GPT-4o-mini and Gemma3-12B are discussed in Section G.7.

**Configuration.** We train three instruct-tuned LLM backbones with varying size, including Llama3.2-1B/3B and Llama3.1-8B, with DPO using 7,491 chosen–rejected pairs across all methods. We fine-tune the models with QLoRA (Dettmers et al., 2024) for 6,000 steps using AdamW, with an initial learning rate 1e-6 (cosine decayed) and a weight decay 0.05. See Appendix E for additional training details. A detailed comparison between data-level length control (equal-length vs. no-control) and loss-level normalization methods such as SimPO and KTO is provided in Appendix G.8.[NXZW]

**Metric.** We evaluate the effectiveness of rejected summaries by measuring faithfulness improvement on summaries generated for 1,400 test documents in FeedSum (Song et al., 2025). Faithfulness is evaluated using three recent automatic factuality metrics, including FineSurE (Song et al., 2024), G-Eval (Liu et al., 2023), and AlignScore (Zha et al., 2023), with human evaluation (Human).[3] Note that G-Eval produces scores on a 1–10 Likert scale, whereas the other metrics report scores as the proportion of faithful content on a 0–1 scale. The metric details are presented in Appendix F, including the evaluation protocol in Appendix F.3 and human evaluation scale in Appendix F.4.1[NXZW].

## 6.1 MAIN EXPERIMENT

### 6.1.1 SUMMARY FAITHFULNESS IMPROVEMENT

Table 4 presents the faithfulness scores of summaries generated by 7 different methods across two backbone models. Our proposed method, SPICE, consistently achieves the highest factuality scores across all model sizes and metrics, including both human and automated evaluations. Specifically in human evaluation, SPICE achieves the highest factuality scores of 0.833 (3B) and 0.909 (8B), clearly surpassing other methods by significant margins. Notably, the smaller 3B model trained with SPICE achieves faithfulness scores exceeding those of the larger 8B zero-shot baseline; it achieves higher scores in Human (0.833 vs. 0.824), FineSurE (0.885 vs. 0.864), G-Eval (9.037 vs. 8.317) and AlignScore (0.854 vs. 0.796), highlighting its exceptional efficiency and alignment effectiveness.

Additionally, we compute each method's overall faithfulness score by averaging eight instances (four metrics and two models), yielding the following averages: Zero-Shot (0.794), SFT (0.813), CLIFF-Entity (0.810), CLIFF-Mask (0.813), SYNFAC-EDIT (0.822), SummLlama (0.854), and SPICE (0.895). SPICE achieves the highest score, outperforming SummLlama by +4.1%p, significantly surpassing the most recent and competitive baseline. All results in the main tables are statistically

---

[3]Human evaluation was done for 1,400 document-summary pairs via Amazon MTurk, with three qualified annotators per pair, achieving an inter-annotator agreement of 0.53 (Krippendorff's $\alpha$). This agreement level is commonly regarded as high in text summarization due to the inherent subjectivity and complexity of the evaluation. Refer to Appendix F.4.2 for details.[jYGQ].

Table 4: Faithfulness of summaries generated by two LLM backbones using 6 different training methods, along with zero-shot inference. Best scores are highlighted in **bold**.

| Method | Backbone: Llama3.2-3B-Instruct | | | | Backbone: Llama3.1-8B-Instruct | | | |
|---|---|---|---|---|---|---|---|---|
| | Human | FineSurE | G-Eval | AlignScore | Human | FineSurE | G-Eval | AlignScore |
| Zero-Shot | $0.736_{(0.00)}$ | $0.798_{(0.00)}$ | $7.519_{(0.00)}$ | $0.751_{(0.00)}$ | $0.824_{(0.00)}$ | $0.864_{(0.00)}$ | $8.317_{(0.00)}$ | $0.796_{(0.00)}$ |
| SFT | $0.693_{(0.00)}$ | $0.823_{(0.00)}$ | $7.947_{(0.00)}$ | $0.792_{(0.00)}$ | $0.835_{(0.00)}$ | $0.867_{(0.00)}$ | $8.789_{(0.00)}$ | $0.818_{(0.00)}$ |
| CLIFF-Entity | $0.743_{(0.00)}$ | $0.816_{(0.00)}$ | $8.011_{(0.00)}$ | $0.785_{(0.00)}$ | $0.787_{(0.00)}$ | $0.854_{(0.00)}$ | $8.742_{(0.00)}$ | $0.820_{(0.00)}$ |
| CLIFF-Mask | $0.730_{(0.00)}$ | $0.828_{(0.00)}$ | $7.981_{(0.00)}$ | $0.794_{(0.00)}$ | $0.789_{(0.00)}$ | $0.861_{(0.00)}$ | $8.773_{(0.00)}$ | $0.826_{(0.00)}$ |
| SYNFAC-EDIT | $0.765_{(0.00)}$ | $0.812_{(0.00)}$ | $8.264_{(0.00)}$ | $0.789_{(0.00)}$ | $0.816_{(0.00)}$ | $0.855_{(0.00)}$ | $8.797_{(0.00)}$ | $0.834_{(0.00)}$ |
| SummLlama | $0.792_{(0.01)}$ | $0.843_{(0.01)}$ | $8.473_{(0.00)}$ | $0.813_{(0.01)}$ | $0.879_{(0.02)}$ | $0.908_{(0.02)}$ | $9.038_{(0.01)}$ | $0.848_{(0.01)}$ |
| **SPICE (Ours)** | **0.833** | **0.885** | **9.037** | **0.854** | **0.909** | **0.934** | **9.438** | **0.895** |

significant ($p < 0.05$) and further details are provided in the Appendix F.1. These results highlight that our simple prompting, by effectively controlling factuality and preference margins, enables strong faithfulness alignment in text summarization.

### 6.1.2 FACTUALITY AND PREFERENCE MARGIN

Table 5 shows the average factuality margin ($\Delta_{fact}$) and preference margin ($\Delta_{pref}$), along with their alignment potential ($|\Delta_{fact} - \Delta_{pref}|$), computed from the training datasets used by each rejection methods. Our proposed method, SPICE, achieves the highest alignment potential among all approaches, reaching a value of 74.3 on the generated rejected summaries. This value is significantly greater than those by other approaches, indicating that SPICE produces far more informative and non-trivial rejected summaries for faithfulness alignment. Particularly, although SPICE does not yield the smallest

Table 5: Average alignment potential of rejected summaries over five methods. Llama3.1-70B-Inst (external evaluator) and Llama3.2-3B-Inst (policy model) are used to compute $\Delta_{fact}$ and $\Delta_{pref}$.

| Method | $\Delta_{fact}$ | $\Delta_{pref}$ | $|\Delta_{fact} - \Delta_{pref}|$ |
|---|---|---|---|
| CLIFF-Entity | 47.6 | 28.5 | 19.1 |
| CLIFF-Mask | 69.6 | 41.9 | 27.7 |
| SYNFAC-EDIT | 45.2 | 16.3 | 28.9 |
| SummLlama | 56.7 | 16.5 | 40.2 |
| SPICE (Ours) | 102.0 | 28.7 | **74.3** |

preference margin, it uniquely combines a substantially larger factuality margin with a moderate preference margin, resulting in the largest margin gap. This demonstrates that **maximizing the interaction between the two margins, rather than pushing each margin to its extreme in isolation, is the key to generating high-value rejections that most effectively drive faithful summarization**, consistent with the theoretical insights established in Section 3.2.[v7Lg, jYGQ]

On the other hand, the baseline methods show clear limitations. CLIFF-Mask increases the factuality margin but also inflates the preference margin, making rejected summaries overly distinguishable. SummLlama attains a small preference margin (16.5) but only a modest factuality margin, yielding a much smaller alignment potential of 40.2. CLIFF-Entity performs worst, with both low factuality and relatively large preference margins. Lastly, although SYNFAC-EDIT is reference-based like SPICE, it fails to generate sufficiently hallucinated rejections and thus exhibits the smallest factuality margin (45.2), reflecting limited control over factual contrast. Detailed additional analyses of these baselines and alignment potential are provided in Appendices D and G.13.[v7Lg, jYGQ]

## 6.2 ADDITIONAL EXPERIMENT

### 6.2.1 DATA EFFICIENCY BY SPICE.

To show the data efficiency of SPICE, we compare the effectiveness of chosen–rejected pairs generated by SPICE with those produced by the second-best performing method, SummLlama, as reported in Table 4. Table 6 reports the faithfulness scores obtained when the number of chosen–rejected examples used for training is progressively reduced. It is noteworthy that **SPICE's generated examples enable the model to achieve performance comparable to SummLlama using only 1/16 of the training examples** (see the last row), and continues to improve as the number of

Table 6: Effectiveness of SPICE's rejected summaries used for DPO on Llama3.2-3B-Instruct. We sample the full training set of 7,391 examples at four scales: 3,745 (1/2), 1,872 (1/4), 936 (1/8), and 468 (1/16), respectively.

| Method | FineSurE | G-Eval | AlignScore |
|---|---|---|---|
| SummLlama | 0.843 | 8.473 | 0.813 |
| SPICE | 0.885 | 9.037 | 0.854 |
| SPICE$^{1/2}$ | 0.874 | 8.814 | 0.846 |
| SPICE$^{1/4}$ | 0.864 | 8.736 | 0.833 |
| SPICE$^{1/8}$ | 0.858 | 8.689 | 0.825 |
| SPICE$^{1/16}$ | 0.846 | 8.503 | 0.812 |

training samples increases (see the 2nd-5th rows). These results indicate that SPICE achieves robust faithfulness alignment even with less training data, demonstrating its data efficiency and effectiveness in low-resource settings. In terms of data generation latency, our method achieves a $2.1\times$ speed-up over SummLlama to obtain an equivalent number of preference pairs, as detailed in Appendix G.9.

### 6.2.2 ANALYSIS ON ALIGNMENT TAX.

Faithfulness alone does not define summary quality. It is essential to examine how alignment tuning for faithfulness affects other evaluation dimensions, since such improvements may come with an alignment tax (Song et al., 2025) that diminishes performance in those areas. Table 7 presents the performance differences across four quality dimensions, commonly adopted in recent research, before and after alignment tuning with SPICE. The three quality dimensions, including faithfulness, coherence, and relevance, are measured using G-Eval (Liu et al., 2023) (see prompts in Tables 39–41), while abstractiveness is assessed using the novel $n$-gram method. Detailed descriptions and formulations are provided in Appendix F.

Table 7: Multi-dimensional summary quality on Llama3.2-3B-Instruct before and after training with SPICE across four evaluation dimensions: faithfulness (Faith.), coherence (Coh.), and relevance (Rel.), and abstractiveness (Abs.).

| Method | Faith ($\uparrow$) | Coh. ($\uparrow$) | Rel. ($\uparrow$) | Abs. ($\uparrow$) |
|---|---|---|---|---|
| Zero-Shot | 7.519 | 8.102 | 8.563 | 0.686 |
| DPO-SPICE | 9.037 | 8.059 | 8.579 | 0.689 |

The results show that **while SPICE substantially improves faithfulness, the trained model does not incur an alignment tax**, maintaining comparable or slightly better relevance (8.579 vs. 8.563) and abstractiveness (0.689 vs. 0.686), with only a negligible change in coherency (8.059 vs. 8.102). This suggests that SPICE generates rejected summaries with high alignment potential, providing strong factual alignment signals while preserving quality across other evaluation dimensions.

### 6.2.3 COMPARISON WITH ADVERSARIAL SAMPLING

We further analyze whether adversarial sampling (*i.e.*, top-$k$ selection of rejected summaries with the largest alignment potential from a large candidate pool) can replace our simple SPICE prompt that directly generates high-alignment-potential rejections. For comparison, we adversarially sample rejected summaries from the full FeedSum seed set. For each of the 7,491 unique documents (each accompanied by 13 rejected-summary candidates), we select the top-1 candidate with the largest

Table 8: Factuality and preference margins on two training datasets. Llama3.1-70B-Inst (evaluator) and Llama3.2-3B-Inst (policy) are used to compute $\Delta_{\text{fact}}$ and $\Delta_{\text{pref}}$.

| Method | $\Delta_{\text{fact}}$ | $\Delta_{\text{pref}}$ | $|\Delta_{\text{fact}}{-}\Delta_{\text{pref}}|$ |
|---|---|---|---|
| Adv. Sampling | 58.5 | 13.4 | 45.1 |
| SPICE (Ours) | 102.0 | 28.7 | 74.3 |

alignment potential, ensuring that the resulting dataset matches the size of SPICE for a fair comparison. Table 8 summarizes the factuality and preference margins with alignment potential. Notably, even after selecting the best candidate among 13 candidates, the alignment potential remains substantially lower than that of SPICE. This suggests that **large pools generated by unoptimized prompts, like SummLlama, struggle to produce rejected summaries with sufficiently high alignment potential**, making it difficult to obtain effective rejections through quantity alone.[NXZW]

Furthermore, adversarial sampling is highly inefficient, requiring over $200\times$ more candidates to match SPICE's performance; see Appendix G.12 for the data-efficiency comparison.[NXZW]

## 7 CONCLUSION

In this paper, we provided theoretical and empirical analyses of what made rejected summaries effective for faithfulness alignment in summarization. We showed that high hallucination, matched length, and simple prompts were key to balancing preference and factuality margins, thereby shaping alignment potential. Building on these insights, we designed SPICE, which produced rejections with higher training value and substantially improved factuality, while maintaining coherence, relevance, and abstractiveness. Additionally, we showed that SPICE achieves the highest alignment potential among all methods, offers strong data efficiency by matching competitive baselines with only a small fraction of the data, and surpasses adversarial sampling despite requiring neither large candidate pools nor complex selection procedures.[NXZW] Together, these findings provide practical guidance for constructing preference data that enables the development of more faithful summarizers.

ETHICS STATEMENT

Our work focuses on leveraging LLM-generated feedback for alignment in text summarization, which does not raise ethical concerns during training, as all data was generated and evaluated using publicly available models or APIs. For human evaluation, we followed established protocols from prior work to ensure fairness and quality. Annotators were compensated at a rate exceeding the U.S. minimum wage and received additional bonuses based on consistency and performance.

REPRODUCIBILITY STATEMENT

In Section 4.1 and Appendices C and D, we clearly describe the data, including its selection and processing details. The experimental setup in Section 6 outlines the baseline methods, model configurations, and evaluation metrics. Training details, including hyperparameters, are provided in Appendix E, and details of the evaluation metrics are available in Appendix F. For open-source models, we list their publicly available checkpoints from Hugging Face. For proprietary models, we state that we access them through paid APIs, such as OpenAI. Detailed model and checkpoint information is included in Table 29 in the appendix. Moreover, all prompts used in this work are provided in the Appendix. This comprehensive description ensures the reproducibility of our work.

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

## LIMITATIONS

In our study, we focus primarily on the task of faithfulness alignment within the text summarization domain. However, SPICE has the potential to be applied to a wide range of alignment objectives with simple modification, such as helpfulness and safety, owing to its simplicity and ease of implementation. It can also be adapted to other generation tasks beyond summarization, including machine translation, question answering, and code generation. We leave the validation of its generality and robustness in these broader settings to future work.

# A DISCUSSION

## A.1 DISCUSSION ON PREFERENCE DEFINITION AND LENGTH NORMALIZATION

A natural question arises regarding the role of sequence length in defining model preference.

The distinction stems from the optimization objectives. Since we can assume a uniform reference policy $\pi_{ref}(y|x) = \frac{1}{|\mathcal{Y}|}$ according to Shi et al. (2025), DPO (Rafailov et al., 2023a) defines the loss using the *unnormalized cumulative log-probability* of the chosen and rejected summaries:

$$\mathcal{L}_{\text{DPO}} \propto \log \pi_\theta(y_w|x) - \log \pi_\theta(y_l|x),$$

which directly compares raw sequence likelihoods. Because cumulative log-probabilities naturally decrease with sequence length, this formulation is inherently length-sensitive Liu et al. (2024b). Note that if we set $\pi_{ref} = \pi_0$ in the DPO implementation, then $\frac{\pi_\theta(y|x)}{\pi_{ref}(y|x)} = 1$ prior to training, collapsing all preferences into a trivial ratio. *Hence, the reference term must be omitted in the data metric level.*

In contrast, SimPO (Meng et al., 2024) employs *length-normalized* log-probabilities:

$$\mathcal{L}_{\text{SimPO}} \propto \frac{1}{|y_w|} \log \pi_\theta(y_w|x) - \frac{1}{|y_l|} \log \pi_\theta(y_l|x),$$

explicitly removing length bias by averaging per token. This difference in normalization leads to different optimization dynamics, not just differences in evaluation.

It is worth noting that our analysis primarily focuses on the standard DPO objective, where the loss is defined on the cumulative (unnormalized) log probabilities as in Eq. 1. Accordingly, we did not apply length normalization, and instead explicitly incorporated length as an independent factor (Factor 2) in our framework. However, normalization-based variants such as SimPO apply token-level average log probabilities to remove length bias. In such cases, the effect of length is already normalized away, and the length factor can be ignored.

Interestingly, our results in Section 4.2.2 show that alignment is most effective when the chosen and rejected summaries are of similar length. This implies that a form of length normalization naturally arises at the data-construction level, which makes the DPO-style formulation effectively consistent with length-normalized variants such as SimPO. We view this as a positive signal: despite the differences in objective formulation, both approaches converge to the same practical guideline regarding length, highlighting the robustness of this factor in preference optimization.

**Implications.** These formulations highlight that the appropriate way to compute preference margins should match the underlying objective of the optimization method: unnormalized for DPO, normalized for SimPO. Therefore, length normalization should be viewed as a design choice consistent with the intended learning objective.

## A.2 Factors Influencing the Preference Margin

A natural question arises regarding what factors, beyond factuality, may influence the preference margin in preference optimization.

By definition, the preference margin reflects the model's log-probability of generating a given output (e.g., a summary) in response to a specific input prompt (e.g., "Factually summarize the input document"). Our setup explicitly encourages factuality to be the primary preference dimension. Nonetheless, residual effects from other factors such as output length, writing style, or tone may still influence the margin, even if factuality is the intended focus.

However, to our knowledge, output length is the most significant residual factor influencing the preference margin, and recent studies have focused heavily on this aspect. This is because longer outputs naturally accumulate more log-probability mass, which can inflate margin values regardless of content quality. As a result, output length has been the primary target of both theoretical and empirical analyses in prior work (Meng et al., 2024; Liu et al., 2024b; Lu et al., 2024).

In line with prior studies, we primarily focus on summary length as an explicit factor influencing the preference margin. This is particularly important in summarization, as different models vary in how much content they include to achieve completeness. As highlighted in recent benchmark studies (Song et al., 2024; Lee et al., 2024), the degree of completeness, i.e., how thoroughly a summary covers key information, is often reflected in the summary length, making it a critical factor for fair and meaningful evaluation.

**Implications.** These observations suggest that while factuality is the intended optimization dimension, length remains an unavoidable confounding factor that requires explicit consideration. Controlling for length thus ensures that the preference margin faithfully reflects factuality rather than being dominated by token count.

## A.3 Alignment Potential vs. Hard Negative Mining

Although alignment potential is conceptually related to hard negative mining, it differs from the traditional contrastive learning principle that *"difficult yet informative negatives provide stronger training signals"* in several important ways.

In both supervised and unsupervised contrastive learning, negative labels are determined or assumed in advance by the framework: labels explicitly define negatives in supervised settings, and once an anchor–positive pair is selected in unsupervised settings, all remaining samples are implicitly treated as negatives. Hard negative mining then evaluates difficulty of discrimination solely through embedding distance, measuring only how confusing a negative is for the model. Importantly, hard negative mining does not consider whether the negative label itself is correct; informativeness is assumed purely from difficulty. In other words, hard negatives in contrastive learning are informative only under the assumption that the example truly belongs to the negative class.

In contrast, alignment potential evaluates two signals simultaneously: the factuality margin, which captures how clearly incorrect the rejected is (informativeness), and the preference margin, which reflects how non-trivial it is for the model to distinguish it from the chosen (difficulty of discrimination). A rejected example is informative only when these two signals interact appropriately. For example, a hard negative (very small preference margin) with a very small factuality margin may remain uninformative because it lacks meaningful errors.

**Implications.** The key difference is that while contrastive learning focuses exclusively on the difficulty of discrimination, alignment potential quantitatively considers both the informative and the difficulty of a discrimination simultaneously.[v7Lg]

## B  CONVERGENCE OF FIXED–WINNER DPO WITH SAMPLED LOSER

**Proof of Section 3.2.**

**Contextual bandit setting.**    We frame the problem in a contextual bandit setting like previous works in RLHF (Rafailov et al., 2023b; Rosset et al., 2024; Huang et al., 2025) with a context space $\mathcal{X}$ and a full action (response) space $\mathcal{Y}$. For each $x \in \mathcal{X}$, a policy $\pi_\theta(\cdot \mid x)$ is a probability distribution on $\mathcal{Y}$, and all evaluation metrics are defined on this space.

**Setting and population loss.**    We use DPO population loss to optimize policy $\pi_\theta$:

$$
\mathbb{E}_{(x,y_w,y_l)\sim s}\big[\mathcal{L}(x, y_w, y_l; \theta)\big] = \mathbb{E}_{(x,y_w,y_l)\sim s}\Big[ - p(y_w \succ y_l) \log \sigma\big(r_\theta(y_w; x) - r_\theta(y_l; x)\big)\Big]
$$

$$
= \mathbb{E}_{(x,y_w,y_l)\sim s}\Big[ - \sigma\big(f(y_w; x) - f(y_l; x)\big) \log \sigma\big(r_\theta(y_w; x) - r_\theta(y_l; x)\big)\Big].
$$

Under stochastic gradient descent, at iteration $t$ we draw a triple $(x^t, y_w^t, y_l^t)$ and update $\theta^t$ using the empirical loss. Because the sampled context and pair allow us to evaluate both $\mathcal{L}(x^t, y_w^t, y_l^t; \theta^t)$ and its swapped counterpart $\mathcal{L}(x^t, y_l^t, y_w^t; \theta^t)$, we follow prior work (Azar et al., 2024; Rosset et al., 2024; Huang et al., 2025) and adopt the symmetric update

$$
\theta^{t+1} = \theta^t - \tfrac{1}{2}\, \eta^t \, \nabla_\theta\Big(\mathcal{L}(x^t, y_w^t, y_l^t; \theta^t) + \mathcal{L}(x^t, y_l^t, y_w^t; \theta^t)\Big),
$$

where $\eta^t$ is the step size.

Following (Shi et al., 2025), we take the reference policy to be uniform, $\pi_{\mathrm{ref}}(y \mid x) = 1/|\mathcal{Y}|$. Under this choice, the model preference satisfies $\hat{r}_\theta(x, y) = \beta\, \theta(x, y) + \log Z(x)$, and the ground truth model $\theta^\star$ characterized by $f(x, y) = \hat{r}_{\theta^\star}(x, y) + \log Z^\star(x)$ implies $\beta\, \theta^\star(x, y) = f(x, y) + c(x)$ where $c(x)$ is constant.

Therefore, we can define the potential (misalignment) of summary and its variance:

$$
\gamma_y^t := \beta\theta^t(x, y) - f(x, y), \qquad \bar{\gamma}^t := \frac{1}{|\mathcal{Y}|} \sum_{y\in\mathcal{Y}} \gamma_y^t, \qquad V^t := \frac{1}{|\mathcal{Y}|} \sum_{y\in\mathcal{Y}} (\gamma_y^t - \bar{\gamma}^t)^2.
$$

Then MSME can be written *explicitly* in terms of the $\gamma$'s as

$$
\mathrm{MSME}(\theta^t \mid y_w A) = \frac{1}{|\mathcal{Y}|} \sum_{y\in\mathcal{Y}} (\gamma_{y_w}^t - \gamma_y^t)^2 \quad \text{where } \mathrm{SME} = (\gamma_{y_w}^t - \gamma_y^t)^2 = (\Delta_{\mathrm{fact}} - \Delta_{\mathrm{pref}})^2.
$$

$$
= \frac{1}{|\mathcal{Y}|} \sum_{y\in\mathcal{Y}} \left( (\gamma_{y_w}^t - \bar{\gamma}^t) - (\gamma_y^t - \bar{\gamma}^t) \right)^2
$$

$$
= \frac{1}{|\mathcal{Y}|} \sum_{y\in\mathcal{Y}} \left[ (\gamma_{y_w}^t - \bar{\gamma}^t)^2 + (\gamma_y^t - \bar{\gamma}^t)^2 - 2(\gamma_{y_w}^t - \bar{\gamma}^t)(\gamma_y^t - \bar{\gamma}^t) \right]
$$

$$
= (\gamma_{y_w}^t - \bar{\gamma}^t)^2 + \frac{1}{|\mathcal{Y}|} \sum_{y\in\mathcal{Y}} (\gamma_y^t - \bar{\gamma}^t)^2 - \frac{2}{|\mathcal{Y}|} (\gamma_{y_w}^t - \bar{\gamma}^t) \underbrace{\sum_{y\in\mathcal{Y}} (\gamma_y^t - \bar{\gamma}^t)}_{=0}
$$

$$
= (\gamma_{y_w}^t - \bar{\gamma}^t)^2 + V^t. \tag{8}
$$

Thus controlling MSME is equivalent to controlling $V^t$ up to the (winner–mean) offset $(\gamma_{y_w}^t - \bar{\gamma}^t)^2$.

**Per-step decrease.**    With the optimal learning rate by step $t$, the variance decreases in expectation as

$$
\mathbb{E}_{y\sim s}[V^{t+1} - V^t] = -\frac{1}{2|\mathcal{Y}|} \mathbb{E}_{y\sim s}[(\gamma_{y_w}^t - \gamma_y^t)^2], \tag{9}
$$

as shown in previous work (Huang et al., 2025).

**(1) Uniform loser sampling.** If $s$ is uniform over $\mathcal{Y}$, then

$$\mathbb{E}_{y \sim s}\big[(\gamma_{y_w}^t - \gamma_y^t)^2\big] = \frac{1}{|\mathcal{Y}|} \sum_{y \in \mathcal{Y}} (\gamma_{y_w}^t - \gamma_y^t)^2 = (\gamma_{y_w}^t - \bar{\gamma}^t)^2 + V^t = \mathrm{MSME}(\theta^t \mid y_w).$$

Plugging equation 10 into equation 9 yields, for the uniform sampler,

$$\mathbb{E}[V^{t+1} - V^t] = -\frac{1}{2|\mathcal{Y}|}\, \mathbb{E}_{y \sim \mathrm{Unif}(\mathcal{Y})}\big[(\gamma_{y_w}^t - \gamma_y^t)^2\big] = -\frac{1}{2|\mathcal{Y}|}\, \mathrm{MSME}(\theta^t \mid y_w) \tag{10}$$

$$\mathbb{E}[V^{t+1}] = V^t - \frac{1}{2|\mathcal{Y}|}\, \mathrm{MSME}(\theta^t \mid y_w), \tag{11}$$

Under the stepsize choice used to derive equation 9, the per-pair update satisfies $\delta^t := \gamma_{y_w}^{t+1} - \gamma_{y_w}^t = -\frac{1}{2}(\gamma_{y_w}^t - \gamma_y^t)$, so for the uniform sampler

$$\mathbb{E}\big[(\gamma_{y_w}^{t+1} - \bar{\gamma}^t)^2\big] = \mathbb{E}\big[(\gamma_{y_w}^t - \bar{\gamma}^t + \delta^t)^2\big] \tag{12}$$

$$= (\gamma_{y_w}^t - \bar{\gamma}^t)^2 \; + \; 2(\gamma_{y_w}^t - \bar{\gamma}^t)\, \mathbb{E}[\delta^t] \; + \; \mathbb{E}[(\delta^t)^2] \tag{13}$$

$$= (\gamma_{y_w}^t - \bar{\gamma}^t)^2 \; - \; (\gamma_{y_w}^t - \bar{\gamma}^t)^2 \; + \; \frac{1}{4}\mathbb{E}\big[(\gamma_{y_w}^t - \gamma_y^t)^2\big] \tag{14}$$

$$= \frac{1}{4}\Big((\gamma_{y_w}^t - \bar{\gamma}^t)^2 + V^t\Big) = \frac{1}{4}\, \mathrm{MSME}(\theta^t \mid y_w), \tag{15}$$

where we used $\mathbb{E}[\delta^t] = -\frac{1}{2}(\gamma_{y_w}^t - \bar{\gamma}^t)$ and $\mathbb{E}[(\delta^t)^2] = \frac{1}{4}\mathbb{E}[(\gamma_{y_w}^t - \gamma_y^t)^2] = \frac{1}{4}\big((\gamma_{y_w}^t - \bar{\gamma}^t)^2 + V^t\big)$.

Combining equation 11 and equation 15, the next-step MSME obeys the exact recursion

$$\mathbb{E}\big[\mathrm{MSME}(\theta^{t+1} \mid y_w)\big] = \mathbb{E}\big[(\gamma_{y_w}^{t+1} - \bar{\gamma}^t)^2\big] + \mathbb{E}[V^{t+1}] \tag{16}$$

$$= \frac{1}{4}\, \mathrm{MSME}(\theta^t \mid y_w) + V^t - \frac{1}{2|\mathcal{Y}|}\, \mathrm{MSME}(\theta^t \mid y_w) \tag{17}$$

$$= \Big(\frac{1}{4} - \frac{1}{2|\mathcal{Y}|}\Big)\mathrm{MSME}(\theta^t \mid y_w) \; + \; V^t. \tag{18}$$

**(2) Adversarial loser sampling.** Let adversarial sampler $y \sim q_\tau$ with

$$q_\tau(y) = \frac{\big|\gamma_{y_w}^t - \gamma_y^t\big|^\tau}{Z_\tau}, \qquad Z_\tau = \sum_{y \in \mathcal{Y}} \big|\gamma_{y_w}^t - \gamma_y^t\big|^\tau, \qquad \tau \geq 0.$$

Here, the parameter $\tau$ controls the sharpness of the sampling distribution. When $\tau = 0$, the sampler reduces to uniform sampling, while larger $\tau$ values place higher probability mass on losers whose margin discrepancy $|\gamma_{y_w}^t - \gamma_y^t| = |\Delta_{\mathrm{fact}} - \Delta_{\mathrm{pref}}|$ are larger, corresponding to more adversarial selections. In this part, we show that selecting rejected responses $y_l$ in an adversarial manner that is, sampling those with larger absolute margins $|\gamma_{y_w}^t - \gamma_y^t|$ leads to a greater per-step decrease in MSME compared to uniform sampling, and thus accelerates convergence. In other words, given a fixed winner $y_w$, we analyze why choosing the loser $y_l$ adversarially provides a clear advantage over uniform selection.

Define

$$\mu_\tau^t := \mathbb{E}_{y \sim q_\tau}[\gamma_y^t], \qquad \mathrm{Var}_\tau^t := \mathbb{E}_{y \sim q_\tau}\big[(\gamma_y^t - \mu_\tau^t)^2\big].$$

Then

$$G_\tau^t = \mathbb{E}_{y \sim q_\tau}\Big[\big((\gamma_{y_w}^t - \mu_\tau^t) - (\gamma_y^t - \mu_\tau^t)\big)^2\Big]$$

$$= \mathbb{E}_{y \sim q_\tau}\big[(\gamma_{y_w}^t - \mu_\tau^t)^2\big] - 2(\gamma_{y_w}^t - \mu_\tau^t)\, \mathbb{E}_{y \sim q_\tau}\big[\gamma_y^t - \mu_\tau^t\big] + \mathbb{E}_{y \sim q_\tau}\big[(\gamma_y^t - \mu_\tau^t)^2\big]$$

$$= (\gamma_{y_w}^t - \mu_\tau^t)^2 + \mathrm{Var}_\tau^t.$$

Then the winner term expands as

$$\mathbb{E}_{q_\tau}\big[(\gamma_{y_w}^{t+1} - \bar{\gamma}^t)^2\big] = \mathbb{E}_{q_\tau}\big[(\gamma_{y_w}^t - \bar{\gamma}^t + \delta^t)^2\big]$$

$$= (\gamma_{y_w}^t - \bar{\gamma}^t)^2 + 2(\gamma_{y_w}^t - \bar{\gamma}^t)\, \mathbb{E}_{q_\tau}[\delta^t] + \mathbb{E}_{q_\tau}\big[(\delta^t)^2\big]$$

$$= (\gamma_{y_w}^t - \bar{\gamma}^t)^2 - (\gamma_{y_w}^t - \bar{\gamma}^t)(\gamma_{y_w}^t - \mu_\tau^t) + \tfrac{1}{4}\, G_\tau^t,$$

and the variance potential updates as

$$\mathbb{E}_{q_\tau}[V^{t+1}] \;=\; V^t - \frac{1}{2|\mathcal{Y}|}\, G_\tau^t.$$

Therefore we obtain the *exact* recursion

$$\mathbb{E}_{q_\tau}\big[\mathrm{MSME}(\theta^{t+1}\mid y_w)\big] \;=\; \mathrm{MSME}(\theta^t\mid y_w) \;-\; (\gamma_{y_w}^t - \bar{\gamma}^t)(\gamma_{y_w}^t - \mu_\tau^t) \;+\; \left(\tfrac{1}{4} - \tfrac{1}{2|\mathcal{Y}|}\right) G_\tau^t. \quad (19)$$

**(2.1) Adversarial is always faster than uniform.**   From the exact adversarial recursion we have equation 19 and for the uniform sampler ($\tau = 0$),

$$\mathbb{E}_{\mathrm{uni}}\big[\mathrm{MSME}(\theta^{t+1}\mid y_w)\big] = \mathrm{MSME}(\theta^t\mid y_w) - (\gamma_{y_w}^t - \bar{\gamma}^t)^2 + \left(\tfrac{1}{4} - \tfrac{1}{2|\mathcal{Y}|}\right) G_0^t, \qquad (20)$$

$$G_0^t = (\gamma_{y_w}^t - \bar{\gamma}^t)^2 + V^t. \qquad (21)$$

Subtracting,

$$\Delta_\tau^t := \mathbb{E}_{\mathrm{uni}}\big[\mathrm{MSME}(\theta^{t+1}\mid y_w)\big] - \mathbb{E}_{q_\tau}\big[\mathrm{MSME}(\theta^{t+1}\mid y_w)\big]$$
$$= -(\gamma_{y_w}^t - \bar{\gamma}^t)\big(\mu_\tau^t - \bar{\gamma}^t\big) + \alpha\,(G_0^t - G_\tau^t), \qquad \alpha := \tfrac{1}{4} - \tfrac{1}{2|\mathcal{Y}|} \in (0, \tfrac{1}{4}]. \qquad (22)$$

We will show $\Delta_\tau^t \geq 0$ for all $\tau \geq 0$.

**(2.1.1) Base case.** ($\tau = 0$). Since $\mu_0^t = \bar{\gamma}^t$ and $G_0^t = G_0^t$, clearly $\Delta_0^t = 0$.

**(2.1.2) Recursive relation.** ($\tau \to \tau + 1$). By definition,

$$q_{\tau+1}(y) = \frac{|\gamma_{y_w}^t - \gamma_y^t|\, q_\tau(y)}{\mathbb{E}_{q_\tau}[|\gamma_{y_w}^t - \gamma_y^t|]},$$

so for any function $f$,

$$\mathbb{E}_{q_{\tau+1}}[f(y)] = \frac{\mathbb{E}_{q_\tau}[f(y)\,|\gamma_{y_w}^t - \gamma_y^t|]}{\mathbb{E}_{q_\tau}[|\gamma_{y_w}^t - \gamma_y^t|]}.$$

Therefore

$$\mu_{\tau+1}^t - \mu_\tau^t = \frac{\mathrm{Cov}_{q_\tau}\big(\gamma_y^t, |\gamma_{y_w}^t - \gamma_y^t|\big)}{\mathbb{E}_{q_\tau}[|\gamma_{y_w}^t - \gamma_y^t|]}, \qquad G_{\tau+1}^t - G_\tau^t = \frac{\mathrm{Cov}_{q_\tau}\big((\gamma_{y_w}^t - \gamma_y^t)^2, |\gamma_{y_w}^t - \gamma_y^t|\big)}{\mathbb{E}_{q_\tau}[|\gamma_{y_w}^t - \gamma_y^t|]}.$$

Thus from equation 22,

$$\Delta_{\tau+1}^t - \Delta_\tau^t = -(\gamma_{y_w}^t - \bar{\gamma}^t)(\mu_{\tau+1}^t - \mu_\tau^t) + \alpha\,(G_\tau^t - G_{\tau+1}^t)$$
$$= \frac{1}{\mathbb{E}_{q_\tau}[|\gamma_{y_w}^t - \gamma_y^t|]}\Big( -(\gamma_{y_w}^t - \bar{\gamma}^t)\,\mathrm{Cov}_{q_\tau}\big(\gamma_y^t, |\gamma_{y_w}^t - \gamma_y^t|\big) - \alpha\,\mathrm{Cov}_{q_\tau}\big((\gamma_{y_w}^t - \gamma_y^t)^2, |\gamma_{y_w}^t - \gamma_y^t|\big)\Big).$$
$$(23)$$

We split the loser sampling subspace $\mathcal{Y}$ into two disjoint subsets:

$$\mathcal{Y}_1 = \{\, y \in \mathcal{Y} : \gamma_{y_w}^t < \gamma_y^t \,\}, \quad \mathcal{Y}_2 = \{\, y \in \mathcal{Y} : \gamma_{y_w}^t \geq \gamma_y^t \,\}.$$

Hence, the entire subspace can be written as

$$\mathcal{Y} = \mathcal{Y}_1 \cup \mathcal{Y}_2, \qquad \mathcal{Y}_1 \cap \mathcal{Y}_2 = \emptyset.$$

By construction, $y_w$ is the minimizer of $\{\gamma_y^t : y \in \mathcal{Y}_1\}$, and $y_w$ is the maximizer of $\{\gamma_y^t : y \in \mathcal{Y}_2\}$.

In case of $\mathcal{Y}_1$, $|\gamma_{y_w}^t - \gamma_y^t| = -\gamma_{y_w}^t + \gamma_y^t$ and hence $\gamma_y^t = \gamma_{y_w}^t + |\gamma_{y_w}^t - \gamma_y^t|$.

Thus

$$\mathrm{Cov}_{q_\tau}(\gamma_y^t, |\gamma_{y_w}^t - \gamma_y^t|) = \mathrm{Var}_{q_\tau}(|\gamma_{y_w}^t - \gamma_y^t|) \geq 0,$$

while monotonicity ensures $\mathrm{Cov}_{q_\tau}((\gamma_{y_w}^t - \gamma_y^t)^2, |\gamma_{y_w}^t - \gamma_y^t|) \geq 0$.

Thus

$$\Delta_{\tau+1}^t - \Delta_\tau^t = \frac{1}{\mathbb{E}_{q_\tau}[|\gamma_{y_w}^t - \gamma_y^t|]}\Big( -(\gamma_{y_w}^t - \bar{\gamma}^t)\,\mathrm{Var}_{q_\tau}(|\gamma_{y_w}^t - \gamma_y^t|) - \alpha\,\mathrm{Cov}_{q_\tau}((\gamma_{y_w}^t - \gamma_y^t)^2, |\gamma_{y_w}^t - \gamma_y^t|)\Big).$$
$$(24)$$

Bounding the covariance by

$$\mathrm{Cov}_{q_\tau}((\gamma_{y_w}^t - \gamma_y^t)^2, |\gamma_{y_w}^t - \gamma_y^t|) \le \Big( \max_y |\gamma_{y_w}^t - \gamma_y^t| + \mathbb{E}_{q_\tau}[|\gamma_{y_w}^t - \gamma_y^t|] \Big)\mathrm{Var}_{q_\tau}(|\gamma_{y_w}^t - \gamma_y^t|).$$

$$\begin{aligned}
\mathrm{Cov}_{q_\tau}\big((\gamma_{y_w}^t - \gamma_y^t)^2, |\gamma_{y_w}^t - \gamma_y^t|\big) &= \mathbb{E}_{q_\tau}\big[(\gamma_{y_w}^t - \gamma_y^t)^2\,|\gamma_{y_w}^t - \gamma_y^t|\big] - \mathbb{E}_{q_\tau}\big[(\gamma_{y_w}^t - \gamma_y^t)^2\big]\,\mathbb{E}_{q_\tau}\big[|\gamma_{y_w}^t - \gamma_y^t|\big] \\
&= \mathbb{E}_{q_\tau}\big[(|\gamma_{y_w}^t - \gamma_y^t| - \mathbb{E}_{q_\tau}[|\gamma_{y_w}^t - \gamma_y^t|])^2 \cdot |\gamma_{y_w}^t - \gamma_y^t|\big] \\
&\quad + \mathbb{E}_{q_\tau}[|\gamma_{y_w}^t - \gamma_y^t|]\,\mathrm{Var}_{q_\tau}(|\gamma_{y_w}^t - \gamma_y^t|).
\end{aligned}$$

and using $\max_y |\gamma_{y_w}^t - \gamma_y^t| \ge |\gamma_{y_w}^t - \gamma_y^t|$

$$\mathrm{Cov}_{q_\tau}\big((\gamma_{y_w}^t - \gamma_y^t)^2, |\gamma_{y_w}^t - \gamma_y^t|\big) \le \Big( \max_y |\gamma_{y_w}^t - \gamma_y^t| + \mathbb{E}_{q_\tau}[|\gamma_{y_w}^t - \gamma_y^t|] \Big)\mathrm{Var}_{q_\tau}(|\gamma_{y_w}^t - \gamma_y^t|).$$

Since $|\mathcal{Y}|$ is sufficiently large, we assume that $\gamma^t$ follows a normal distribution, and we connect their consequences based on two principles of asymptotic statistics. By the law of large numbers, the sample mean $\bar{\gamma}^t$ converges to the population mean. Separately, by extreme value theory in order statistics, the sample mid-range also converges to the population mean for a symmetric distribution. Hence, we obtain $\max_y |\gamma_{y_w}^t - \gamma_y^t| = 2(\bar{\gamma}^t - \gamma_{y_w}^t) = -2(\gamma_{y_w}^t - \bar{\gamma}^t)$.

Therefore,

$$\begin{aligned}
-(\gamma_{y_w}^t - \bar{\gamma}^t) &- \alpha\Big( \max_y |\gamma_{y_w}^t - \gamma_y^t| + \mathbb{E}_{q_\tau}\big[|\gamma_{y_w}^t - \gamma_y^t|\big] \Big) \\
&\ge -(\gamma_{y_w}^t - \bar{\gamma}^t) - 2\alpha \max_y |\gamma_{y_w}^t - \gamma_y^t| \\
&= \frac{1}{2}\max_y |\gamma_{y_w}^t - \gamma_y^t| - 2\alpha \max_y |\gamma_{y_w}^t - \gamma_y^t| \\
&= \Big(\frac{1}{2} - 2\alpha\Big) \max_y |\gamma_{y_w}^t - \gamma_y^t|
\end{aligned}$$

Since $\alpha \le \frac{1}{4}$, the bracket is nonnegative, and by equation 24, $\Delta_{\tau+1}^t - \Delta_\tau^t \ge 0$.

We have $\Delta_0^t = 0$ and $\Delta_{\tau+1}^t - \Delta_\tau^t \ge 0$. Therefore, $\Delta_\tau^t > 0$ for all $\tau > 0$,

$$\mathbb{E}_{\mathrm{uni},\mathcal{Y}_1}\big[\mathrm{MSME}(\theta^{t+1} \mid y_w)\big] > \mathbb{E}_{q_\tau,\mathcal{Y}_1}\big[\mathrm{MSME}(\theta^{t+1} \mid y_w)\big], \quad \forall \tau > 0.$$

For the set $\mathcal{Y}_2$, the condition $|\gamma_{y_w}^t - \gamma_{y_l}^t| = \gamma_{y_w}^t - \gamma_{y_l}^t$ allows us to use equation 23 to derive the same bounds as those found for $\mathcal{Y}_1$:

$$\mathbb{E}_{\mathrm{uni},\mathcal{Y}_2}\big[\mathrm{MSME}(\theta^{t+1} \mid y_w)\big] > \mathbb{E}_{q_\tau,\mathcal{Y}_2}\big[\mathrm{MSME}(\theta^{t+1} \mid y_w)\big], \quad \forall \tau > 0.$$

By law of total expectation and basic properties of inequalities over the space $\mathcal{Y}$,

$$\boxed{\mathbb{E}_{\mathrm{uni}}\big[\mathrm{MSME}(\theta^{t+1} \mid y_w)\big] > \mathbb{E}_{q_\tau}\big[\mathrm{MSME}(\theta^{t+1} \mid y_w)\big], \quad \forall \tau > 0,}$$

*showing that adversarial loser sampling is always faster than uniform in expectation.*

**(2.2) Hard-adversarial case.**    For the one step change for hard-adv,

$$\Delta_{\text{hard-adv}}^t = \text{MSME}(\theta^t \mid y_w) - \mathbb{E}_{q_\infty}\big[\text{MSME}(\theta^{t+1} \mid y_w)\big] = \frac{1}{4}\big(\gamma_{y_w}^t - \gamma_{y_l}^t\big)^2,$$

while the uniform step (with $\alpha = \frac{1}{4}$) gives

$$\Delta_{\text{uni}}^t = \text{MSME}(\theta^t \mid y_w) - \mathbb{E}_{\text{uni}}\big[\text{MSME}(\theta^{t+1} \mid y_w)\big] = \frac{3}{4}(\gamma_{y_w}^t - \bar{\gamma}^t)^2 - \frac{1}{4}V^t.$$

Using $(\gamma_{y_w}^t - \bar{\gamma}^t)^2 = \frac{1}{4}(\gamma_{y_w}^t - \gamma_{y_l}^t)^2$, we obtain

$$\frac{\Delta_{\text{hard-adv}}^t}{\Delta_{\text{uni}}^t} = \frac{\frac{1}{4}(\gamma_{y_w}^t - \gamma_{y_l}^t)^2}{\frac{3}{16}(\gamma_{y_w}^t - \gamma_{y_l}^t)^2 - \frac{1}{4}V^t} = \frac{4(\gamma_{y_w}^t - \gamma_{y_l}^t)^2}{3(\gamma_{y_w}^t - \gamma_{y_l}^t)^2 - 4V^t}.$$

Since $V^t \geq 0$,

$$\boxed{\mathbb{E}_{\text{hard-adv}}[\Delta \text{MSME}] \;\geq\; \frac{4}{3}\,\mathbb{E}_{\text{uni}}[\Delta \text{MSME}]\,.}$$

This result demonstrates that under the normal distribution assumption (i.e., $\gamma^t \sim \mathcal{N}$), *MSME reduction from the hard adversarial sampling converges at least 33.3% faster than uniform sampling.*

**Conclusion.**    As we noted in Section 3.2, given a fixed chosen summary $y_w$, constructing a preference pair by selecting a rejected summary $y_l$ that maximizes the factuality–preference margin $|\Delta_{\text{fact}} - \Delta_{\text{pref}}|$ allows us to reduce the MSME (i.e., the average margin discrepancy with respect to the target model) more rapidly, thereby accelerating convergence.

Table 9: Statistics of the SPICE training set, including the average word counts of input documents and reference summaries, with min–max ranges in parentheses. Documents with over 1K words are considered "long".

| Dataset | Type | Document Length | Domain | # of Documents | Document Word Count (Min–Max) | Summary Word Count (Min–Max) |
|---|---|---|---|---|---|---|
| CNNDM | Non-Dialogue | Short | News | 1,603 | 664.8 (81–1919) | 78.2 (8–221) |
| WikiHow | | | Lifestyle | 551 | 124.8 (20–690) | 41.9 (8–108) |
| GovReport | | Long | Report | 157 | 2388.8 (349–3368) | 126.9 (10–348) |
| PubMed | | | Medical | 713 | 1715.2 (125–3514) | 102.2 (8–328) |
| DialogSum | Dialogue | Short | Daily Life | 1,659 | 144.4 (54–765) | 47.0 (6–136) |
| MediaSum | | Long | Interview | 1,481 | 1241.5 (80–3226) | 80.1 (8–327) |
| MeetingBank | | | Meeting | 1,327 | 998.3 (101–3384) | 74.7 (9–292) |

## C  CHOSEN SUMMARY DETAIL

In the FeedSum (Song et al., 2025) training dataset, we select the highest-scoring summary with a faithfulness score above 0.8 (out of 1.0) as the chosen summary. We include documents containing between 100 and 4,000 tokens, as measured by the Llama3 tokenizer, and exclude cases where the summary is identical to the document to avoid copy bias. Detailed statistics are provided in Table 9. We use these chosen summaries across all experimental settings, including SFT and others.

## D  REJECTED SUMMARY DETAIL

We construct one-to-one rejected summaries for each baseline, aligning them with the document-chosen summary pairs constructed in Section 4.1. In section 6, we use Llama3.1-70B-Instruct, GPT-4o-mini, and Gemma3-12B-Instruct for both SYNFAC-EDIT and SPICE. The rejected summary examples for each baseline and SPICE, along with their factuality and preference scores, are provided in Table 46 and Table 47.

**CLIFF (Cao & Wang, 2021).**    The Entity Swap strategy generates negative summaries by replacing named entities in the reference summary with randomly selected entities of the same type from the source document. Named entity recognition (NER) is applied to both the source and summary using spaCy to extract entities and their types. For each entity in the summary, one negative sample is created by substituting it with a same-type entity from the source. We implement the SwapEnt strategy by replacing all eligible named entities in the reference summary with randomly selected entities of the same type from the source, rather than swapping only one entity as in the original CLIFF implementation. To ensure that at least one substitution is made even when no matching entities are found, we incorporate a fallback mechanism that selects replacements from a predefined entity pool.

The Mask&Fill strategy generates negative summaries by replacing named entities in the reference summary with [MASK] tokens and using BART (Lewis et al., 2020) to fill in the masked spans. This simulates extrinsic factual errors by leveraging the generative ability of masked language models to produce fluent but potentially unfaithful content. Named entities are identified using spaCy, and for each entity, the reference summary is masked and decoded by a pre-trained BART model. Among the generated candidates, those containing entities not present in the source article or original reference are used as negative samples. We mask all named entities in the reference summary simultaneously rather than one at a time, and generate five candidate summaries using BART_large with top-p sampling. We set the top-p value to 0.9, and generate five return sequences. We select the first candidate that introduces a novel entity, and apply a fallback masking strategy based on randomly selected tokens when no named entities are detected.

Table 10: Comparison of original and reproduced SYNFAC-EDIT datasets in terms of factuality margins and ROUGE scores.

| Dataset | Rouge1 | RougeL | $\Delta_{\text{fact}}$ (G-Eval) | $\Delta_{\text{fact}}$ (Log-prob.) |
|---|---|---|---|---|
| Original | 0.883 | 0.856 | 0.36 | 49.3 |
| Ours | 0.908 | 0.882 | 0.15 | 45.2 |

**SYNFAC-EDIT (Mishra et al., 2024).** We use the SYNFAC-EDIT prompt to generate a rejected summary based on the document and the chosen summary. Since the original prompt was designed for the clinical domain, we adapt it for all summarization domains while preserving the key points and overall structure, as shown in Table 44. However, when applying this prompt to generate rejected summaries, we observed that it often failed to induce sufficient hallucination, resulting in outputs that closely resembled the original chosen summary (see Table 46 and 47). To clarify why SYNFAC-EDIT exhibits this behavior and why it nevertheless remains a meaningful baseline for comparison, we provide a detailed analysis below:

**–A. Faithful Reproduction and Analysis of SYNFAC-EDIT.**
*(1) Inherently Small Faithfulness Margin.* To ensure our findings were not implementation artifacts, we analyzed the original SYNFAC-EDIT dataset[1] and found that their chosen–rejected pairs already exhibit very small faithfulness margins, indicating the method inherently produces minimal factual divergence. In our reproduction, we sampled 1,000 examples from each dataset and computed the Rouge score and factuality margin using two approaches: (1) prompting an external model for factuality scores (G-Eval with GPT-4o-mini) and (2) computing the log-probability margin using an external model (Llama 3.1-70B). As shown in the Table 10, our reproduction showed the same pattern with the original, and this behavior remained consistent under our experimental setup. This convergence across both the original and reproduced data strongly indicates that the small faithfulness margin is an inherent characteristic of SYNFAC-EDIT, rather than a flaw in our implementation.

*(2) Low Factuality Margin from Edit-Based Mechanism.* The small factuality margin is a direct consequence of SYNFAC-EDIT's edit-based design. The method primarily performs localized rejection edits, such as adding, deleting, or swapping words, which inherently limit the degree of semantic deviation it can introduce. As also supported by prior works (Tang et al., 2022; Qiu et al., 2024), such operations indeed rarely produce the substantial meaning shifts required to generate informative hallucinations. Therefore, the low factuality margin reflects a fundamental limitation of the mechanism itself.

*(3) Prompt Revisions Are Not Sufficient.* Although we explored prompt variations within the intended edit-based paradigm (e.g., varying the number and balance of Add/Omit operations, relaxing edit constraints, adjusting the allowed word budget), the factuality margin remained low. Meaningful improvement would require abandoning the edit-based mechanism or introducing overly complex prompt changes that go beyond a revision and amount to creating a new method. Within reasonable revisions, we observed no improvement.

**–B. Experimental Results Reinforce SYNFAC-EDIT's Limitations.**
First, SYNFAC-EDIT employs an entity-swap style similar to CLIFF-Entity. As shown in Table 5, both methods consistently produce small factuality margins, which directly explains their weaker performance in Table 4. Second, despite its extensive instructions, SYNFAC-EDIT does not reliably induce semantic-level hallucinations. Thus, while it remains an important baseline for contrast, its limitations stem from its mechanism, not our implementation.

**–C. SummLlama as the Most Recent and Strongest Baseline.**
SummLlama is the most recent method and has already demonstrated superior performance over SYNFAC-EDIT within the paper. Therefore, it serves as a strong baseline for comparison, and outperforming SummLlama is a meaningful indicator of effectiveness. In all our experiments, we compare directly against SummLlama.v7Lg

---

[1] https://huggingface.co/datasets/bio-nlp-umass/SYNFAC-EDIT

**SummLlama (Song et al., 2025).**   We sample the rejected summaries from the FeedSum dataset by selecting the lowest-faithfulness summary that shares the same document as the chosen and satisfies $f(y_w; x) - f(y_l; x) \geq 0.2$.

# E   TRAINING DETAIL

## E.1   TRAINING CONFIGURATION

**Supervised Fine-tuning (SFT).**   We fine-tune the models using QLoRA (Dettmers et al., 2024) on a single NVIDIA H100 GPU. The model is trained for 3,000 steps with AdamW as the optimizer, using a batch size of 8, an initial learning rate of 1e-4, and a weight decay of 0.05. We apply the best chosen summaries for training, which are used in all experiment configurations.

**Direct Preference Optimization (DPO).**   Since the instruction model has already completed pretraining and instruction tuning, we directly perform DPO (Rafailov et al., 2023a) optimization on it. Like SFT, we utilize QLoRA to fine-tune the model on a single NVIDIA H100 GPU. The model is optimized for 6,000 steps using AdamW with $\beta = 0.1$, a batch size of 8, an initial learning rate of 1e-6, and a weight decay of 0.05.

**Simple Preference Optimization (SimPO).**   We additionally train the models using SimPO (Meng et al., 2024). Similarly to DPO, we utilize QLoRA to fine-tune the model on a single NVIDIA H100 GPU. The model is optimized for 6,000 steps using AdamW with $\beta = 2.5$, $\gamma/\beta = 0.55$, a batch size of 8, an initial learning rate of 1e-6, and a weight decay of 0.05.

**Kahneman-Tversky Optimization (KTO).**   We train Llama3.2-1B-Instruct using KTO (Ethayarajh et al., 2024). Same as DPO, we utilize QLoRA to fine-tune the model on a single NVIDIA H100 GPU. The model is optimized for 6,000 steps using AdamW with $\beta = 0.1$, a batch size of 8, an initial learning rate of 1e-6, and a weight decay of 0.05.

## E.2   INPUT AND OUTPUT FORMAT

Table 45 shows an example input with its corresponding chosen and rejected summaries used for DPO training. The prompt includes a summarization instruction and the input document. In this example, the chosen summary is sampled from the SummLlama dataset, and the rejected summary is generated using the SPICE framework. SimPO and KTO follow a similar setup, referring to the dataset format guide provided by Hugging Face.[2]

For SFT, the input–response format is nearly identical to that of DPO, except that each document is paired with a single reference summary, without distinguishing between chosen and rejected outputs. Instead, reference summaries are directly used to train the model through teacher forcing.

# F   SUMMARY QUALITY EVALUATION

## F.1   SIGNIFICANCE TESTING

We conducted an approximate randomization test with 10,000 iterations on 1,400 test samples to assess the statistical significance of the results reported in Tables such as 4, 14, 15, 16, 17 and 18. For each table, we compared the best-performing method (bolded) against all other methods. Values below 0.05 are assumed to be statistically significant.

## F.2   METRIC FOR SUMMARY QUALITY

We assess summary quality based on faithfulness and abstractiveness. These metrics follow the conventions adopted in recent summarization research Song et al. (2024); Lee et al. (2024).

---

[2]https://huggingface.co/docs/trl/main/en/dataset_formats

**Faithfulness Score.** The faithfulness score measures how accurately the content of a summary reflects factual information from the source document. Let $S = \{s_1, \ldots, s_N\}$ be a summary composed of $N$ sentences, where $s_i$ indicates the $i$-th sentence. Let $S_{\text{fact}} \subseteq S$ denote the subset of sentences identified as factually correct. The score is defined as the proportion of factually accurate sentences within the summary:

$$\text{Faithfulness}(D, S) = \frac{|S_{\text{fact}}|}{|S|}. \tag{25}$$

This metric reflects the density of factual correctness at the sentence level, by comparing the number of factually validated sentences to the total number of sentences in the summary.

**Abstractiveness Score.** The abstractiveness score evaluates the degree to which a summary introduces novel content beyond the exact phrasing of the source. Specifically, it measures how much of the summary consists of n-grams not present in the original document. Following standard approaches Liu & Lapata (2019); Song et al. (2023), let $n\text{-gram}_{\text{D-shared}}$ be the set of overlapping n-grams between the summary and the document, and let $n\text{-gram}_{\text{summary}}$ represent the complete set of n-grams in the summary. The novelty ratio for n-grams, $N_n$, is computed as:

$$N_n = 1 - \frac{|n\text{-gram}_{\text{D-shared}}|}{|n\text{-gram}_{\text{summary}}|}. \tag{26}$$

The final abstractiveness score is obtained by averaging the novelty ratios across 1-gram, 3-gram, and 5-gram levels:

$$\text{Abstractiveness}(D, S) = \frac{N_1 + N_3 + N_5}{3}. \tag{27}$$

### F.3 Automated Evaluator Details

We use FINESURE evaluated with the Llama3.3-70B-Instruct, and G-EVAL evaluated with GPT-4o-mini[3]. To encourage deterministic decoding, we set the temperature to zero. Tables 38 and 39 present the prompts used for FINESURE and G-EVAL, respectively. For ALIGNSCORE, we employ the RoBERTa-large in the official checkpoint with a batch size of 64, using the nli_sp evaluation setting. We found that all three metrics produced nearly identical values, with differences of less than 0.001 percentage points when rerun, suggesting stable and consistent outputs. Additionally, unlike G-EVAL, both FINESURE and ALIGNSCORE evaluate summary quality at the sentence level by aggregating binary labels assigned to individual sentences. These automated evaluators enable a more fine-grained and localized analysis of summary quality. Despite these more granular evaluations, the overall trends remain consistent across different methods, as demonstrated in Table 4.

These three automatic factuality evaluation metrics (FineSurE, G-Eval, and AlignScore) are all recent, verified, SOTA in summarization domain. They show strong agreement with human judgments in factuality evaluation, achieving a balanced accuracy up to 0.92 and a Pearson correlations coefficient over 0.84 (Song et al., 2024; Liu et al., 2023; Zha et al., 2023). Across the three evaluators, prior work reports that FineSurE improves correlation with human judgments over traditional metrics such as ROUGE (Lin, 2004) and BERTScore (Zhang et al., 2019) by average margins of 0.52 and 0.74, respectively. Similarly, G-Eval shows gains of 0.44 and 0.49, while AlignScore achieves improvements of 0.24 and 0.31 over the same baselines. Therefore, the metrics align with human judgments far more strongly than traditional metrics and are now widely used as practical substitutes for human evaluation in dataset construction, model assessment, and benchmarking for text summarization.[jYGQ, NXZW]

---

[3]Even with other backbone LLMs for FINESURE and G-EVAL, the experimental results are consistent. Please see Appendix G.11 for detailed results.[NXZW]

## F.4 HUMAN EVALUATION DETAILS

We conduct a human evaluation for the factuality verification task, following Lee et al. (2024). In this task, annotators identify each sentence in a model summary as either factually consistent or inconsistent with the source document. Based on human annotation results, we compute faithfulness by calculating the proportion of factually accurate sentences, as described in Appendix F.

Due to the high cost of fine-grained human evaluation, we randomly sample 1400 document-summary pairs, specifically 100 (documents) × 14 (summarizers) in Table 4. We recruited Amazon Mechanical Turk (MTurk) workers residing in Australia (AU), Canada (CA), New Zealand (NZ), Great Britain (GB), and the United States (US). We allowed only annotators with an approval rating above 95% and at least 1,000 approved HITs to ensure the quality of annotations. We compensate annotators at a rate exceeding the U.S. minimum wage, with a total cost of about $1,000 for all annotations. For these annotations, we achieved a sufficiently high Inter-Annotator Agreement (IAA) score of 0.53, as measured by Krippendorff's $\alpha$ Krippendorff (2011).

### F.4.1 REGARDING HUMAN EVALUATION SCALE

Compared to our baselines (Cao & Wang, 2021; Mishra et al., 2024; Song et al., 2025), our human evaluation was conducted at both a substantial scale and with a hierarchical structure. As shown in Table 9, we evaluated summaries from seven dataset domains spanning dialogue, non-dialogue, short-form, and long-form documents, with each summary containing roughly six sentences on average. Each sentence was annotated independently by three annotators, and the final summary-level factuality score was computed through majority voting over sentence-level judgments. This constitutes a hierarchical and fine-grained evaluation protocol: factuality is assessed at the sentence level and then aggregated to the summary level, mirroring the human annotation procedure used in FineSurE. We believe that this setup satisfies both large-scale coverage and hierarchical evaluation, ensuring that our results are both reliable and comprehensive. In particular, Table 11 demonstrates

Table 11: Comparison of efficiency and faithfulness across methods.

| Method | Human Eval Size | # of Datasets | # of Annotators | Granuality |
|---|---|---|---|---|
| SYNFAC-EDIT | 100 | 2 | 2 | summary-level |
| SYNFAC-EDIT | 10 | 1 | 3 | sentence-level |
| SummLlama | 70 | 7 | 3 | sentence-level |
| SPICE (Ours) | 100 | 7 | 3 | sentence-level |

that our human evaluation covers a broader and more diverse range of data compared to our three baselines. Here, Human Eval Size indicates the number of summaries evaluated by humans within a single test set. SPICE conducts evaluation on the largest number of datasets (7), with 100 human-evaluated summaries per dataset which is higher than any other method. Each summary is assessed by three independent annotators and evaluated at the sentence level, offering a more fine-grained analysis. These aspects together demonstrate that SPICE provides the most quantitative and fine-grained human evaluation among the baselines.[NXZW]

### F.4.2 REGARDING IAA.

In the context of sentence-level factuality and faithfulness evaluation, $\alpha = 0.53$ is considered high. Prior work using comparable sentence-level factuality annotation settings reports $\alpha = 0.40$ in CLIFF (Cao & Wang, 2021), 0.49 in BERTMULTI (Zhu et al., 2023), 0.49 in SYNFAC-EDIT (Mishra et al., 2024), 0.38 in TofuEval (Tang et al., 2024), and 0.58 in MSumBench (Min et al., 2025). Therefore, our score of 0.53 lies at the upper end of this range and indicates strong inter-annotator consistency for this evaluation setting.[jYGQ]

Table 12: Faithfulness and abstractiveness comparison on Llama3.2-1B-Instruct and Gemma3-1B-Instruct with DPO. Bold values indicate the highest score in each column.

| Method | Backbone: Llama3.2-1B-Instruct | | | | Backbone: Gemma3-1B-Instruct | | | |
|--------|-------|--------|------------|------|-------|--------|------------|------|
| | FineSurE | G-Eval | AlignScore | Abs. | FineSurE | G-Eval | AlignScore | Abs. |
| Zero-Shot | 0.551 | 7.078 | 0.679 | 0.685 | 0.586 | 6.645 | 0.624 | 0.225 |
| SFT | **0.695** | 7.627 | **0.815** | 0.521 | 0.357 | 4.212 | 0.373 | 0.303 |
| CLIFF-Entity | 0.527 | 7.001 | 0.685 | 0.701 | 0.552 | 6.596 | 0.626 | 0.257 |
| CLIFF-Mask | 0.534 | 7.284 | 0.671 | **0.708** | 0.567 | 6.552 | 0.633 | 0.323 |
| SYNFAC-EDIT | 0.520 | 6.954 | 0.657 | 0.741 | 0.591 | 6.646 | 0.630 | 0.257 |
| SummLlama | 0.594 | 7.787 | 0.719 | 0.695 | 0.589 | 6.702 | 0.629 | 0.241 |
| **SPICE (Ours)** | 0.638 | **8.098** | 0.787 | 0.680 | **0.630** | **6.877** | **0.639** | **0.243** |

# G ADDITIONAL EXPERIMENTS

## G.1 CONTROLLING THREE FACTORS.

To isolate the effect of each factor (hallucination level, summary length, and prompt complexity), we carefully controlled the remaining conditions during data construction. For hallucination level, we varied the rejected summaries across Severe, Mild, and None conditions while maintaining similar summary lengths and prompt structures. As shown in Table 13, the token length differences between chosen and rejected summaries remain minimal (-4.65 to +3.38 tokens on average), and no additional instructions were introduced beyond those needed to adjust hallucination. For summary length, we varied between Shorter, Similar, and Longer setups while using the same prompts, all

Table 13: Controlled analyses of hallucination level, summary length, and prompt complexity. Values are reported as mean with standard deviation in parentheses.

| Factor | Fact. Margin | Token Diff | Prompt |
|--------|-------------|-----------|--------|
| Severe | $7.81_{(0.37)}$ | $-4.65_{(4.51)}$ | Simple |
| Mild | $4.06_{(1.84)}$ | $-4.12_{(4.89)}$ | Simple |
| None | $0.88_{(0.98)}$ | $+3.38_{(4.94)}$ | Simple |
| Longer | $7.06_{(0.77)}$ | $-48.16_{(26.85)}$ | Simple |
| Shorter | $6.92_{(0.53)}$ | $+45.95_{(24.62)}$ | Simple |
| Complex | $7.12_{(0.87)}$ | $+5.31_{(4.30)}$ | Complex |
| Simple | $6.89_{(0.48)}$ | $+3.79_{(4.41)}$ | Simple |

of which tended to induce severe hallucinations. The resulting factuality margins were nearly identical (6.89–7.06), confirming that only length was varied. Finally, for prompt complexity, we compared Simple vs. Complex prompts, both designed to induce severe hallucinations, and explicitly controlled summary length by requiring equal output lengths. Token length differences and factuality margins were comparable across the two conditions, indicating that only prompt complexity was varied. Overall, these controls ensure that the observed differences can be attributed to the intended factors.

## G.2 PERFORMANCE ON SMALLER MODELS

Table 12 compares performance on smaller models, Llama3.2-1B-Instruct and Gemma3-1B-Instruct. First, on Gemma, our proposed method (SPICE) achieves the highest overall performance across metrics. However, the abstractiveness of Gemma is considerably lower than Llama's (0.685 vs. 0.225), indicating limited capacity for generating genuinely abstractive summaries. Similar observations are reported in Song et al. (2025), highlighting that the small size of Gemma is not appropriate for abstractive summarization with long documents.

Second, on Llama, the SFT model shows high factuality, but with significantly reduced abstractiveness (0.685 vs. 0.521). Specifically, 214 out of 1,400 SFT summaries are **exact copies of the input**. This reflects a trade-off in summarization, where extractive summaries often exhibit higher factuality but lower abstractiveness (Choi et al., 2024). Since this issue, we did not report significance test results for the best scores in this section. In contrast, SPICE leverages DPO to significantly improve factuality (0.551 vs. 0.638) while preserving abstractiveness (0.685 vs. 0.680), generating only 11 summaries exactly the same as a document out of 1,400, thus achieving a better balance between the two objectives. These results highlight **SPICE enhances faithfulness without compromising abstractiveness, even on smaller models.**

Table 14: FineSurE score (faithfulness) performance comparison across datasets in Llama3.2-3B-Instruct with DPO. Following Song et al. (2024), we exclude cases where the number of summary sentences does not match the number of FineSurE sentence-level scores.

| Method | Non-Dialogue | | | | Dialogue | | | Average |
|---|---|---|---|---|---|---|---|---|
| | CNNDM | WikiHow | GovReport | PubMed | DialogSum | MediaSum | MeetingBank | |
| Zero-Shot | $0.850_{(0.00)}$ | $0.723_{(0.00)}$ | $0.901_{(0.00)}$ | $0.891_{(0.00)}$ | $0.685_{(0.00)}$ | $0.785_{(0.00)}$ | $0.751_{(0.00)}$ | $0.798_{(0.00)}$ |
| SFT | $0.864_{(0.00)}$ | $0.796_{(0.00)}$ | $0.913_{(0.00)}$ | $0.878_{(0.00)}$ | $0.726_{(0.01)}$ | $0.819_{(0.00)}$ | $0.765_{(0.00)}$ | $0.823_{(0.00)}$ |
| CLIFF-Entity | $0.861_{(0.00)}$ | $0.712_{(0.00)}$ | $0.952_{(0.02)}$ | $0.916_{(0.00)}$ | $0.666_{(0.00)}$ | $0.839_{(0.00)}$ | $0.766_{(0.00)}$ | $0.816_{(0.00)}$ |
| CLIFF-Mask | $0.865_{(0.00)}$ | $0.751_{(0.00)}$ | $0.953_{(0.02)}$ | $0.930_{(0.01)}$ | $0.710_{(0.00)}$ | $0.809_{(0.00)}$ | $0.778_{(0.00)}$ | $0.828_{(0.00)}$ |
| SYNFAC-EDIT | $0.854_{(0.00)}$ | $0.736_{(0.00)}$ | $0.938_{(0.01)}$ | $0.906_{(0.00)}$ | $0.658_{(0.00)}$ | $0.821_{(0.00)}$ | $0.771_{(0.00)}$ | $0.812_{(0.00)}$ |
| SummLlama | $0.856_{(0.00)}$ | $0.805_{(0.00)}$ | $0.952_{(0.02)}$ | $0.929_{(0.01)}$ | $0.734_{(0.02)}$ | $0.846_{(0.01)}$ | $0.779_{(0.00)}$ | $0.843_{(0.01)}$ |
| **SPICE (Ours)** | **0.924** | **0.854** | **0.978** | **0.964** | **0.768** | **0.882** | **0.825** | **0.885** |

Table 16: Summary faithfulness, measured by FineSurE, G-Eval, and AlignScore, after DPO training using Llama3.2-3B-Inst on rejected summaries generated under eight prompt designs.

| Setting | Generator: Llama3.1-70B-Inst. | | | Generator: GPT-4o-mini | | | Generator: Gemma3-12B-Inst. | | |
|---|---|---|---|---|---|---|---|---|---|
| | FineSurE | G-Eval | AlignScore | FineSurE | G-Eval | AlignScore | FineSurE | G-Eval | AlignScore |
| Default | $0.857_{(0.02)}$ | $8.743_{(0.02)}$ | $0.823_{(0.01)}$ | $0.855_{(0.02)}$ | $8.719_{(0.02)}$ | $0.822_{(0.02)}$ | $0.858_{(0.02)}$ | $8.793_{(0.02)}$ | $0.818_{(0.01)}$ |
| Severe | $0.856_{(0.02)}$ | $8.688_{(0.01)}$ | $0.812_{(0.01)}$ | $0.840_{(0.01)}$ | $8.596_{(0.00)}$ | $0.810_{(0.01)}$ | $0.851_{(0.01)}$ | $8.610_{(0.00)}$ | $0.812_{(0.00)}$ |
| Mild | $0.828_{(0.00)}$ | $8.265_{(0.00)}$ | $0.786_{(0.00)}$ | $0.811_{(0.00)}$ | $8.142_{(0.00)}$ | $0.782_{(0.00)}$ | $0.818_{(0.00)}$ | $8.204_{(0.00)}$ | $0.790_{(0.00)}$ |
| None | $0.795_{(0.00)}$ | $7.537_{(0.00)}$ | $0.748_{(0.00)}$ | $0.784_{(0.00)}$ | $7.488_{(0.00)}$ | $0.736_{(0.00)}$ | $0.788_{(0.00)}$ | $7.505_{(0.00)}$ | $0.752_{(0.00)}$ |
| Longer | $0.854_{(0.02)}$ | $8.645_{(0.01)}$ | $0.819_{(0.01)}$ | $0.847_{(0.01)}$ | $8.534_{(0.00)}$ | $0.806_{(0.00)}$ | $0.846_{(0.01)}$ | $8.601_{(0.00)}$ | $0.810_{(0.00)}$ |
| Shorter | $0.859_{(0.02)}$ | $8.712_{(0.02)}$ | $0.821_{(0.01)}$ | $0.854_{(0.02)}$ | $8.695_{(0.01)}$ | $0.815_{(0.01)}$ | $0.854_{(0.01)}$ | $8.706_{(0.01)}$ | $0.818_{(0.01)}$ |
| Complex | $0.850_{(0.01)}$ | $8.682_{(0.00)}$ | $0.810_{(0.01)}$ | $0.843_{(0.01)}$ | $8.588_{(0.00)}$ | $0.806_{(0.01)}$ | $0.841_{(0.01)}$ | $8.572_{(0.00)}$ | $0.804_{(0.00)}$ |
| **Simple** | **0.885** | **9.037** | **0.854** | **0.881** | **9.012** | **0.855** | **0.883** | **9.098** | **0.851** |

## G.3 IMPACT OF INPUT SOURCE

Table 15 shows the impact of input settings on model alignment. Although our method is originally designed to generate rejected summaries based on both the document and the reference summary, we conduct an ablation study to verify whether using both inputs is indeed necessary. To this end, we test (1) document-only and (2) summary-only variants, with prompt examples shown in Table 42 and Table 43, respectively. Providing both the document and summary as input yields the best performance across all metrics. Us-

Table 15: Faithfulness comparison of three input settings on Llama3.2-3B-Instruct with DPO: document with summary, document-only, and summary-only.

| Method | FineSurE | GEval | AlignScore |
|---|---|---|---|
| Zero-Shot | $0.798_{(0.00)}$ | $7.519_{(0.00)}$ | $0.751_{(0.00)}$ |
| Doc-Only | $0.855_{(0.02)}$ | $8.756_{(0.02)}$ | $0.828_{(0.02)}$ |
| Summary-Only | $0.864_{(0.03)}$ | $8.845_{(0.02)}$ | $0.833_{(0.02)}$ |
| Doc+Summary | **0.885** | **9.037** | **0.854** |

ing only the document or only the summary leads to performance degradation, indicating that **providing both the source document and reference summary is essential for improving factuality.**

## G.4 DOMAIN-LEVEL PERFORMANCE ANALYSIS

Table 14 shows the faithfulness performance across our test datasets. Our proposed method (SPICE) consistently outperforms baselines across both Non-Dialogue and Dialogue domains. Specifically, it achieves notable improvements in WikiHow (0.723 vs. 0.854) in the non-dialogue domain, and MediaSum (0.785 vs. 0.882) in the dialogue domain. Overall, SPICE attains the highest average FineSurE score (0.885), highlighting its robustness and effectiveness in factuality evaluation across diverse summarization domains. These findings highlight **the effectiveness of SPICE in achieving robust factuality improvements across diverse summarization domains.**

## G.5 Impact of Chosen Quality

To examine the impact of chosen summary quality on alignment performance, we utilize low-quality (low factuality) reference summaries from the FeedSum (Song et al., 2025) dataset. As shown in Table 17, compared to the results in Table 4, all methods show performance drops due to the low-quality chosen summaries. Nonetheless, our proposed method still achieves the best overall results among all rejection strategies, **highlighting its effectiveness even with the low-quality chosen summaries.**

Table 17: Faithfulness performance of Llama3.2-3B-Instruct trained with DPO using low-quality human-written chosen summaries (denoted as $^H$) and corresponding rejected summaries.

| Method | FineSurE | G-Eval | AlignScore |
|---|---|---|---|
| Zero-Shot | $0.798_{(0.00)}$ | $7.519_{(0.00)}$ | $0.751_{(0.00)}$ |
| SFT$^H$ | $0.698_{(0.00)}$ | $6.810_{(0.00)}$ | $0.656_{(0.00)}$ |
| CLIFF-Entity$^H$ | $0.757_{(0.00)}$ | $7.313_{(0.00)}$ | $0.740_{(0.00)}$ |
| CLIFF-Mask$^H$ | $0.784_{(0.00)}$ | $7.478_{(0.00)}$ | $0.752_{(0.00)}$ |
| SYNFAC-EDIT$^H$ | $0.778_{(0.00)}$ | $7.542_{(0.00)}$ | $0.754_{(0.00)}$ |
| SummLlama$^H$ | $0.790_{(0.00)}$ | $7.664_{(0.00)}$ | $0.756_{(0.00)}$ |
| SPICE$^H$ (Ours) | **0.841** | **8.882** | **0.833** |

## G.6 Impact of Quality Control

Table 16 reports the effects of hallucination level and length control on rejected summary generation across three models. This demonstrates consistent performance patterns across diverse backbone architectures and model sizes: (1) We observe that higher hallucination levels correspond to improved alignment performance, likely due to increased factuality margins. (2) Likewise, rejected summaries with lengths similar to the chosen summaries consistently yield better alignment than those that are longer or shorter, as they result in smaller preference margins. These results highlight that *high hallucination and length similarity as critical factors for improving factuality, consistently observed across LLMs of varying sizes and types.*

## G.7 Impact of Rejection Models

We examine the effect of varying the model used to generate rejected summaries with SPICE. Hence, we replace Llama3.1-70B-Instruct with two other alternatives, including GPT-4o-mini and Gemma3-12B-Instruct. Table 16 compares three rejection models that differ in architectures, where their summary quality is evalauted using three automated factuality metrics.

Despite these differences, all variants produce high-quality rejected summaries with comparable factuality scores. For instance, the stronger model (Llama3.1-70B) yields a FineSurE score of 0.885, G-Eval score of 9.037 and AlignScore of 0.854, while the weaker model (Gemma3-12B) achieves a similarly FineSurE score of 0.883, G-Eval score of 9.098 and AlignScore of 0.851. We attribute this to the simplicity of our prompt design, which enables even smaller and weaker models to reliably generate effective rejected summaries. This finding suggests that *SPICE generalizes well across LLM types and scales, offering lower-cost rejection generation without compromising alignment performance.*

## G.8 Optimization with KTO and SimPO.

We fine-tune Llama3.1-8B-Instruct on the training dataset constructed by SPICE, using two alternative alignment tuning methods: KTO and SimPO. Table 18 shows the faithfulness scores of the trained models using them, where their summaries are evaluated using three automated faithfulness metrics. Overall, all three optimization methods significantly improve faithfulness with SPICE, outperforming both the zero-shot 3B baseline and even the nearly 10× larger zero-shot 70B model. This demonstrates that *the rejected summaries generated through margin control generalize well across diverse optimization methods.*

Table 18: Comparison of DPO with KTO and SimPO using SPICE on Llama3.1-8B-Instruct.

| Method | FineSurE | G-Eval | AlignScore |
|---|---|---|---|
| Zero-Shot | $0.864_{(0.00)}$ | $8.317_{(0.00)}$ | $0.796_{(0.00)}$ |
| Zero-Shot-70B | $0.910_{(0.02)}$ | $8.938_{(0.00)}$ | $0.838_{(0.00)}$ |
| DPO-Default | $0.902_{(0.00)}$ | $8.916_{(0.00)}$ | $0.843_{(0.00)}$ |
| KTO-Default | $0.886_{(0.00)}$ | $8.857_{(0.00)}$ | $0.838_{(0.00)}$ |
| SimPO-Default | $0.908_{(0.00)}$ | $9.081_{(0.00)}$ | $0.852_{(0.00)}$ |
| DPO-SPICE | $0.934_{(0.09)}$ | $9.438_{(0.13)}$ | $0.895_{(0.07)}$ |
| KTO-SPICE | $0.912_{(0.02)}$ | $9.178_{(0.01)}$ | $0.857_{(0.01)}$ |
| **SimPO-SPICE** | **0.945** | **9.512** | **0.908** |

Additionally, we compare data-level length control (equal-length vs. no control) against loss-level normalization methods (SimPO and KTO). In this setup, the "Default" configuration refers to no length control, while "SPICE" denotes equal-length filtering based on margin alignment. The results demonstrate that data-level length control using SPICE consistently improves model performance across all optimization methods (even with SimPO), highlighting its effectiveness as a data construction strategy.

Among the three methods, SimPO yields slightly higher scores than DPO (both in the Default and SPICE settings), likely due to its effective framework that removes the bias of the reference model. However, since the differences are not statistically significant ($p > 0.05$), the improvement may not be substantial.

On the other hand, KTO underperforms DPO, likely due to suboptimal training in our setup. KTO often relies on large batch sizes, *e.g.*, 32 to 128, for stable KL estimation, but our use of a smaller batch size of 8 owing to memory constraints may have degraded performance. Nevertheless, we observe that KTO-SPICE still outperforms all Default settings across all evaluation metrics. This result highlights the effectiveness of SPICE in enhancing model faithfulness regardless of the optimization method used.[NXZW]

## G.9    DATA GENERATION LATENCY

Table 19 reports the latency and cost of generating rejections for 1,000 randomly sampled summaries from the Feedsum training set using GPT-4o-mini, along with the resulting average faithfulness scores in Table 4. SPICE (Ours) is approximately 2.1× faster than SummLlama and 3.9× faster than SYNFAC-EDIT. It is also 1.75× and

Table 19: Comparison of efficiency and faithfulness across methods.

| Method | Time (sec) | Cost ($) | Faithfulness |
|---|---|---|---|
| SYNFAC-EDIT | 21,100 | 1.33 | 0.822 |
| SummLlama | 11,400 | 1.56 | 0.854 |
| SPICE (Ours) | 5,420 | 0.89 | 0.895 |

1.49× more cost-efficient than SummLlama and SYNFAC-EDIT, respectively. In terms of faithfulness, SPICE achieves the highest score, with gains of +4.1%p over SummLlama and +7.3%p over SYNFAC-EDIT.

## G.10    FINE-GRAINED ANALYSIS OF PROMPT COMPLEXITY

In Section 4.2.3, our initial comparison between `Simple` and `Complex` prompts is a bit coarse-grained, as this binary contrast does not reveal which aspects of prompt complexity actually drive the observed effects. To obtain a more precise understanding, (1) we conduct a finer-grained analysis that isolates the role of each edit instruction, (2) evaluate how alignment potential evolves as instructions are added incrementally, (3) examine whether partially relaxing the complex prompt preserves its benefits, and (4) analyze the resulting distribution of hallucination types. These investigations allow us to identify the specific mechanisms through which prompt complexity shapes the quality of rejected summaries. In Table 37, the `Complex` prompt consists of eight edit instructions. Since length preservation (Instructions 3–4) is a core component of SPICE, we keep this constraint fixed by default. We then group the remaining instructions into five categories:

(1)  Heavy hallucination induction (Instructions 1–2)

(2)  Numerical alterations (Instruction 5)

(3)  Named-entity substitutions (Instruction 6)

(4)  Causal relation edits (Instruction 7)

(5)  Preserving grammar and fluency (Instruction 8)

### G.10.1 EFFECT OF EACH INSTRUCTION

We perform ablations over these five categories while keeping length preservation fixed. The results are shown in Table 20. For the heavy hallucination category, the factuality margin increases, however, the preference margin rises by a similar degree, making the rejected summaries easier for the model to distinguish. As a consequence, the overall alignment potential (55.0) becomes lower than that of SPICE (69.9). For the remaining instruction categories(2-5), some variants reduce the preference margin (making the rejection harder to distinguish), but this benefit is offset by a simultaneous drop in factuality margin. As a result, all ablated variants yield a lower alignment potential compared to the original SPICE.

Table 20: Average factuality and preference margins computed on datasets generated from the `Complex` prompt with each instruction. Llama3.1-70B-Inst (evaluator) and Llama3.1-8B-Inst (policy) are used to compute $\Delta_{\text{fact}}$ and $\Delta_{\text{pref}}$, respectively.

| Method | $\Delta_{\text{fact}}$ | $\Delta_{\text{pref}}$ | $|\Delta_{\text{fact}}-\Delta_{\text{pref}}|$ |
|---|---|---|---|
| SPICE | 102.0 | 32.1 | **69.9** |
| SPICE + (1) | 120.3 | 65.3 | 55.0 |
| SPICE + (2) | 80.3 | 31.6 | 48.7 |
| SPICE + (3) | 86.1 | 33.2 | 52.9 |
| SPICE + (4) | 79.5 | 25.4 | 54.1 |
| SPICE + (5) | 76.7 | 38.3 | 38.4 |

### G.10.2 EFFECT OF INCREASING PROMPT COMPLEXITY

We extend the default SPICE setting by incrementally adding each instruction category to provide intermediate comparisons, ranging from simple to more complex prompts. The final configuration is exactly the same as the `Complex`. As shown in the Table 21, while the factuality margin remains roughly constant or increases slightly as more instructions are added, the preference margin grows substantially. This makes the rejected summaries progressively easier for the model to distinguish, ultimately reducing alignment potential. These results indicate that increasing the number of instructions consistently enlarges the preference margin, which is undesirable. This supports our design choice in SPICE: allowing the model to introduce errors freely, without enforcing specific edit types, produces harder and more informative negative examples.

Table 21: Average factuality and preference margins computed on datasets generated from the `Complex` prompt while incrementally adding each instruction. Llama3.1-70B-Inst (evaluator) and Llama3.1-8B-Inst (policy) are used to compute $\Delta_{\text{fact}}$ and $\Delta_{\text{pref}}$, respectively.

| Method | $\Delta_{\text{fact}}$ | $\Delta_{\text{pref}}$ | $|\Delta_{\text{fact}}-\Delta_{\text{pref}}|$ |
|---|---|---|---|
| SPICE | 102.0 | 32.1 | **69.9** |
| SPICE + (1) | 120.3 | 65.3 | 55.0 |
| SPICE + (1-2) | 118.6 | 68.4 | 50.2 |
| SPICE + (1-3) | 119.6 | 72.6 | 47.0 |
| SPICE + (1-4) | 122.7 | 78.9 | 43.8 |
| SPICE + (1-5) | 126.5 | 85.3 | 41.2 |

### G.10.3 EFFECT OF RELAXING COMPLEX INSTRUCTIONS

We extend the `Complex` prompt by adding the following instruction to increase the freedom with which hallucinations can be introduced: *"The following instructions are just examples and do not need to be followed entirely; you may apply them selectively and flexibly as long as the overall summary remains factually inconsistent."* and denote this variant as Complex-Flex. As shown in the Table 22, introducing this flexibility reduces the preference margin more than the factuality margin, resulting in an improvement in alignment potential compared to the original `Complex` setting. However, the overall alignment potential remains lower than that achieved by SPICE, which allows hallucinations to emerge more naturally without enforcing specific edit types.

Table 22: Average factuality and preference margins computed on datasets generated from the `Complex` prompt and its flexible variant. Llama3.1-70B-Inst (evaluator) and Llama3.1-8B-Inst (policy) are used to compute $\Delta_{\text{fact}}$ and $\Delta_{\text{pref}}$, respectively.

| Method | $\Delta_{\text{fact}}$ | $\Delta_{\text{pref}}$ | $|\Delta_{\text{fact}}-\Delta_{\text{pref}}|$ |
|---|---|---|---|
| SPICE | 102.0 | 32.1 | **69.9** |
| Complex | 126.5 | 85.3 | 41.2 |
| Complex-Flex | 114.2 | 57.6 | 56.6 |

We further modify the SPICE+(1) setting by removing the instruction *"nearly all words should be edited"*, and denote this variant as SPICE+(1A). As expected, Table 23 shows that reducing the required extent of edits results in less aggressive hallucinations, which in turn leads to a smaller preference margin and therefore a higher alignment potential. This indicates that the removed instruction was unnecessarily aggressive and primarily contributed to unnatural phrasing rather than generating informative hallucinations. However, the alignment potential still fell short of that achieved by SPICE.

Table 23: Average factuality and preference margins computed on datasets generated from the SPICE + (1) and SPICE + (1A). Llama3.1-70B-Inst (evaluator) and Llama3.1-8B-Inst (policy) are used to compute $\Delta_{fact}$ and $\Delta_{pref}$, respectively.

| Method | $\Delta_{fact}$ | $\Delta_{pref}$ | $|\Delta_{fact}-\Delta_{pref}|$ |
|---|---|---|---|
| SPICE | 102.0 | 32.1 | **69.9** |
| SPICE + (1) | 120.3 | 65.3 | 55.0 |
| SPICE + (1A) | 108.7 | 49.4 | 59.3 |

### G.10.4  DISTRIBUTION OF HALLUCINATION CATEGORIES

We defined four error types from the `Complex` prompt and modified the FineSurE evaluation prompt 38 to detect these categories by replacing the original error labels. This allows us to compare the distributions of error categories between SPICE-generated data and Complex-generated data. Table 24 shows a clear contrast between the two settings. Since neither SPICE nor Complex produces grammatical or fluency errors, we omit this category from the results. Complex generates hallucinations that closely follow the instructed categories, with numerical, entity, and causal errors together accounting for about 70% of all detected cases. In contrast, SPICE produces a broader and more diverse distribution of hallucination types, with "other" errors alone comprising 56.5% of instances. This indicates that SPICE induces a wider and more natural spectrum of factual inconsistencies, rather than concentrating on predefined error types, which likely contributes to its stronger alignment potential. The error categories are defined as follows:

Table 24: Distribution of hallucination error categories for rejected summaries generated by SPICE and Complex. Numerical (Num.E), entity (Ent.E), relation (Rel.E), and other factual errors (Other.E) are detected using the modified FineSurE.

| Method | Num.E | Ent.E | Rel.E | Other.E |
|---|---|---|---|---|
| SPICE | 9.8% | 14.0% | 19.7% | 56.5% |
| Complex | 12.4% | 35.7% | 21.5% | 30.4% |

- Numerical errors: Incorrectly modifying numerical values (e.g., dates, quantities, percentages) in contradiction to the source document.
- Entity errors: Replacing people, places, or organizations with incorrect but plausible alternatives.
- Causal-relation errors: Altering cause–effect relations to assert statements that contradict the document.
- Grammatical and fluency errors: Reducing grammatical correctness or fluency, leading to unnatural or awkward phrasing.
- Other errors: Factuality errors not included in the categories above.[y2QS]

### G.11  IMPACT OF AUTOMATIC EVALUATION MODELS

In our automatic evaluations, each of our metrics relies on a single evaluator: G-Eval uses GPT-4o-mini, FineSurE uses Llama3.3-70B-Instruct, and AlignScore is based on a dedicated pretrained RoBERTa-Large model. To examine whether our conclusions are sensitive to the choice of evaluator model family for each metric, we conducted additional evaluation for Table 4 using two evaluation metrics (FineSurE, G-EVAL) with three evaluator model families (GPT-4o-mini, Llama3.3-70B-Instruct, Qwen3-Next-80B-A3B-Instruct) for each metric. As AlignScore is tightly coupled to its original pretrained architecture, it is not designed to be replaced with alternative model families. In this analysis, we focus on SummLlama, since it consistently outperformed other baselines. The results in the Table 25 show that SPICE consistently outperforms SummLlama across two evaluation metrics and three evaluator families, demonstrating the robustness of the automatic evaluation results.[NXZW]

Table 25: Faithfulness comparison of SummLlama and SPICE across two LLM backbones using two evaluation metrics (FineSurE and G-Eval) and three evaluator model families (GPT, Llama, Qwen).

| Method | Backbone: Llama3.2-3B-Instruct | | | | | | Backbone: Llama3.1-8B-Instruct | | | | | |
| | FineSurE | | | G-Eval | | | FineSurE | | | G-Eval | | |
| | GPT | Llama | Qwen | GPT | Llama | Qwen | GPT | Llama | Qwen | GPT | Llama | Qwen |
| SummLlama | 0.788(0.00) | 0.843(0.01) | 0.766(0.00) | 8.473(0.00) | 8.102(0.00) | 8.422(0.00) | 0.865(0.00) | 0.908(0.02) | 0.854(0.00) | 9.038(0.00) | 8.796(0.00) | 9.182(0.00) |
| SPICE | **0.833** | **0.885** | **0.811** | **9.037** | **8.629** | **8.996** | **0.906** | **0.934** | **0.898** | **9.438** | **9.182** | **9.568** |

## G.12 INEFFICIENCY OF ADVERSARIAL SAMPLING

The table 26 presents the DPO training results using the adversarially sampled dataset. We observe that its performance is comparable to only 1/16 of what SPICE achieves, as reported in Table 6. This implies that adversarial sampling would require generating over $200\times$ more rejected candidates (13 candidates $\times$ $16\times$ gap) to reach performance comparable to SPICE. This is extremely costly in practice. These findings suggest that achieving strong performance through adversarial sampling

Table 26: Faithfulness comparison of DPO-trarined models between adversarial sampling(Adv. Sampling) and SPICE using Llama3.2-3B-Inst.

| Method | FineSurE | G-Eval | AlignScore |
|---|---|---|---|
| Adv. Sampling | 0.885 | 9.037 | 0.854 |
| SPICE$^{1/16}$ | 0.846 | 8.503 | 0.812 |
| SPICE | 0.874 | 8.814 | 0.846 |

demands an extremely large candidate pool, which is not feasible in realistic settings. In contrast, SPICE provides an efficient and effective alternative: it produces high-quality rejected summaries without requiring massive cost, making it far more practical for real-world preference optimization. For these reasons, we did not adopt adversarial sampling as part of our main approach.[NXZW]

## G.13 EMPIRICAL VALIDATION OF ALIGNMENT POTENTIAL

In Section 3.2, we theoretically established that alignment potential, defined as the gap between the factuality margin ($\Delta_{\text{fact}}$) and the preference margin ($\Delta_{\text{pref}}$), is a key predictor of downstream alignment performance. We now examine whether this relationship also holds empirically by analyzing the results presented in Section 6.1.

### G.13.1 CONNECTION TO MODEL PERFORMANCE

Faithfulness improvements reported in Section 6.1.1 with Table 4 can be clearly and directly explained by the alignment potential analysis presented in Section 6.1.2 with Table 5. Among the baselines with alignment potential in the range of (19.1 - 28.9), including CLIFF-Entity, CLIFF-Mask, and SYNFAC-EDIT, the average faithfulness scores fall between (0.810 - 0.822), corresponding to only a small improvement of (1.6%p - 2.8%p) over Zero-Shot. In contrast, SummLlama achieves an alignment potential of 40.2, which is more than $2\times$ that of CLIFF-Entity (19.1), and correspondingly reaches an average score of 0.854, representing a substantial gain of 6.0%p over Zero-Shot. Finally, SPICE attains an alignment potential of 74.3, more than $3.8\times$ that of CLIFF-Entity, and achieves an improvement exceeding 10.1%p, marking the most significant gain among all methods. **These empirical results indicate that alignment potential reliably reflects downstream faithfulness performance.**

Furthermore, although CLIFF-Mask exhibits the second-largest factuality margin of 69.6 among baselines, its performance remains low (0.813), while SPICE attains the largest performance (0.895) boost despite having a relatively large preference margin of 28.7. This confirms that **neither the factuality margin nor the preference margin alone is sufficient and that their gap, captured by alignment potential, is the most informative predictor of model performance.**

### G.13.2 IMPORTANCE OF PREFERENCE MARGIN BEYOND FACTUALITY MARGIN

At first glance, one may question the relative importance of the preference margin because SPICE exhibits a comparatively large preference margin (28.7), the second largest among all baselines. This could suggest that its performance gains are dominated by the factuality margin rather than a balanced contribution of both margins. As we also discuss in Section G.13.1, it is essential to consider both

margins jointly. We therefore provide an additional analysis to clarify why both factors must be taken into account.

To show that its effectiveness is not due to the factuality margin alone, we designed an experiment that examines how the preference margin behaves when the factuality margin is high. We refer to this experiment as "Random Rejection." In this experiment, for each (document,chosen summary) pair used in SPICE, we replace the rejected summary with a randomly selected summary from the dataset, which is unrelated to the source document. This setup satisfies our purpose because the randomly chosen rejection is guaranteed to be highly incorrect for both factuality and preference, pushing both margins to very high values, as shown in Table 27.

Table 28 shows that training DPO on the "Random Rejection" dataset yields the lowest performance, clearly demonstrating that a high factuality margin alone is insufficient. Despite the extremely high factuality margin, the resulting model achieves the lowest performance across all evaluation metrics (FineSurE, G-Eval, AlignScore). This demonstrates that such summary pairs do not constitute a high-quality chosen–rejected set, yield very low alignment potential (6.3–8.4), and ultimately fail to support effective preference optimization. **Therefore, factuality margin alone is not sufficient; both margins are equally essential for achieving high alignment potential.**[v7Lg]

Table 27: Average alignment potentials computed on datasets generated by the Random Rejection setting. "Random-3B" and "Random-8B" indicate that the policy models are Llama3.2-3B and Llama3.1-8B, respectively. Llama3.1-70B is used as the external evaluator to compute $\Delta_{fact}$ and $\Delta_{pref}$, respectively.

| Method | $\Delta_{fact}$ | $\Delta_{pref}$ | $|\Delta_{fact}-\Delta_{pref}|$ |
|---|---|---|---|
| Random-3B | 159.8 | 168.2 | 8.4 |
| Random-8B | 159.8 | 153.5 | 6.3 |

Table 28: Faithfulness comparison of DPO-trained models between random rejection (Random) and SPICE using Llama3.2-3B-Instruct and Llama3.1-8B-Instruct as policy models.

| Method | FineSurE | GEval | AlignScore |
|---|---|---|---|
| Random-3B | $0.702_{(0.00)}$ | $6.984_{(0.00)}$ | $0.698_{(0.00)}$ |
| SPICE-3B | **0.885** | **9.037** | **0.854** |
| Random-8B | $0.805_{(0.00)}$ | $7.561_{(0.00)}$ | $0.747_{(0.00)}$ |
| SPICE-8B | **0.934** | **9.438** | **0.895** |

Table 29: Checkpoints of 4 rejected summary generators, 4 training models, and 3 automated summary evaluators used in our experiments. For open-source models, we use publicly available checkpoints from HuggingFace. For proprietary models, we utilize paid API services provided by OpenAI.

| Model Name | Checkpoints |
|---|---|
| **Rejected Summary Generation** | |
| Bart-Large | facebook/bart-large-cnn |
| Llama3.1-70B-Inst. | meta-llama/Llama3.1-70B-Instruct |
| GPT-4o-mini | gpt-4o-mini-2024-07-18 (OpenAI) |
| Gemma3-12B-Inst. | google/gemma-3-12b-it |
| **Model Training** | |
| Llama3.2-1B-Inst. | meta-llama/Llama3.2-1B-Instruct |
| Llama3.2-3B-Inst. | meta-llama/Llama3.2-3B-Instruct |
| Llama3.1-8B-Inst. | meta-llama/Llama3.1-8B-Instruct |
| Gemma3-1B-Inst. | google/gemma-3-1b-it |
| **Summary Evaluation** | |
| Llama3.3-70B-Inst. | meta-llama/Llama3.3-70B-Instruct |
| GPT-4o-mini | gpt-4o-mini-2024-07-18 (OpenAI) |
| Roberta-Large | yzha/AlignScore-large |

Table 30: **Default prompt** for generating a negative summary.

**User Prompt**

You are given a document and a reference summary.
Your task is to generate a factually inconsistent summary based on the provided document and a reference summary.

**Document:** {Document}
**Reference Summary:** {Reference Summary}

Your answer MUST be in JSON format.
The dictionary key should be hallucinated_summary as a string.

Table 31: Prompt for generating a **no hallucination** summary with similar length.

**User Prompt**

You are given a text. Your task is to paraphrase the text while preserving its original meaning.

Ensure that the generated summary has the same length as the reference summary.

**Text:** {Text}

Your answer MUST be in JSON format.
The dictionary key should be paraphrased_text as a string.

Table 32: Prompt for generating a **mildly hallucinated** summary with similar length.

**User Prompt**

You are given a document and a reference summary.
Your task is to generate a factually inconsistent summary based on the provided document and a reference summary.
Ensure that the hallucinated summary includes factually incorrect information compared to the document and reference summary by following the instruction below.

<Instruction>
1. About 10% of the summary should be edited with hallucinated content.
2. The summary is almost entirely accurate, containing very few hallucinated sentences and almost no errors. A few words should be edited.
3. Do not add additional sentences to generate hallucinations.
4. Ensure that the generated summary has the same length as the reference summary.

**Document:** {Document}
**Reference Summary:** {Reference Summary}

Your answer MUST be in JSON format.
The dictionary key should be hallucinated_summary as a string.

Table 33: Prompt for generating a **severely hallucinated** summary with similar length.

| User Prompt |
| --- |

You are given a document and a reference summary.
Your task is to generate a factually inconsistent summary based on the provided document and a reference summary.
Ensure that the hallucinated summary includes factually incorrect information compared to the document and reference summary by following the instruction below.

<Instruction>
1. About 90-100% of the summary should be edited with hallucinated content.
2. The summary is almost entirely hallucinated, containing very little accurate information and a large number of errors. Nearly all words should be edited.
3. Do not add additional sentences to generate hallucinations.
4. Ensure that the generated summary has the same length as the reference summary.

**Document:** {Document}
**Reference Summary:** {Reference Summary}

Your answer MUST be in JSON format.
The dictionary key should be hallucinated_summary as a string.

Table 34: Prompt for generating a highly hallucinated summary with a **shorter length**.

| User Prompt |
| --- |

You are given a document and a reference summary.
Your task is to generate a factually inconsistent summary based on the provided document and a reference summary.
Ensure that the generated summary is shorter than the reference summary.

**Document:** {Document}
**Reference Summary:** {Reference Summary}

Your answer MUST be in JSON format.
The dictionary key should be hallucinated_summary as a string.

Table 35: Prompt for generating a highly hallucinated summary with **similar length**.

| User Prompt |
| --- |

You are given a document and a reference summary.
Your task is to generate a factually inconsistent summary based on the provided document and a reference summary.
Ensure that the generated summary has the same length as the reference summary.

**Document:** {Document}
**Reference Summary:** {Reference Summary}

Your answer MUST be in JSON format.
The dictionary key should be hallucinated_summary as a string.

Table 36: Prompt for generating a highly hallucinated summary with a **longer length**.

| User Prompt |
|---|
| You are given a document and a reference summary. |

You are given a document and a reference summary.
Your task is to generate a factually inconsistent summary based on the provided document and a reference summary.
Ensure that the generated summary is longer than the reference summary.

**Document:** {Document}
**Reference Summary:** {Reference Summary}

Your answer MUST be in JSON format.
The dictionary key should be hallucinated_summary as a string.

Table 37: **Complex prompt** with multiple constraints for generating a highly hallucinated summary of similar length.

| User Prompt |
|---|

You are given a document and a reference summary.
Your task is to generate a factually inconsistent summary based on the provided document and a reference summary.
Ensure that the hallucinated summary includes factually incorrect information compared to the document and reference summary by following the instructions below.

<Instruction>
1. About 90-100% of the summary should be edited with hallucinated content.
2. The summary is almost entirely hallucinated, containing very little accurate information and a large number of errors. Nearly all words should be edited.
3. Do not add additional sentences to generate hallucinations.
4. Ensure that the generated summary has the same length as the reference summary.
5. Include at least one numerical value that contradicts the document (e.g., wrong year, quantity, or percentage).
6. Replace named entities (people, places, organizations) with incorrect but realistic alternatives.
7. Add contradictory causal relations (e.g., claim X caused Y, even if the document says otherwise).
8. Maintain grammatical fluency and readability so that the hallucinations are not trivially detectable by poor language quality.

**Document:** {Document}
**Reference Summary:** {Reference Summary}

Your answer MUST be in JSON format.
The dictionary key should be hallucinated_summary as a string.

Table 38: Prompt of the FineSurE (Song et al., 2024) for factuality.

**User Prompt**

You will receive a document followed by a corresponding summary.
Your task is to assess the factuality of each summary sentence across nine categories:
* no error: the statement aligns explicitly with the content of the document and is factually consistent with it.
* out-of-context error: the statement contains information not present in the document.
* entity error: the primary arguments (or their attributes) of the predicate are wrong.
* predicate error: the predicate in the summary statement is inconsistent with the document.
* circumstantial error: the additional information (like location or time) specifying the circumstance around a predicate is wrong.
* grammatical error: the grammar of the sentence is so wrong that it becomes meaningless.
* coreference error: a pronoun or reference with wrong or non-existing antecedent.
* linking error: error in how multiple statements are linked together in the discourse (for example temporal ordering or causal link).
* other error: the statement contains any factuality error which is not defined here.

**Instruction:**
First, compare each summary sentence with the document.
Second, provide a single sentence explaining which factuality error the sentence has.
Third, answer the classified error category for each sentence in the summary.
Provide your answer in JSON format. The answer should be a list of dictionaries whose keys are "sentence", "reason", and "category":
["sentence": "first sentence", "reason": "your reason", "category": "no error", "sentence": "second sentence", "reason": "your reason", "category": "out-of-context error", "sentence": "third sentence", "reason": "your reason", "category": "entity error",]

**Document:** {Document}
**Summary with** {# of sentences} **sentences:**
{sentences}

JSON Output:

Table 39: Prompt of the modified G-Eval (Liu et al., 2023) for factuality. We adjusted the score range to 1–5, 1–9, or 1–10 depending on the task. Specifically, a 1–5 scale is used in Figure 1 and Figure 3 for visualization clarity, while Table 1 adopted a 1–9 scale to better separate three levels of quality. In the experimental results (Section 6), we used a 1–10 scale for consistency with other evaluation scores on a tens-place scale.

**User Prompt**

You will then be given one summary written for the source text. Your task is to rate the summary on one metric. Please make sure you read and understand these instructions carefully. Please keep this document open while reviewing, and refer to it as needed.

Evaluation Criteria:

Factual Consistency (1-10): The factual alignment between the summary and the source text. A factually consistent summary contains only statements that are entailed by the source text.

Evaluation Steps:
1. Read the article carefully and identify the main facts and details it presents.
2. Read the summary and compare it to the source text. Check if the summary contains any factual errors that are not supported by the source text.
3. Assign a score for factual consistency on a scale of 1 to 10, where 1 is the lowest and 10 is the highest based on the Evaluation Criteria.

**Source Text:** {Source Text}
**Summary:** {Summary}

Factual Consistency (scores ONLY):

Table 40: Prompt of the modified G-Eval (Liu et al., 2023) for coherency. We adjusted the score range to 1–10.

**User Prompt**

You will be given one summary written for an article. Your task is to rate the summary on one metric. Please make sure you read and understand these instructions carefully. Please keep this document open while reviewing, and refer to it as needed.

Evaluation Criteria:

Coherence (1–10) – the collective quality of all sentences. We align this dimension with the DUC quality question of structure and coherence whereby "the summary should be well-structured and well-organized. The summary should not just be a heap of related information, but should build from sentence to a coherent body of information about a topic."

Evaluation Steps:
1. Read the article carefully and identify the main topic and key points.
2. Read the summary and compare it to the article. Check if the summary covers the main topic and key points of the article, and if it presents them in a clear and logical order.
3. Assign a score for coherence on a scale of 1 to 10, where 1 is the lowest and 10 is the highest based on the Evaluation Criteria.

Example:

Source Text:{Document}

Summary:{Summary}

Coherency (scores ONLY):

Table 41: Prompt of the modified G-Eval (Liu et al., 2023) for relevance. We adjusted the score range to 1–10.

**User Prompt**

You will be given one summary written for an article. Your task is to rate the summary on one metric. Please make sure you read and understand these instructions carefully. Please keep this document open while reviewing, and refer to it as needed.

Evaluation Criteria:

Relevance (1–10) – selection of important content from the source. The summary should include only important information from the source document. Annotators were instructed to penalize summaries which contained redundancies and excess information.

Evaluation Steps:
1. Read the summary and the source document carefully.
2. Compare the summary to the source document and identify the main points of the article.
3. Assess how well the summary covers the main points of the article, and how much irrelevant or redundant information it contains.
4. Assign a relevance score from 1 to 10.

Example:

Source Text:{Document}

Summary:{Summary}

Relevance (scores ONLY):

Table 42: Prompt for generating a negative summary based on the document only.

**User Prompt**

You are given a document.
Your task is to generate a factually inconsistent summary based on the provided document.

**Document:** {Document}

Your answer MUST be in JSON format.
The dictionary key should be hallucinated_summary as a string.

Table 43: Prompt for generating a negative summary based on the reference summary only.

**User Prompt**

You are given a reference summary.
Your task is to generate a factually inconsistent summary based on the provided reference summary.
Ensure that the generated summary has the same length as the reference summary.

**Reference Summary:** {Reference Summary}

Your answer MUST be in JSON format.
The dictionary key should be hallucinated_summary as a string.

Table 44: Modified SYNFAC-EDIT (Mishra et al., 2024) prompt for generating rejected summaries, adapted from its original use in clinical summarization to be applicable across diverse summarization domains. The core logic and structure are preserved with minimal modifications.

**User Prompt**

»»»» **Instruction** »»»»
You are a writing assistant who is in edit mode. You are tasked with generating a hallucinated summary based on the provided article and a reference summary for the article.
The goal is to edit the reference summary to generate a hallucinated summary that sounds plausible but includes edits introduced through an edit operation, which can be one of the following:

**Add Operation:** Intentionally add contextually important words from the source that are not necessary for a basic understanding but carry implications for accurate interpretation or decision-making.
**Omit Operation:** Intentionally omit contextually important words in the reference summary that are necessary for accurate interpretation or decision-making.

For these operations, focus on words that, if missing or incorrect in the hallucinated summary, could lead to significant misunderstandings or incorrect conclusions. Maintain coherence while deliberately excluding essential terms. The hallucinated summary should be concise, contain no more than five extra words compared to the reference summary, and have an equal number of Add and Omit operations.

Steps for generating the hallucinated summary:
**Step 1:** List the proposed edit operations to introduce hallucination on the reference summary.
**Step 2:** Use the proposed edit operations to edit the reference summary.

»»»» **Output Format** »»»»
The output format is:
Numbererd List hallucination edits made:
Edit 1, Edit 2, Edit 3 ...
Hallucinated Summary:

»»»» **Follow the above Instructions, Hallucination Method and Output Format** »»»»
Now, let's start.
Generate the hallucinated summary:
**Article:** {Article}
**Reference Summary:** {Reference Summary}

Table 45: An example of an input prompt and its corresponding chosen and rejected summaries used for DPO training.

| | |
|---|---|
| **Input** | <\|begin_of_text\|><\|start_header_id\|>user<\|end_header_id\|> 

 Below is an instruction that describes a task. Write a response that appropriately completes the request. 

 ### Instruction: 
 Please factually summarize the input document. 

 ### Input: 
 (CNN) – On Tuesday, new Apple CEO Tim Cook is expected to help unveil Apple's latest iPhone at its first big event since co-founder Steve Jobs stepped aside in August. Cook will no doubt be scrutinized for how he fulfills his new role Tuesday as chief pitchman for Apple's products. And the new iPhone, expected to go on sale sometime in mid-October, will almost certainly draw lines of shoppers outside Apple's stores. Then the question becomes: Can Apple make enough iPhones fast enough? As demand for its gadgets continues to skyrocket, keeping Apple growing at its amazing pace will be a key hurdle for Cook's regime. The consumer shift toward cheaper, portable computers – and Apple's huge success there with the iPhone and iPad – means creating and selling an ever-increasing number of devices each year. Right now, for example, Apple will have to ramp up production of millions of new iPhones, and possibly new iPods, for the holidays. After that there will be more iPads, Macs, and other products. It's important that Apple's new devices be stylish and technically impressive. But it's equally important that they be in stock when people want to buy them, something that Apple hasn't always been able to deliver. Only recently, for example, has the iPad 2 supply been able to meet demand. And as time goes on, thanks in part to new markets such as China, demand for each gadget will only grow. It's not enough for Cook and his team to keep coming up with exciting new products. They also must continue Apple's logistics revolution, so that consumers don't have to wait forever to buy new Apple devices – or turn instead to the competition. Consider the total number of devices Apple must now have built and shipped per year. In 2011, including all iPhones, iPods, iPads, and Macs, Apple could end up shipping more than 175 million gadgets – some 50 million more than it did last year, representing roughly 40% growth. Next year, Apple is expected to ship an all-time high of around 215 million devices, or another 40 million more than this year. And so on. It wouldn't be insane for Apple to have to ship 500 million units per year by the end of the decade. So Cook will have to figure out how to become the type of company that can ship 500 million gadgets per year – from staffing to product design to iCloud server infrastructure to components to manufacturing, sales and support. The good news: Cook is already, arguably, the best in the business when it comes to this stuff. Over the past decade, he's made Apple's supply chain an industry leader. For Cook and Apple's SVP of Operations Jeff Williams, the plan is to keep it that way. (See Fortune Magazine's recent profile of Williams here.) As Apple enters its next chapter, some of Cook's biggest practical challenges will include scaling production, deciding how to prioritize the company's existing businesses and figuring out which new businesses to enter – such as making TVs. (Then there's the whole ¨vision thing,¨which is a whole different challenge – and where Jobs, the master innovator, may prove to be irreplaceable.) Later this month, Cook will make his first pitch to Wall Street as Apple's CEO, as Apple reports its quarterly results. A year from now, we'll be in good position to judge his performance at the helm of the world's most valuable tech company. But first, it's time for a new iPhone. Copyright © 2011 Business Insider. 

 ### Response: <\|eot_id\|> |
| **Output (Chosen)** | <\|start_header_id\|>assistant<\|end_header_id\|> Apple's new CEO Tim Cook is expected to help unveil Apple's latest iPhone. Cook will no doubt be scrutinized for how he fulfills his new role Tuesday. As demand for Apple's gadgets continues to skyrocket, keeping Apple growing at its amazing pace will be a key hurdle for Cook's regime. <\|eot_id\|> |
| **Output (Rejected)** | <\|start_header_id\|>assistant<\|end_header_id\|> Apple's new CEO Tim Cook is expected to discontinue the iPhone and focus on producing only iPads. Cook will no doubt be praised for how he fulfills his new role Tuesday. As demand for Apple's gadgets continues to decline, keeping Apple growing at its amazing pace will be a key hurdle for Cook's regime. <\|eot_id\|> |

Table 46: Comparison of chosen and rejected summaries on a dialogue document, showing factuality, preference, and delta scores across baselines. Factuality is measured by GPT-4o-mini using the G-Eval scale (1–10), and preference (log probability) is measured by Llama3.2-3B-Instruct.

| | |
|---|---|
| Document | #Person1#: May I take your drink order while you are looking over your menu?
#Person2#: Yes, do you have a wine list?
#Person1#: The wine list is on the second page of your menu.
#Person2#: Do you have mixed drinks in this restaurant?
#Person1#: Yes, we have a full bar here.
#Person2#: I am not sure what I want. Do you have any house specials?
#Person1#: Actually, we are famous for our Cuervo Gold margaritas.
#Person2#: That sounds good! Please bring me one of those.
#Person1#: Would you like that drink blended or on the rocks?
#Person2#: I would like it blended.
#Person1#: Would you like it with salt or no salt?
#Person2#: I would like my margarita with no salt, thank you. |

| Chosen | A customer is ordering a drink at a restaurant. They ask about the wine list and mixed drinks, and the server recommends the house special, a Cuervo Gold margarita. The customer orders a blended margarita with no salt. | |
|---|---|---|
| | factuality : 10 | preference : -55.3 |

| Rejected (CLIFF-Entity) | A customer is ordering a drink at a restaurant. They ask about the wine list and mixed drinks, and the server recommends the house special, a Greenpeace margarita. The customer orders a blended margarita with no salt. | | | |
|---|---|---|---|---|
| | factuality : 3 | $\Delta_{\text{fact}}$ : 7 | preference : -79.5 | $\Delta_{\text{pref-3b}}$ : 24.2 |

| Rejected (CLIFF-Mask) | A selection of the best margaritas from around the world have been sent to Newsround in the US. | | | |
|---|---|---|---|---|
| | factuality : 1 | $\Delta_{\text{fact}}$ : 9 | preference : -99.0 | $\Delta_{\text{pref-3b}}$ : 43.7 |

| Rejected (SummLlama) | #Person1#: Would you like that drink blended or on the rocks?
#Person1#: Would you like it with salt or no salt? | | | |
|---|---|---|---|---|
| | factuality : 8 | $\Delta_{\text{fact}}$ : 2 | preference : -57.5 | $\Delta_{\text{pref-3b}}$ : 2.2 |

| Rejected (SYNFAC-EDIT) | The customer looks over the menu and asks first about wine and then mixed drinks. The server mentions their popular house special, the Cuervo Gold margarita. After a brief hesitation, the customer decides to try it and asks for it blended with salt. | | | |
|---|---|---|---|---|
| | factuality : 10 | $\Delta_{\text{fact}}$ : 0 | preference : -77.5 | $\Delta_{\text{pref-3b}}$ : 22.2 |

| Rejected (SPICE) | A customer is ordering a drink at a restaurant. They ask about the wine list and mixed drinks, and the server recommends the house special, a Cuervo Gold margarita. The customer orders a margarita with salt. | | | |
|---|---|---|---|---|
| | factuality : 2 | $\Delta_{\text{fact}}$ : 8 | preference : -66.0 | $\Delta_{\text{pref-3b}}$ : 10.7 |

Table 47: Comparison of chosen and rejected summaries on a non-dialogue document, showing factuality, preference, and delta scores across baselines. Factuality is measured by GPT-4o-mini using the G-Eval scale (1–10), and preference (log probability) is measured by Llama3.2-3B-Instruct.

| | |
|---|---|
| Document | By . Amanda Williams . PUBLISHED: . 22:19 EST, 9 April 2013 . | . UPDATED: . 04:03 EST, 10 April 2013 . Toni Pearce is the first National Union of Students (NUS) President to not have attended university . The National Union of Students has elected its first president to not have attended university. Toni Pearce, currently the union's vice president for further education, studied at Cornwall College, but never went on to a university education because taking three years 'out' to do so 'was not attractive to her'. She was elected to serve as the NUS's new president at the union's annual conference in Sheffield on Tuesday. Miss Pearce, 22, is currently serving a second term as the union's vice president for further education. In her current NUS role she has claimed Britain is 'run by Oxbridge' and admitted she 'can't stand' Education Secretary Michael Gove, The Daily Telegraph reports. Miss Pearce, who describes herself as a feminist, was brought up in Camborne, Cornwall, by her parents, an ex-Navy aircraft engineer and a chartered management accountant. She won the election in the first round with 424 of 732 valid votes. Speaking afterwards she said: 'I'm really proud to have been given the opportunity to build the student movement around a vision for public education, and to be leading NUS as we build towards the next general election. 'Between now and 2015 we need to hold a full and frank debate about what education means to society and to properly articulate the public value of education in communities up and down the country.' More than 450 of the student unions . affiliated with the NUS are in further education (FE) colleges, the . union said, and most of the students it represents are studying in FE . institutions. She was elected at NUS at its annual conference in Sheffield . |

| Chosen | Toni Pearce, 22, studied at Cornwall College but never went on to university. She was elected to serve as the NUS's new president at the union's annual conference in Sheffield on Tuesday. | | |
|---|---|---|---|
| | factuality : 9 | | preference : -49.0 | |

| Rejected (CLIFF-Entity) | Alex Johnson, 22, studied at Cornwall College but never went on to university. She was elected to serve as the NUS's new president at the union's annual conference in Sheffield on Tuesday. | | | |
|---|---|---|---|---|
| | factuality : 4 | $\Delta_{fact}$ : 5 | preference : -62.0 | $\Delta_{pref-3b}$ : 13.0 |

| Rejected (CLIFF-Mask) | A woman from Cornwall has been elected as the new president of the Association of Metal Workers. | | | |
|---|---|---|---|---|
| | factuality : 1 | $\Delta_{fact}$ : 8 | preference : -101.5 | $\Delta_{pref-3b}$ : 52.5 |

| Rejected (SummLlama) | The first National Union of Students President, Toni Pearce, did not attend university as taking a year out was unattractive to her. She has claimed that Britain is influenced by Oxbridge and has a dislike for Education Secretary Michael Gove. Pearce was elected at the NUS Conference in Sheffield and serves as Vice-President for Further Education. She is the first woman to join the National Union of Students President role without a university background. | | | |
|---|---|---|---|---|
| | factuality : 7 | $\Delta_{fact}$ : 2 | preference : -159.0 | $\Delta_{pref-3b}$ : 110.0 |

| Rejected (SYNFAC-EDIT) | Toni Pearce, 22, studied at Cornwall College but never went on to university. She was elected as the first NUS President without a university degree to serve as the NUS's new president. | | | |
|---|---|---|---|---|
| | factuality : 8 | $\Delta_{fact}$ : 1 | preference : -58.3 | $\Delta_{pref-3b}$ : 9.3 |

| Rejected (SPICE) | Toni Pearce, 25, studied at Oxford University but never attended college. She was elected to serve as the NUS's new president at the union's annual conference in London on Monday. | | | |
|---|---|---|---|---|
| | factuality : 1 | $\Delta_{fact}$ : 8 | preference : -81.0 | $\Delta_{pref-3b}$ : 32.0 |

