# OpenReview forum: "Quality of Rejections Matters: Preference Data Construction for Faithful Summarization"
_ICLR.cc/2026/Conference — Submitted to ICLR 2026_

### Official Review · Reviewer_v7Lg · 2025-10-26

**Soundness:** 2
**Presentation:** 3
**Contribution:** 2
**Rating:** 2
**Confidence:** 4

**Summary:**

This paper investigates the construction of preference data for improving factual consistency in abstractive summarization. The authors introduce the concept of "alignment potential," arguing that high-quality preference pairs require a large "Factuality Margin" (a clear factual difference between chosen and rejected summaries) and a small "Preference Margin" (the rejected summary is difficult for the base model to distinguish from the chosen one).

Through theoretical analysis and controlled empirical studies, the paper identifies three key factors influencing these margins: hallucination level, summary length, and prompt complexity. Based on these insights, the authors propose SPICE (Simple Prompt for Informative and Contrastive Examples), a prompting strategy that generates rejected summaries with high hallucination levels, matched lengths, and minimal instructions.

Experiments across various model scales (1B-8B) and alignment algorithms (DPO, SimPO, KTO) demonstrate that SPICE empirically outperforms existing reference-based (CLIFF) and model-based (SummLlama, SYNFAC-EDIT) strategies in improving faithfulness.

**Strengths:**

S1: **Systematic Empirical Analysis**: Section 4 provides a controlled analysis that systematically investigates the impact of hallucination level, length, and prompt complexity on the defined margins. The methodology of this analysis is rigorous.

S2: **Empirical Performance**: The proposed method, SPICE, demonstrates strong empirical performance across diverse datasets, model scales, and alignment algorithms. The results reported in Table 4 show consistent improvements over the baselines used.

S3: **Practicality and Simplicity**: The resulting method (SPICE) is simple and easy to implement, suggesting practical utility.

**Weaknesses:**

W1: **Fundamental Inconsistency Between Framework and Results (Preference Margin)**: This is the most critical weakness. The paper's core theoretical framework argues that minimizing the preference margin (making the rejection "harder" to distinguish) is crucial for high alignment potential. However, the empirical data in Table 5 contradicts this. SPICE's preference margin (e.g., 28.7 for 3B) is substantially larger (easier to distinguish) than that of SummLlama (16.5). This suggests that the framework, as proposed, does not accurately explain the results, and the performance gains may be driven entirely by the large factuality margin, undermining the central thesis of the paper.

W2: **Weakness of Baselines (SYNFAC-EDIT)**: The comparison against baselines appears inflated by weak competition. Most notably, the comparison against SYNFAC-EDIT (the most similar LLM-based editing approach) is highly suspect. Table 5 shows SYNFAC-EDIT achieved an average factuality margin (delta_fact) of only 0.15, compared to SPICE's 7.07. This indicates that the SYNFAC-EDIT implementation fundamentally failed to generate meaningful factual errors, rendering it an ineffective comparison.

W3: **Limited Conceptual Novelty**: The concept of "alignment potential"—requiring negative examples that are difficult (small preference margin) yet clearly incorrect (large factuality margin)—is essentially a rebranding of the well-established principle of Hard Negative Mining. The paper formalizes common knowledge in the context of LLM faithfulness without offering significant technical or conceptual innovation.

*Overall*: While the paper addresses an important topic and presents a systematic empirical study (S1) with positive results (S2), I recommend rejection due to fundamental inconsistencies in the core argument and limited novelty. The primary reason for rejection is W1. The paper introduces the framework of "alignment potential," which hinges on minimizing the preference margin, but the empirical data in Table 5 directly contradicts this requirement. Furthermore, the conceptual novelty of the work is marginal (W3), as it largely formalizes the intuition behind Hard Negative Mining. Finally, the empirical success must be contextualized by the apparent failure of a key baseline (W2). The fact that SYNFAC-EDIT generated almost no factual errors invalidates the comparison and makes it difficult to assess the true significance of the empirical gains.

**Questions:**

Q1 (Regarding W1): In Table 5, SPICE exhibits a significantly higher (easier to distinguish) preference margin than SummLlama (e.g., 28.7 vs 16.5 for the 3B model). How do you reconcile this observation with the paper's core argument that minimizing the preference margin is necessary for high alignment potential? Is it possible that the success of SPICE is driven solely by the very large factuality margin, making the preference margin less relevant than argued?

Q2 (Regarding W2): Given the extremely low  (0.15) for SYNFAC-EDIT and the discussion in Appendix D suggesting the prompt failed to induce sufficient hallucinations, were attempts made to revise the SYNFAC-EDIT prompt to ensure it generated a comparable level of factual errors to SPICE before conducting the main experiments?

Q3 (Regarding W3): How do the authors specifically differentiate their concept of "alignment potential" and the subsequent insights from the long-established principle of hard negative mining in contrastive learning?

---

> ### Author Response · Authors · 2025-11-25
> **Response to Reviewer v7Lg (1/4)**
>
> We appreciate your valuable comments and suggestions. We hope that the concerns can be resolved through our clarifications in this response, and all discussed revisions will be incorporated into the updated manuscript.
>
> ---
> > **Weakness 1.** *Fundamental Inconsistency Between Framework and Results (Preference Margin): This is the most critical weakness. The paper's core theoretical framework argues that minimizing the preference margin (making the rejection "harder" to distinguish) is crucial for high alignment potential. However, the empirical data in Table 5 contradicts this. SPICE's preference margin (e.g., 28.7 for 3B) is substantially larger (easier to distinguish) than that of SummLlama (16.5). This suggests that the framework, as proposed, does not accurately explain the results, and the performance gains may be driven entirely by the large factuality margin, undermining the central thesis of the paper.*
>
> We acknowledge that the ambiguity in our Introduction and Section 6.1.2 may have led to a misunderstanding regarding our core claim.
>
> First, we made the mistake of unintentionally presenting the goals of "maximizing the factuality margin ($\Delta_{fact}$)" and "minimizing the preference margin ($\Delta_{pref}$)" as an isolated objective, which may give the impression that the theory is inconsistent with the results in Table 5. Our intended message is that alignment potential depends on both margins jointly, and specifically on achieving a "large gap" between the factuality margin and the preference margin ($\vert\Delta_{fact}-\Delta_{pref}\vert$).
>
> Second, we made an error in how the two margins were computed in Table 5. Specifically, the factuality margin was derived from G-Eval scores, while the preference margin was computed from log probabilities. Because these two quantities lie on incompatible numerical scales, this made the margin discrepancy ($\Delta_{fact} - \Delta_{pref}$) theoretically invalid [1, 2]. As clarified in Section 3.1 of our paper, once the preference margin is defined through log probabilities, the factuality margin should also be computed using log probabilities from an external reward model to be meaningful. We acknowledge that the mismatched computation in the initial version was our mistake.
>
> To correct this, we recomputed the factuality margin using log probabilities from Llama3.1-70B (external reward model), ensuring that both margins share a consistent scale. Under this corrected formulation, SPICE exhibits the largest alignment potential among all baselines, as shown in the two updated tables. Therefore, our intended claim is validated: "SPICE achieves the greatest margin discrepancy that leads to the highest alignment potential," rather than "simply having the largest factuality margin and a small preference margin."
>
> (External Evaluator: Llama3.1-70B, Target Policy: Llama3.2-3B)
> | Method | $\Delta_{fact}$ | $\Delta_{pref}$|$\vert\Delta_{fact}-\Delta_{pref}\vert$|
> |-----------------|:-------:|:------:|:-----:|
> | `CLIFF-Entity`  |   47.6  |  28.5  | 19.1  |
> | `CLIFF-Mask`    |   69.6  |  41.9  | 27.7  |
> | `SYMFAC-EDIT`   |   45.2  |  16.3  | 28.9  |
> | `SummLlama`     |   56.7  |  16.5  | 40.2  |
> | `SPICE (Ours)`  |   102.0 |  28.7  | **74.3**  |
>
> (External Evaluator: Llama3.1-70B, Target Policy: Llama3.1-8B)
> | Method | $\Delta_{fact}$ | $\Delta_{pref}$|$\vert\Delta_{fact}-\Delta_{pref}\vert$|
> |-----------------|:-------:|:------:|:-----:|
> | `CLIFF-Entity`  |   47.6  |  29.6  | 18.0  |
> | `CLIFF-Mask`    |   69.6  |  48.3  | 21.3  |
> | `SYMFAC-EDIT`   |   45.2  |  19.3  | 25.9  |
> | `SummLlama`     |   56.7  |  16.4  | 40.3  |
> | `SPICE (Ours)`  |   102.0 |  32.1  | **69.9**  |
>
> We will revise the Introduction and Section 6.1.2 to reflect this corrected claims. We will emphasize that alignment potential depends on both margins jointly, specifically defined by the gap between them.
>
> [1] A General Theoretical Paradigm to Understand Learning from Human Preferences, In ICML, 2024.
>
> [2] The Crucial Role of Samplers in Online Direct Preference Optimization, In ICLR, 2025.
>
> ---

---

> ### Author Response · Authors · 2025-11-25
> **Response to Reviewer v7Lg (2/4)**
>
> > ***Question 1.** How do you reconcile this observation with the paper's core argument that minimizing the preference margin is necessary for high alignment potential? Is it possible that the success of SPICE is driven solely by the very large factuality margin, making the preference margin less relevant than argued?*
>
> Thank you for the thoughtful question. We believe the first part has already been addressed in our response to W1, so we focus here on the second part, which raises the concern that SPICE’s success might be driven solely by a large factuality margin.
>
> To show that its effectiveness is not due to the factuality margin alone, we designed an experiment that examines how the preference margin behaves when the factuality margin is high. We refer to this experiment as “Random Rejection.” In this experiment, for each (document,chosen summary) pair used in SPICE, we replace the rejected summary with a randomly selected summary from the dataset, which is unrelated to the source document. This setup satisfies our purpose because the randomly chosen rejection is guaranteed to be highly incorrect for both factuality and preference, pushing both margins to very high values, as shown below for the two policy models.
>
> (External Evaluator: Llama3.1-70B-Instruct, Policy: Llama3.2-3B-Instruct)
> | Method | $\Delta_{fact}$ | $\Delta_{pref}$|$\vert\Delta_{fact}-\Delta_{pref}\vert$|
> |-----------------|:-------:|:------:|:-----:|
> | `Random rejection`  |   159.8  |  168.2  | 8.4  |
>
> (External Evaluator: Llama3.1-70B-Instruct, Policy: Llama3.1-8B-Instruct)
> | Method | $\Delta_{fact}$ | $\Delta_{pref}$|$\vert\Delta_{fact}-\Delta_{pref}\vert$|
> |-----------------|:-------:|:------:|:-----:|
> | `Random rejection`  |   159.8  |  153.5  | 6.3  |
>
> Crucially, when we train DPO using this "Random Rejection" dataset, the results clearly show that a high factuality margin alone is insufficient. Despite the extremely high factuality margin, the resulting model achieves the lowest performance across all evaluation metrics (FineSurE, G-Eval, AlignScore). This demonstrates that such summary pairs do not constitute a high-quality chosen–rejected set, yield very low alignment potential (6.3--8.4), and ultimately fail to support effective preference optimization. Therefore, factuality margin alone is not sufficient; both margins are equally essential for achieving high alignment potential.
>
> (Backbone for DPO: Llama3.2-3B-Instruct)
> | Method | FineSurE | G-Eval | AlignScore |
> |-----------------|:-------:|:------:|:-----:|
> | `Random rejection`  | 0.702(0.00) | 6.984(0.00) | 0.698(0.00) |
> | `SPICE`  | 0.885 | 9.037 | 0.854 |
>
> (Backbone for DPO: Llama3.1-8B-Instruct)
> | Method | FineSurE | G-Eval | AlignScore |
> |-----------------|:-------:|:------:|:-----:|
> | `Random rejection`  | 0.805(0.00) | 7.561(0.00) | 0.747(0.00) |
> | `SPICE`  | 0.934 | 9.438 | 0.895 |
>
> ---

---

> ### Author Response · Authors · 2025-11-25
> **Response to Reviewer v7Lg (3/4)**
>
> > **Weakness 2.** *Weakness of Baselines (SYNFAC-EDIT): The comparison against baselines appears inflated by weak competition. Most notably, the comparison against SYNFAC-EDIT is highly suspect. Table 5 shows SYNFAC-EDIT achieved an average factuality margin of only 0.15, compared to SPICE's 7.07. This indicates that the SYNFAC-EDIT implementation fundamentally failed to generate meaningful factual errors, rendering it an ineffective comparison.*
>
> >> ***Question 2.** Given the extremely low (0.15) for SYNFAC-EDIT and the discussion in Appendix D suggesting the prompt failed to induce sufficient hallucinations, were attempts made to revise the SYNFAC-EDIT prompt to ensure it generated a comparable level of factual errors to SPICE before conducting the main experiments?*
>
> We appreciate your thoughtful concern. We approached this part of the study with great care and made effort to verify it thoroughly, including repeated checks and faithful reproduction. In what follows, we summarize the key evidence demonstrating that our comparison is fair and not inflated by a weak baseline.
>
> **A. We faithfully reproduced and analyzed SYNFAC-EDIT implementation**
>
> *(1) SYNFAC-EDIT’s Inherently Small Faithfulness Margin*: To ensure our findings were not implementation artifacts, we analyzed the original SYNFAC-EDIT data and found that their chosen–rejected pairs already exhibit very small faithfulness margins, indicating the method inherently produces minimal factual divergence. Our reproduction showed the same pattern, and this behavior remained consistent under our experimental setup. In our reproduction, we sampled 1,000 examples from each dataset and computed the factuality margin using two approaches: (1) prompting an external model for factuality scores (G-Eval with GPT-4o-mini) and (2) computing the log-probability margin using an external model (LLaMA 3.1-70B), as in Lines 165–168 of the manuscript.
>
> | Dataset | Rouge1 | RougeL | $\Delta_{fact}$ (G-Eval) | $\Delta_{fact}$ (Log-prob.) |
> |-----------------|:-------:|:------:|:-----:|:-----:|
> | `Original`  | 0.883 | 0.856 | 0.36 | 49.3 |
> | `Ours`  | 0.908 | 0.882 | 0.15 | 45.2 |
>
> This convergence across both the original and reproduced data strongly indicates that the small faithfulness margin is an inherent characteristic of SYNFAC-EDIT, rather than a flaw in our implementation.
>
> *(2) Low Factuality Margin from Edit-Based Mechanism:* The small factuality margin is a direct consequence of SYNFAC-EDIT’s edit-based design. The method primarily performs localized rejection edits, such as adding, deleting, or swapping words, which inherently limit the degree of semantic deviation it can introduce. As also supported by Appendix D and prior works [5, 6, 7], such operations indeed rarely produce the substantial meaning shifts required to generate informative hallucinations. Thus, the low factuality margin reflects a fundamental limitation of the mechanism itself.
>
> *(3) Prompt Revisions are Insufficient:* Although we explored prompt variations within the intended edit-based paradigm (e.g., varying the number and balance of Add/Omit operations, relaxing edit constraints, adjusting the allowed word budget), the factuality margin remained low. Meaningful improvement would require abandoning the edit-based mechanism or introducing overly complex prompt changes that go beyond a revision and amount to creating a new method. Within reasonable revisions, we observed no improvement.
>
> **B. Our experiment results reinforces the limitation of SYNFAC-EDIT**
>
> First, SYNFAC-EDIT adopts an entity–swap–based rejection strategy similar to CLIFF-Entity. As shown in Table 5, both methods consistently produce very small factuality margins, which directly leads to their weaker performance in Table 4. Second, SYNFAC-EDIT is a complex prompting strategy due to its extensive instructions (Table 32). Although its performance is limited, it serves as an essential baseline that contrasts with our simple prompting approach. These observations highlight the key challenge is not the quantity of rejected summaries, but the ability to generate high-quality rejections.
>
> **C. SummLlama is the Most Recent and Strongest Baseline**
>
> SummLlama is the most recent method and has already demonstrated superior performance over SYNFAC-EDIT within the paper. Therefore, it serves as a strong baseline for comparison, and outperforming SummLlama is a meaningful indicator of effectiveness. In all our experiments, we compare directly against SummLlama.
>
> We will incorporate these analysis into the revised manuscript to fully address the reviewer’s concern.
>
> [5] CONFIT: Toward Faithful Dialogue Summarization with Linguistically-Informed Contrastive Fine-tuning, In NAACL, 2022.
>
> [6] AMRFact: Enhancing Summarization Factuality Evaluation with AMR-Driven Negative Samples Generation, In NAACL, 2024.
>
> [7] Enhancing Faithfulness in Abstractive Summarization via Span-Level Fine-Tuning, In ArXiv, 2025.
>
> ---

---

> ### Author Response · Authors · 2025-11-25
> **Response to Reviewer v7Lg (4/4)**
>
> > **Weakness 3.** *Limited Conceptual Novelty: The concept of "alignment potential"—requiring negative examples that are difficult (small preference margin) yet clearly incorrect (large factuality margin)—is essentially a rebranding of the well-established principle of Hard Negative Mining. The paper formalizes common knowledge in the context of LLM faithfulness without offering significant technical or conceptual innovation.*
>
> While the intuition that "difficult yet informative negatives provide stronger training signals" is widely shared, we note that this intuition alone has not typically been used to characterize conceptual novelty. Prior work across contrastive learning, metric learning, summarization, and others suggests that the process of formalizing what constitutes difficulty and informativeness, rather than simply acknowledging the idea, has often been regarded as a meaningful contribution. For example, [8], [9], and [10] each offer different task-specific interpretations of hard negatives, indicating that translating this intuition into clear operational criteria has been recognized as valuable in the literature. In particular, [8] examines negatives that are highly similar yet incorrect, [9] defines hard negatives as summaries that remain close to positives despite being wrong, and [10] introduces factual perturbations as hard negatives for dialogue summarization.
>
> Building on this understanding, our work aims to contribute by moving beyond heuristic or task-specific notions of hard negatives, particularly in the context of faithful summarization. We introduce a margin-based formulation in which difficulty is represented through a preference margin and incorrectness through a factuality margin, providing two complementary, model-measurable quantities tailored to this setting. These margins allow us to characterize the properties that make certain rejected summaries genuinely useful for improving faithfulness, and they further enable us to examine how these properties relate to alignment potential in faithful summarization. Through this framework, we show that the effectiveness of rejected summaries in faithful summarization depends not on their number but on their quality, as reflected in their preference and factuality margins.
>
> Overall, our formulation is not a simple rebranding of hard negative mining, but a structured framework that makes the intuition operational in faithful summarization and offers explanatory value.
>
> [8] Improving Text Embedding Models with Positive-aware Hard-negative Mining, CIKM 2025
>
> [9] Trainable hard negative examples in contrastive learning for unsupervised abstractive summarization, EACL 2024
>
> [10] CONFIT: Toward faithful dialogue summarization with linguistically-informed contrastive fine-tuning, NAACL 2022
>
>
> ---
>
> > ***Question 3.** How do the authors specifically differentiate their concept of "alignment potential" and the subsequent insights from the long-established principle of hard negative mining in contrastive learning?*
>
> Although alignment potential is conceptually related to hard negative mining, our formulation differs from the hard negative mining principle in contrastive learning in several important ways.
>
> In both supervised and unsupervised contrastive learning, negative labels are determined or assumed in advance by the framework: labels explicitly define negatives in supervised settings, and once an anchor–positive pair is selected in unsupervised settings, all remaining samples are implicitly treated as negatives. Hard negative mining then evaluates difficulty of discrimination solely through embedding distance, measuring only how confusing a negative is for the model. Importantly, hard negative mining does not consider whether the negative label itself is correct; informativeness is assumed purely from difficulty. In other words, hard negatives in contrastive learning are informative only under the assumption that the example truly belongs to the negative class.
>
> In contrast, alignment potential evaluates two signals simultaneously: the factuality margin, which captures how clearly incorrect the rejected is (informativeness), and the preference margin, which reflects how non-trivial it is for the model to distinguish it from the chosen (difficulty of discrimination). A rejected example is informative only when these two signals interact appropriately. For example, a hard negative (very small preference margin) with a very small factuality margin may remain uninformative because it lacks meaningful errors.
>
> Therefore, the key difference is that while contrastive learning focuses exclusively on the difficulty of discrimination, alignment potential quantitatively considers both the informative and the difficulty of a discrimination simultaneously.
>
>
> ---
>
> If there are any aspects of our response that remain unclear or could benefit from further clarification, we would be more than happy to address them.
>
> ---

---

### Official Review · Reviewer_NXZW · 2025-10-30

**Soundness:** 2
**Presentation:** 3
**Contribution:** 2
**Rating:** 4
**Confidence:** 4

**Summary:**

This paper examines the factors that render rejected summaries effective for faithfulness alignment in text summarization. The authors argue that effective rejected summaries should possess high "alignment potential," which is defined by a narrow preference margin (indicating that distinguishing between options is non-trivial) and a wide factuality margin (representing a clear contrast in factual accuracy). They identify three key influencing factors hallucination level, summary length, and prompt complexity. Proposing SPICE, a straightforward prompting strategy. Although the study demonstrates solid methodology and comprehensive analysis, its innovative contribution is relatively incremental.

**Strengths:**

- The paper clearly formalizes the gap between preference margin and factuality margin as "alignment potential", providing a principled lens on why some rejections are more useful than others. Meanwhile, the author propose two quantifiable metrics: preference margin and factuality margin
- Beyond the concept, the paper also proposes SPICE, a simple yet highly effective prompting strategy that maximizes alighment potential by enforcing high hallucination, matched length, and prompt complexity
- The paper demonstrate a comprehensive and robust experiments with wide range of baselines. The proposed SPICE, consistently outperforms the model-based and reference-based method across various model scales and alignment. Highlighting the effectiveness of proposed method

**Weaknesses:**

- External evaluators (G-Eval/FineSurE/AlignScore) are potentially coupled with training signals, lacking robustness ablation studies such as leave-one-evaluator-out and replacement of evaluator families, as well as larger-scale, hierarchical human evaluation comparisons
- One concerns is whether there should be more ablation studies. For example, a comparison between data-level length control (equal-length vs no control) against loss-level normalization (e.g., SimPO/KTO variants)

**Questions:**

Q1: Why not implement actual adversarial sampling based on your theoretical analysis? Would margin-based rejection selection (like top-k by |∆_fact - ∆_pref|) improve over your simple prompt?
Q2: Can you provide any preliminary resutls on other task such as dialogue summarization or QA to demonstrate broader applicability?
Q3: Why not conduct human evaluation for at least some ablations (in Table 2) to validate that margins truly correlate with human judgments?

---

> ### Author Response · Authors · 2025-11-25
> **Response to Reviewer NXZW (1/4)**
>
> We appreciate your valuable comments and suggestions. We hope that the concerns can be resolved through our clarifications in this response, and all discussed revisions will be incorporated into the updated manuscript.
>
> ---
>
> > **Weakness 1.** *External evaluators (G-Eval/FineSurE/AlignScore) are potentially coupled with training signals, lacking robustness ablation studies such as leave-one-evaluator-out and replacement of evaluator families, as well as larger-scale, hierarchical human evaluation comparisons*
>
>
> Thank you for the helpful feedback to improve our paper. We address your comments in three parts: (Weakness 1.1) the independence of training signals from external evaluators, (Weakness 1.2) leave-one-evaluator-out and replacement of evaluator families for robustness, and (Weakness 1.3) larger-scale, hierarchical human evaluation comparisons.
>
> >> **(Weakness 1.1.)** *The independence of training signals from external evaluators.*
>
> We would like to clarify that the external evaluators (G-Eval, FineSurE, AlignScore) are used solely for assessing data quality during post-analysis and are not part of the training signal. Our framework does not rely on these evaluators and remains fully functional without them. Importantly, as highlighted in the Introduction, *our goal is to automatically generate rejected summaries with high alignment potential without depending on any external reward model.* Additionally, the training objective (e.g., DPO loss) relies exclusively on model preference signals (log-probability comparisons between chosen and rejected), and does not involve any external evaluator outputs. Therefore, there is no coupling between the evaluators and the training loss.
>
>
> >> **(Weakness 1.2.)** *Leave-one-evaluator-out and replacement of evaluator families for robustness.*
>
> *(i) Leave-one-evaluator-out.*
>
> All of our experiments that used external evaluators were conducted with each evaluator run independently (all for evaluation purpose), and their results were reported separately. Therefore, all experiments satisfy the leave-one-evaluator-out condition. If we have misunderstood your concern, we would appreciate it if you could clarify.
>
> *(ii) Replacement of evaluator families.*
>
> Currently, each of our metrics relies on a single evaluator: G-Eval uses GPT-4o-mini, FineSurE uses Llama3.3-70B-Instruct, and AlignScore is based on a dedicated pretrained RoBERTa-Large model. As AlignScore is tightly coupled to its original pretrained architecture, it is not designed to be replaced with alternative model families.
>
> Therefore, we conducted additional evaluation for main table (Table 4) using two evaluation metric (FineSurE, G-EVAL) using three evaluator model families (GPT-4o-mini, Llama3.3-70B-Instruct, Qwen3-Next-80B-A3B-Instruct) for each metric. In this analysis, we focus on SummLlama, since it consistently outperformed other baselines. The results are shown below (p-value in parentheses).
>
> (Backbone for Summllama and SPICE: Llama3.2-3B-Instruct)
> |Method|FineSurE:GPT|FineSurE:Llama|FineSurE:Qwen|G-Eval:GPT|G-Eval:Llama|G-Eval:Qwen|
> |------|:-----:|:--------:|:------:|:----------:|:----------:|:----------:|
> |`SummLlama`|0.788(0.00)|0.843(0.01)|0.766(0.00)|8.473(0.00)|8.102(0.00)|8.422(0.00)|
> |`SPICE`|**0.833**|**0.885**|**0.811**|**9.037**|**8.629**|**8.996**|
>
> (Backbone for Summllama and SPICE: Llama3.1-8B-Instruct)
> |Method|FineSurE:GPT|FineSurE:Llama|FineSurE:Qwen|G-Eval:GPT|G-Eval:Llama|G-Eval:Qwen|
> |------|:-----:|:--------:|:------:|:----------:|:----------:|:----------:|
> |`SummLlama`|0.865(0.00)|0.908(0.02)|0.854(0.00)|9.038(0.00)|8.796(0.00)|9.182(0.00)|
> |`SPICE`|**0.906**|**0.934**|**0.898**|**9.438**|**9.182**|**9.568**|
>
> The results show that SPICE consistently outperforms SummLlama across two evaluation metrics and three evaluator families, demonstrating the robustness of the results.

---

> ### Author Response · Authors · 2025-11-25
> **Response to Reviewer NXZW (2/4)**
>
> >> **(Weakness 1.3.)** *Larger-scale and hierarchical human evaluation comparisons.*
>
> Compared to prior baselines [1, 2, 3], our human evaluation was conducted at both a substantial scale and with a hierarchical structure. As shown in Table 8, we evaluated summaries from seven dataset domains spanning dialogue, non-dialogue, short-form, and long-form documents, with each summary containing roughly six sentences on average. Each sentence was annotated independently by three annotators, and the final summary-level factuality score was computed through majority voting over sentence-level judgments. This constitutes a hierarchical and fine-grained evaluation protocol: factuality is assessed at the sentence level and then aggregated to the summary level, mirroring the human annotation procedure used in FineSurE. We believe that this setup satisfies both large-scale coverage and hierarchical evaluation, ensuring that our results are both reliable and comprehensive.
>
> In particular, the table below demonstrates that our human evaluation covers a broader and more diverse range of data compared to prior work [1, 2, 3]. Here, Human Eval Size indicates the number of summaries evaluated by humans within a single test set. SPICE conducts evaluation on the largest number of datasets (7), with 100 human-evaluated summaries per dataset which is higher than any other method. Each summary is assessed by three independent annotators and evaluated at the sentence level, offering a more fine-grained analysis. These aspects together demonstrate that SPICE provides the most quantitative and fine-grained human evaluation among the baselines.
>
> | Method | Human Eval Size | # of Datasets | # of Annotators | Granuality |
> |-----------------|:-------:|:------:|:-----:|:-----:|
> | `CLIFF`  | 100 | 2 | 2 | summary-level |
> | `SYNFAC-EDIT`  | 10  | 1 | 3 | sentence-level |
> | `SummLlama`  | 70  | 7 | 3 | sentence-level |
> | `SPICE (Ours)`  | 100 | 7 | 3 | sentence-level |
>
> We acknowledge that we did not describe this aspect in sufficient detail, and we will clarify it more explicitly in the revised manuscript.
>
> [1] CLIFF: Contrastive Learning for Improving Faithfulness and Factuality
> in Abstractive Summarization, In EMNLP, 2021.
>
> [2] SYNFAC-EDIT: Synthetic Imitation Edit Feedback for Factual Alignment in Clinical Summarization, In EMNLP, 2024.
>
> [3] Learning to Summarize from LLM-generated Feedback, In NAACL, 2025.
>
> ---
>
> > **Weakness 2.** *One concerns is whether there should be more ablation studies. For example, a comparison between data-level length control (equal-length vs no control) against loss-level normalization (e.g., SimPO/KTO variants)*
>
> We present additional experiments to compare data-level length control (equal-length vs. no control) against loss-level normalization methods (SimPO and KTO). In this setup, the "Default" configuration refers to no length control, while "SPICE" denotes equal-length filtering based on margin alignment. Using these two data configurations, we fine-tuned Llama3.1–8B-Instruct with three alignment optimization methods(DPO, KTO, and SimPO). The results are presented in the table below (p-value in parentheses).
>
>
> | Method | FineSurE | G-Eval | AlignScore |
> |-----------------|:-------:|:------:|:-----:|
> | `DPO-Default`  | 0.902(0.00) | 8.916(0.00) | 0.843(0.00) |
> | `KTO-Default`  | 0.886(0.00) | 8.857(0.00) | 0.838(0.00) |
> | `SimPO-Default`  | 0.908(0.00) | 9.081(0.00) | 0.852(0.00) |
> | `DPO-SPICE`  | 0.934(0.09) | 9.438(0.13) | 0.895(0.07) |
> | `KTO-SPICE`  | 0.912(0.02) | 9.178(0.01) | 0.857(0.01) |
> | `SimPO-SPICE`  | **0.945** | **9.512** | **0.908** |
>
> The results demonstrate that data-level length control using SPICE consistently improves model performance across all optimization methods (even with SimPO), highlighting its effectiveness as a data construction strategy.
>
> As shown in Appendix G.8, among the three optimization methods, SimPO yields sligtly higher scores than DPO (both in the Default and SPICE settings), likely due to its effective framework that removes the bias of the reference model. However, since the differences are not statistically significant (p > 0.05), the improvement may not be substantial.
>
> Despite the suboptimal setup for KTO, where the smaller batch size may have limited its performance, we observe that KTO-SPICE still outperforms all Default settings across all evaluation metrics. This result highlights the effectiveness of SPICE in enhancing model faithfulness regardless of the optimization method used.
>
> ---

---

> ### Author Response · Authors · 2025-11-25
> **Response to Reviewer NXZW (3/4)**
>
> > **Question 1.** *Why not implement actual adversarial sampling based on your theoretical analysis? Would margin-based rejection selection (like top-k by $\vert\Delta_{fact}-\Delta_{pref}\vert$) improve over your simple prompt?*
>
> Our fundamental goal is to generate high–alignment-potential rejected summaries without relying on any external reward model. Because typical generation distributions rarely yield such samples, even adversarial or top-k selection cannot guarantee high alignment potential. This motivates directly generating high-quality rejected summaries rather than filtering them afterward, emphasizing that quality matters more than quantity.
>
> **Analysis 1. Limitations of Adversarial Sampling**
>
> To validate this, we conducted adversarial sampling on the full FeedSum seed set generated with the basic prompt. For each of the 7,491 unique documents (each accompanied by 13 rejected-summary candidates), we selected the top-1 candidate with the largest $\vert\Delta_{fact}-\Delta_{pref}\vert$, ensuring that the resulting dataset matches the size of SPICE for a fair comparison.
>
> The table below summarizes the factuality margin, preference margin, and alignment potential. Notably, even after selecting the best candidate among 13 candidates, the alignment potential remains substantially lower than that of SPICE. This suggests that large pools generated by unoptimized prompts, like SummLlama, struggle to produce rejected summaries with sufficiently high alignment potential, making it difficult to obtain effective rejections through quantity alone.
>
> (External Evaluator: Llama3.1-70B, Target Policy: Llama3.1-3B)
> | Method | $\Delta_{fact}$ | $\Delta_{pref}$|$\vert\Delta_{fact}-\Delta_{pref}\vert$|
> |-----------------|:-------:|:------:|:-----:|
> | `SPICE`  |   102.0 |  28.7  | 74.3  |
> | `Adv_Samplng`  |   58.5  |  13.4  | 45.1  |
>
> **Analysis 2. Inefficiency of Adversarial Sampling**
>
> The table below presents the DPO training results using the adversarially sampled dataset (p-value in parenthesis). We observe that its performance is comparable to only 1/16 of what SPICE achieves, as reported in Table 6. This implies that adversarial sampling would require generating over 200× more rejected candidates (13 candidates × 16× gap) to reach performance comparable to SPICE. This is extremely costly in practice.
>
> These findings suggest that achieving strong performance through adversarial sampling demands an extremely large candidate pool, which is not feasible in realistic settings. In contrast, SPICE provides an efficient and effective alternative: it produces high-quality rejected summaries without requiring massive cost, making it far more practical for real-world preference optimization. For these reasons, we did not adopt adversarial sampling as part of our main approach.
>
> (Backbone for SPICE and Adv_Sampling): Llama3.2-3B-Instruct)
> | Method | FineSurE | G-Eval | AlignScore |
> |-----------------|:-------:|:------:|:-----:|
> | `SPICE`  | 0.885 | 9.037 | 0.854 |
> | `SPICE_1/16`  | 0.846(0.00) | 8.503(0.00) | 0.812(0.00) |
> | `Adv_Samplng`  | 0.848(0.00) | 8.551(0.00) | 0.818 (0.00) |
>
> ---
>
> > **Question 2.** *Can you provide any preliminary resutls on other task such as dialogue summarization or QA to demonstrate broader applicability?*
>
> **(i) Dialogue Summarization**
>
> We note that our all experiments already include dialogue summarization (see Table 8). As shown in Table 11, we additionally evaluate every baseline on multiple dialogue datasets (DialogSum, MediaSum, MeetingBank). The results reproduced below demonstrate that SPICE achieves consistent and significant improvements, further supporting the general applicability of our method within the summarization domain.
>
>
> | Method           | DialogSum   | MediaSum    | MeetingBank |
> | -------- | ----------- | ----------- | ----------- |
> | Zero-Shot        | 0.685(0.00) | 0.785(0.00) | 0.751(0.00) |
> | SFT              | 0.726(0.01) | 0.819(0.01) | 0.765(0.00) |
> | CLIFF-Entity     | 0.666(0.00) | 0.839(0.00) | 0.766(0.00) |
> | CLIFF-Mask       | 0.710(0.00) | 0.809(0.00) | 0.778(0.00) |
> | SummLlama        | 0.734(0.02) | 0.846(0.01) | 0.779(0.00) |
> | SYNFAC-EDIT      | 0.658(0.00) | 0.821(0.00) | 0.771(0.00) |
> | **SPICE (Ours)** | **0.768**   | **0.882**   | **0.825**   |
>
>
> **(ii) Question Answering**
>
> Extending SPICE from summarization to QA is certainly a promising direction, and we believe the underlying idea could be beneficial beyond summarization. However, such an extension is non-trivial. Our current method focuses on summary factors (hallucination level, summary length, and prompt complexity) which do not directly transfer to QA. Adapting SPICE to QA would require re-examining these factors, revising the prompt, and identifying appropriate datasets and evaluation metrics for specific alignment purpose. Conducting this investigation thoroughly within the scope and timeline of the current work is challenging, so we leave a full QA extension to future research.
>
> ---

---

> ### Author Response · Authors · 2025-11-25
> **Response to Reviewer NXZW (4/4)**
>
> > **Question 3.** *Why not conduct human evaluation for at least some ablations (in Table 2) to validate that margins truly correlate with human judgments?*
>
> The three automatic factuality metrics used in our study (FineSurE, G-Eval, and AlignScore) are all recent, verified, SOTA evaluators in summarization domain. They show strong agreement with human judgments, achieving balanced accuracy up to 0.92 and Pearson correlations over 0.84 in faithfulness evaluation [4,5,6].
>
> Across the three evaluators, prior work reports that FineSurE improves correlation with human judgments over traditional metrics such as ROUGE [7] and BERTScore [8] by average margins of 0.52 and 0.74, respectively. Similarly, G-Eval shows gains of 0.44 and 0.49, while AlignScore achieves improvements of 0.24 and 0.31 over the same baselines.
>
> Importantly, in Table 2, the ‘Simple’ setting provides clear and consistent improvements over other settings across all three factuality metrics. All configurations show gains of at least 2.6%p (FineSurE: 0.859 → 0.885). In Table 4, the smallest automatic evaluation gap is also 2.6%p (FineSurE: 0.908 → 0.934). At this point, the human evaluation still reveals a statistically significant difference (p < 0.05). This indicates that automatic evaluation gaps of at least 2.6%p were judged as significant gaps (p < 0.05) in human evaluation too. Since this validation already established that the automatic metrics reliably reflect human judgement, and given the substantial cost of human annotation, we chose not to perform human evaluation for the ablation settings.
>
>
> [4] AlignScore: Evaluating Factual Consistency with A Unified Alignment Function, In ACL, 2023
>
> [5] G-Eval: NLG Evaluation using Gpt-4 with Better Human Alignment, In EMNLP, 2023
>
> [6] FineSurE: Fine-grained Summarization Evaluation using LLMs, In ACL, 2024
>
> [7] ROUGE: A package for automatic evaluation of summaries. In Text Summarization Branches Out, 2004.
>
> [8] BERTScore: Evaluating text generation with bert. In ICLR, 2019.
>
> ---
>
> If there are any aspects of our response that remain unclear or could benefit from further clarification, we would be more than happy to address them.
>
> ---

---

> ### Author Response · Authors · 2025-11-26
> **Acknowledgment for Reviewer's Reassessment**
>
> We sincerely appreciate your careful reconsideration of our paper. Thank you for raising your rating from 4 to 6 after reviewing our rebuttal. We are grateful that our clarifications helped address your concerns. Please do not hesitate to request any further information or clarification during the remaining rebuttal period.
>
> Sincerely, Authors

---

### Official Review · Reviewer_y2QS · 2025-10-31

**Soundness:** 3
**Presentation:** 3
**Contribution:** 3
**Rating:** 6
**Confidence:** 3

**Summary:**

This paper analyzes the effects of and proposes a strategy for choosing rejected data for preference learning to improve summarization faithfulness. It first show through theoretical analysis that using preference pairs with high alignment potential (i.e. when there is a large gap between model likelihood and actual factuality of the chosen vs. rejected data) leads to faster convergence than selecting pairs randomly. Then, it analyzes three factors that impact alignment potential, including hallucination level, response length and generation prompt analysis. Based on the observations, the paper proposes SPICE, a strategy of creating rejection summaries with high hallucination level, comparable length with the accepted summaries, and with simple generation prompt. Results show that the proposed strategy consistently improves the summarization faithfulness.

**Strengths:**

* The paper analyzes rejection data selection for summarization faithfulness in a systematic and principled way. The analysis framework includes theoretical justifications, and considers various factors that can impact the end performance.
* The proposed strategy based on the observations is straightforward and lightweight, making it practical and easy-to-apply.
* Results show that it is versatile and robust, and can consistently improve performance gains across summarization models, rejection data generators and preference learning algorithms.
* Results also show that the proposed method is data efficient, and can bring performance gain even with a subset of the training data.

**Weaknesses:**

The Factor 3 Prompt Complexity analysis is a bit coarse-grained. The simple vs. complex prompts seem to be at the two ends of extreme, where the complex prompt requires the generator to have multiple types of edits at the same time in each summary. It would be helpful to see intermediate comparisons (e.g. performing some but not all of the constraints), and the impact of each individual edit instruction, etc. essentially to analyze whether it's unhelpful to have any kind of instructions, or is it that having all of the instructions at the same time is too extreme but having some of them could still be helpful. (See Questions below).

**Questions:**

For Factor 3 Prompt Complexity:
* Among the different instructions for introducing hallucinations (e.g., numerical values, entity names, causal relationships), do all of these lead to trivial cues that models can easily distinguish, or could some of them be useful?
* In some sense, the complex instruction in Table 25 makes the generator to include all specified edits when generating rejected data. Since it is unlikely for all such errors to appear simultaneously in the same summary, it is not unexpected that they will lead to high preference margin. Could this make this comparison too trivial to conclude that complex prompts are less preferable? It would be helpful to see, e.g. instead present these edit instructions as suggestions and allow the generator to decide which to include, rather than asking it to have all of them?
* The complex prompt explicitly say “nearly all words should be edited”. Could this inadvertently push the generator towards producing less natural or uncommon phrasing, even if the outputs appear coherent on the surface?
* It would be helpful to provide some discussion or analysis on how the hallucination edit types look like in the generations with the "Default" prompt (Table 23), e.g. do they have the same categories as the ones listed in the complex prompt (Table 25) and how are they distributed, or do they fall outside these categories?

---

> ### Author Response · Authors · 2025-11-25
> **Response to Reviewer y2QS (1/2)**
>
> We appreciate your valuable comments and suggestions. We hope that the concerns can be resolved through our clarifications in this response, and all discussed revisions will be incorporated into the updated manuscript.
>
> ---
> > **Weakness 1.** *The Factor 3 Prompt Complexity analysis is a bit coarse-grained. The simple vs. complex prompts seem to be at the two ends of extreme, where the complex prompt requires the generator to have multiple types of edits at the same time in each summary. It would be helpful to see intermediate comparisons (e.g. performing some but not all of the constraints), and the impact of each individual edit instruction, etc. essentially to analyze whether it's unhelpful to have any kind of instructions, or is it that having all of the instructions at the same time is too extreme but having some of them could still be helpful. (See Questions below).*
>
> Thank you for the helpful comment. Since Weakness 1 encompasses the four questions (Question 1 through Question 4) you raised, we address each of them separately below.
>
> >> ***Question 1.** Among the different instructions for introducing hallucinations (e.g., numerical values, entity names, causal relationships), do all of these lead to trivial cues that models can easily distinguish, or could some of them be useful?*
>
> In Table 25, the complex prompt consists of eight edit instructions. Since length preservation (Instructions 3–4) is a core component of SPICE, we keep this constraint fixed by default. We then group the remaining instructions into five categories:
>
> (1) Heavy hallucination induction (Instructions 1–2) \
> (2) Numerical alterations (Instruction 5) \
> (3) Named-entity substitutions (Instruction 6) \
> (4) Causal relation edits (Instruction 7) \
> (5) Preserving grammar and fluency (Instruction 8)
>
>
> **Analysis 1. Effect of Each Instruction**
>
> We perform ablations over these five categories while keeping length preservation fixed. The results are shown below. For the heavy hallucination category, the factuality margin increases, however, the preference margin rises by a similar degree, making the rejected summaries easier for the model to distinguish. As a consequence, the overall alignment potential (55.0) becomes lower than that of SPICE (69.9). For the remaining instruction categories(2-5), some variants reduce the preference margin (making the rejection harder to distinguish), but this benefit is offset by a simultaneous drop in factuality margin. As a result, all ablated variants yield a lower alignment potential compared to the original SPICE.
>
> (External Evaluator: Llama3.1-70B, Target Policy: Llama3.1-8B)
> | Method | $\Delta_{fact}$ | $\Delta_{pref}$|$\vert\Delta_{fact}-\Delta_{pref}\vert$|
> |-----------------|:-------:|:------:|:-----:|
> | `SPICE`  |   102.0  |  32.1  | **69.9** |
> | `SPICE+(1)`  |   120.3  |  65.3  | 55.0 |
> | `SPICE+(2)`    |   80.3  |  31.6  | 48.7 |
> | `SPICE+(3)`   |   86.1  |  33.2  | 52.9 |
> | `SPICE+(4)`     |   79.5  |  25.4  | 54.1 |
> | `SPICE+(5)`  |   76.7 |  38.3  | 38.4 |
>
> **Analysis 2. Effect of Increasing Prompt Complexity**
>
> Additionally, we extend the default SPICE setting by incrementally adding each instruction category to provide intermediate comparisons, ranging from simple to more complex prompts. The final configuration with all five instruction types included corresponds exactly to the Complex setting. As shown in the table below, while the factuality margin remains roughly constant or increases slightly as more instructions are added, the preference margin grows substantially. This makes the rejected summaries progressively easier for the model to distinguish, ultimately reducing alignment potential.
>
> These results indicate that increasing the number of instructions consistently enlarges the preference margin, which is undesirable. This supports our design choice in SPICE: allowing the model to introduce errors freely, without enforcing specific edit types, produces harder and more informative negative examples.
>
> (External Evaluator: Llama3.1-70B, Target Policy: Llama3.1-8B)
> | Method | $\Delta_{fact}$ | $\Delta_{pref}$|$\vert\Delta_{fact}-\Delta_{pref}\vert$|
> |-----------------|:-------:|:------:|:-----:|
> | `SPICE`  |   102.0  |  32.1  | **69.9** |
> | `SPICE+(1)`  |   120.3  |  65.3  | 55.0 |
> | `SPICE+(1)+(2)`    |   118.6  |  68.4  | 50.2 |
> | `SPICE+(1)+(2)+(3)`   |   119.6  |  72.6  | 47.0  |
> | `SPICE+(1)+(2)+(3)+(4)`     |   122.7  |  78.9  | 43.8  |
> | `SPICE+(1)+(2)+(3)+(4)+(5)`  |  126.5 |  85.3  | 41.2  |

---

> ### Author Response · Authors · 2025-11-25
> **Response to Reviewer y2QS (2/2)**
>
> >> **Question 2.** In some sense, the complex instruction in Table 25 makes the generator to include all specified edits when generating rejected data. Since it is unlikely for all such errors to appear simultaneously in the same summary, it is not unexpected that they will lead to high preference margin. Could this make this comparison too trivial to conclude that complex prompts are less preferable? It would be helpful to see, e.g. instead present these edit instructions as suggestions and allow the generator to decide which to include, rather than asking it to have all of them?
>
> **Analysis 3. Effect of Relaxing Complex Instructions**
>
> We extend the original complex prompt (Table 25) by adding the following instruction to increase the freedom with which hallucinations can be introduced:
>
> ```
> The following instructions are just examples and do not need to be followed entirely; you may apply them selectively and flexibly as long as the overall summary remains factually inconsistent.
> ```
>
> As shown in the table below, introducing this flexibility reduces the preference margin more than the factuality margin, resulting in an improvement in alignment potential than the original Complex one. However, the overall alignment potential remains lower than that achieved by SPICE, which allows hallucinations to emerge more naturally without enforcing specific edit types.
>
> (External Evaluator: Llama3.1-70B, Target Policy: Llama3.1-8B)
> |Method|$\Delta_{fact}$|$\Delta_{pref}$|$\vert\Delta_{fact}-\Delta_{pref}\vert$|
> |-|:-:|:-:|:-:|
> |`SPICE`|102.0|32.1|69.9|
> |`Complex`|126.5|85.3|41.2|
> |`Complex-Flexible`|114.2|57.6|56.6|
>
> >> ***Question 3.** The complex prompt explicitly say “nearly all words should be edited”. Could this inadvertently push the generator towards producing less natural or uncommon phrasing, even if the outputs appear coherent on the surface?*
>
> We further modify the SPICE+(1) setting by removing the instruction “nearly all words should be edited,” and denote this variant as SPICE+(1A). As expected, reducing the required extent of edits results in less aggressive hallucinations, which in turn leads to a smaller preference margin and therefore a higher alignment potential. This indicates that the removed instruction was unnecessarily aggressive and primarily contributed to unnatural phrasing rather than generating informative hallucinations. However, the alignment potential still fell short of that achieved by SPICE.
>
> (Evaluator: Llama3.1-70B, Policy: Llama3.1-8B)
> |Method|$\Delta_{fact}$|$\Delta_{pref}$|$\vert\Delta_{fact}-\Delta_{pref}\vert$|
> |-|:-:|:-:|:-:|
> |`SPICE`|102.0|32.1|69.9|
> |`SPICE+(1)`|120.3|65.3|55.0|
> |`SPICE+(1A)`|108.7|49.4|59.3|
>
> >> **Question 4.** It would be helpful to provide some discussion or analysis on how the hallucination edit types look like in the generations with the "Default" prompt (Table 23), e.g. do they have the same categories as the ones listed in the complex prompt (Table 25) and how are they distributed, or do they fall outside these categories?
>
> **Analysis 4. Distribution of Hallucination Categories**
>
> We defined the four error types from the Complex prompt (Table 25) and modified the FineSurE evaluation prompt (Table 26) to detect these categories by replacing the original error labels. This allows us to compare the distributions of error categories between SPICE-generated data and Complex-generated data.
>
> ```
> - Numerical errors: Incorrectly modifying numerical values (e.g., dates, quantities, percentages) in contradiction to the source document.
> - Entity errors: Replacing people, places, or organizations with incorrect but plausible alternatives.
> - Causal-relation errors: Altering cause–effect relations to assert statements that contradict the document.
> - Grammatical and fluency errors: Reducing grammatical correctness or fluency, leading to unnatural or awkward phrasing.
> - Other errors: Factuality error which is not defined here.
> ```
>
> The results in the table below show a clear contrast between the two settings. Note that since neither setting contains grammatical errors, we do not report the results in the table. Complex produces hallucinations that closely follow the instructed categories, with numerical, entity, and causal errors together accounting for about 70% of all cases. In contrast, SPICE yields a much broader and more diverse set of hallucination types, where “other” errors alone make up 56.5% of the detected instances. This indicates that SPICE induces a wider and more natural spectrum of factual inconsistencies, rather than concentrating on the predefined error categories, which likely contributes to its stronger alignment potential.
>
> |Setting|Numerical error|Entity error|Causal relation error|Other error|
> |-|:-:|:-:|:-:|:-:|
> |`SPICE`|9.8%|14.0%|19.7%|56.5%|
> |`Complex`|12.4%|35.7%|21.5%|30.4%|
>
> ---
>
> If there are any aspects of our response that remain unclear or could benefit from further clarification, we would be more than happy to address them.
>
> ---

---

### Official Review · Reviewer_jYGQ · 2025-11-01

**Soundness:** 3
**Presentation:** 3
**Contribution:** 2
**Rating:** 6
**Confidence:** 3

**Summary:**

This paper proposes a method for constructing preference data for training models that generate more faithful summaries. They hypothesize that data for preference optimization for faithful summaries can be chosen in a better way that emphasizes more realistic errors in factuality. They verify this hypothesis by testing models fine-tuned on samples with larger factuality margins (less true) and smaller preference margins (more preferable otherwise). They then develop a way to prompt models to generate these summaries investigating the factors that have the biggest influence on generating such summaries. They observe that models fine-tuned on their summaries are able to achieve higher factuality scores than prior work.

**Strengths:**

The work to verify the factuality hypothesis is good, and in general the paper is well written and presented. The method appears sound, and does generally improve on prior work. The experiments detailing which portions of the summaries and prompts are important appear well conducted.

**Weaknesses:**

1. I have some concerns regarding the metrics used for evaluation. Krippendorf's alpha (0.53) for the human evaluation scores is low enough that these evaluations should not be considered trustworthy. The usual lowest threshold for accepting annotated data is alpha=0.67. Additionally, is it verified in prior work that the other metrics are aligned with human preferences?

2. Some claims are not fully explained. In particular, the explanation given for table 5 regarding the preference margins is that SPICE has the biggest factuality margin and "relatively small preference margins". However, in the reported results, SPICE has the largest second largest preference margin, which is in contrast to this claim. It would be good to remove or further clarify this claim.

**Questions:**

1. Is it verified in prior work that other metrics are aligned with human preferences?

2. Is there a reason why inter annotator agreement for the human evaluation is so low?

---

> ### Author Response · Authors · 2025-11-25
> **Response to Reviewer jYGQ (1/2)**
>
> We appreciate your valuable comments and suggestions. We hope that the concerns can be resolved through our clarifications in this response, and all discussed revisions will be incorporated into the updated manuscript.
>
> ---
>
> > **Weakness 1-1.** *I have some concerns regarding the metrics used for evaluation. Krippendorf's alpha (0.53) for the human evaluation scores is low enough that these evaluations should not be considered trustworthy. The usual lowest threshold for accepting annotated data is alpha=0.67. (Same with **Question 2:** Is there a reason why inter annotator agreement for the human evaluation is so low?)*
>
> We would like to clarify that the statement $α=0.67$ as the lowest acceptable threshold is not a universal criterion. Krippendorff explicitly notes that acceptable agreement levels must be interpreted in relation to the subjectivity and difficulty of the annotation task, and that no single numerical cutoff applies uniformly across domains. In open-ended NLP evaluations such as factuality and faithfulness judgments, it is widely acknowledged that IAA tends to be substantially lower. In text summarization and related evaluation settings, agreement values around $α ≈ 0.5$ are commonly considered reliable and are frequently reported in recent studies [1–5].
>
> Our human evaluation follows a challenging protocol in which each summary is segmented into individual sentences and annotated independently by three annotators. This setup is especially demanding because our datasets contain long and domain-specific documents including medical and government reports. Prior work using comparable sentence-level factuality annotation settings reports $α = 0.40$ in CLIFF [1], $0.49$ in BERTMULTI [2], $0.49$ in SYNFAC-EDIT [3], $0.38$ in TofuEval [4], and $0.58$ in MSumBench [5]. Our $α = 0.53$ is therefore situated at the higher end of this distribution and reflects strong inter-annotator consistency given the inherent difficulty of the task. We clarify this point in the revised manuscript.
>
> [1] CLIFF: Contrastive Learning for Improving Faithfulness and Factuality
> in Abstractive Summarization, In EMNLP, 2021.
>
> [2] Annotating and Detecting Fine-grained Factual Errors for Dialogue Summarization. In ACL, 2023.
>
> [3] SYNFAC-EDIT: Synthetic Imitation Edit Feedback for Factual Alignment in Clinical Summarization, In EMNLP, 2024.
>
> [4] TofuEval: Evaluating Hallucinations of LLMs on Topic-Focused Dialogue Summarization. In NAACL, 2024.
>
> [5] Towards Multi-dimensional Evaluation of LLM Summarization across Domains and Languages. In ACL, 2025.
>
> ---
>
> > **Weakness 1-2.** *Additionally, is it verified in prior work that the other metrics are aligned with human preferences? (Same with **Question 1:** Is it verified in prior work that other metrics are aligned with human preferences?)*
>
> The three automatic factuality metrics used in our study (FineSurE, G-Eval, and AlignScore) are all recent, verified, SOTA evaluators in summarization domain. They show strong agreement with human judgments in factuality evaluation, achieving a balanced accuracy up to 0.92 and a Pearson correlations coefficient over 0.84 [6, 7, 8].
>
> Across the three evaluators, prior work reports that FineSurE improves correlation with human judgments over traditional metrics such as ROUGE [9] and BERTScore [10] by average margins of 0.52 and 0.74, respectively. Similarly, G-Eval shows gains of 0.44 and 0.49, while AlignScore achieves improvements of 0.24 and 0.31 over the same baselines.
>
> Therefore, the metrics align with human judgments far more strongly than traditional metrics and are now widely used as practical substitutes for human evaluation in dataset construction, model assessment, and benchmarking for text summarization.
>
>
> [6] AlignScore: Evaluating Factual Consistency with A Unified Alignment Function. In ACL, 2023.
>
> [7] G-Eval: NLG Evaluation using Gpt-4 with Better Human Alignment. In EMNLP, 2023.
>
> [8] FineSurE: Fine-grained Summarization Evaluation using LLMs. In ACL, 2024.
>
> [9] ROUGE: A package for automatic evaluation of summaries. In Text Summarization Branches Out, 2004.
>
> [10] BERTScore: Evaluating text generation with bert. In ICLR, 2019.
>
> ---

---

> ### Author Response · Authors · 2025-11-25
> **Response to Reviewer jYGQ (2/2)**
>
> > **Weakness 2.** *Some claims are not fully explained. In particular, the explanation given for table 5 regarding the preference margins is that SPICE has the biggest factuality margin and "relatively small preference margins". However, in the reported results, SPICE has the largest second largest preference margin, which is in contrast to this claim. It would be good to remove or further clarify this claim.*
>
>
> We acknowledge that our original description did not clearly represent our intended claim. The statement that SPICE has "the largest factuality margin and relatively small preference margins" was misleading. Our intended claim is that SPICE maximizes the margin discrepancy to achieve high alignment potential in Eq. (4), rather than having a small preference margin.
>
> In the original submission, we made an error in how the two margins were computed. Specifically, the factuality margin was derived from G-Eval scores, while the preference margin was computed from log probabilities. Because these two quantities lie on incompatible numerical scales, this made the margin discrepancy ($\Delta_{fact} - \Delta_{pref}$) theoretically invalid. As clarified in Section 3.1 of our paper, once the preference margin is defined through log probabilities, the factuality margin should also be computed using log probabilities from an external reward model to be meaningful [11, 12]. We acknowledge that the mismatched computation in the initial version was our mistake.
>
> To correct this, we recomputed the factuality margin using log probabilities from Llama3.1-70B (external reward model), ensuring that both margins share a consistent scale. Under this corrected formulation, SPICE exhibits the largest alignment potential among all baselines, as shown in the two updated tables. Therefore, our intended claim is validated: "SPICE achieves the greatest margin discrepancy that leads to the highest alignment potential," rather than "simply having the largest factuality margin and a small preference margin."
>
> (External Evaluator: Llama3.1-70B, Target Policy: Llama3.2-3B)
> | Method | $\Delta_{fact}$ | $\Delta_{pref}$|$\vert\Delta_{fact}-\Delta_{pref}\vert$|
> |-----------------|:-------:|:------:|:-----:|
> | `CLIFF-Entity`  |   47.6  |  28.5  | 19.1  |
> | `CLIFF-Mask`    |   69.6  |  41.9  | 27.7  |
> | `SYMFAC-EDIT`   |   45.2  |  16.3  | 28.9  |
> | `SummLlama`     |   56.7  |  16.5  | 40.2  |
> | `SPICE (Ours)`  |   102.0 |  28.7  | **74.3**  |
>
> (External Evaluator: Llama3.1-70B, Target Policy: Llama3.1-8B)
> | Method | $\Delta_{fact}$ | $\Delta_{pref}$|$\vert\Delta_{fact}-\Delta_{pref}\vert$|
> |-----------------|:-------:|:------:|:-----:|
> | `CLIFF-Entity`  |   47.6  |  29.6  | 18.0  |
> | `CLIFF-Mask`    |   69.6  |  48.3  | 21.3  |
> | `SYMFAC-EDIT`   |   45.2  |  19.3  | 25.9  |
> | `SummLlama`     |   56.7  |  16.4  | 40.3  |
> | `SPICE (Ours)`  |   102.0 |  32.1  | **69.9**  |
>
> We will replace Table 5 with these two corrected tables. We sincerely appreciate your constructive feedback. Your comments allowed us to identify and correct the oversight in our analysis. If anything remains unclear, please let us know.
>
> [11] A General Theoretical Paradigm to Understand Learning from Human Preferences, In ICML, 2024.
>
> [12] The Crucial Role of Samplers in Online Direct Preference Optimization, In ICLR, 2025.
>
> ---
>
> If there are any aspects of our response that remain unclear or could benefit from further clarification, we would be more than happy to address them.
>
> ---

---

### Meta-Review · Area_Chair_T25w · 2026-01-09

**Summary:**

The paper proposes a method to construct preference data with the goal of building a more factual summarization model using PO training. They propose to select data pairs that have high factuality margin, and a low preference margin (defined to be the difference in likelihood).

Two reviewers (jYGQ, y2QS) are positive about the paper; they highlight the reported improvements. The reviewers raise the following key concerns:
1. Reliability of the evaluation metrics used.
2. Reviewer NXZW raises issues of evaluation signals leaking into training (this is addressed during rebuttal).
3. The most serious concern was raised by reviewer v7Lg. First, the reviewer points out that the proposal of mining hard negatives is widely studied in prior literature (e.g. https://arxiv.org/pdf/2502.16825, https://arxiv.org/pdf/2508.04149, etc.) The paper essentially uses a similar idea, with a more complex prompting strategy specialized for summarization. The paper does not directly compare against these or contextualize their contributions with these existing works.

**Reviewer Concerns:**

The rebuttal addressed the following concerns:
1. Concern #2 from above, wrt to leaking evaluation signals during training.
2. Other minor concerns, wrt to framing, were resolved. For e.g. Reviewer jYGQ pointed out discrepancy in the paper's written claims and the reported results in the paper. The paper was updated during the rebuttal stage to address these changes.
3. Reviewer jYGQ raised the issue of the reliability of the evaluation. The automatic metrics used in this work were proposed in 2023 (2 metrics) and 2024 (1 metric), and not all were created for this exact summarization task. Therefore, the questions about their validity is legitimate. However, we acknowledge that using LLM-judges are the standard way of reporting benchmarking generation performance.

Outstanding issues:
1. Concern #3 outlined above remains unresolved.
2. Wrt evaluation: reviewer jYGQ raises the question of low inter-annotator agreement. While I agree with the rebuttal that acceptable agreement ranges depend on the subjectivity of the task, the task in this paper, i.e. factuality is quite objective and 0.53 agreement is low. Also, most of the prior works that the authors cite of similarly low agreement scores do not report inter-annotator agreement between humans for factuality.
3. Reviewer v7Lg also raised the concern that the \delta fact reported for the SYNFAC-EDIT  baseline is suspiciously low. While the rebuttal argues that they reproduce the margins of the prior work, the deltas are actually very different when using a reasonable factuality metric (G-Eval).
4. I am not convinced that using an external policy to compute delta between the probabilities of the two outputs is a reasonable metric for factuality. Therefore, the framing that the approach "maximizes factuality margin" is not convincing as the metric used may not capture factuality.

**Reviewer Scores:**

I believe the 2 reviewers with positive scores (jYGQ, y2QS) of 6 would have kept their scores.
The authors mention that reviewer NXZW increased their score to 6, however, I find no explicit evidence of this.
Reviewer v7Lg's main concerns remain after the response; they would have retained their score (2) as well.

---

### Decision · Program_Chairs · 2026-01-26

Reject